# A NOISE IS WORTH DIFFUSION GUIDANCE

Donghoon Ahn[1*]   Jiwon Kang[2*]   Sanghyun Lee[2†]   Jaewon Min[2†]   Wooseok Jang[2]

Minjae Kim[2]   Hyungwon Cho[2]   Sayak Paul[5]   SeonHwa Kim[3]   Eunju Cha[4‡]

Kyong Hwan Jin[3‡]   Seungryong Kim[2‡]

[1]UC Berkeley    [2]KAIST    [3]Korea University

[4]Sookmyung Women's University    [5]Hugging Face

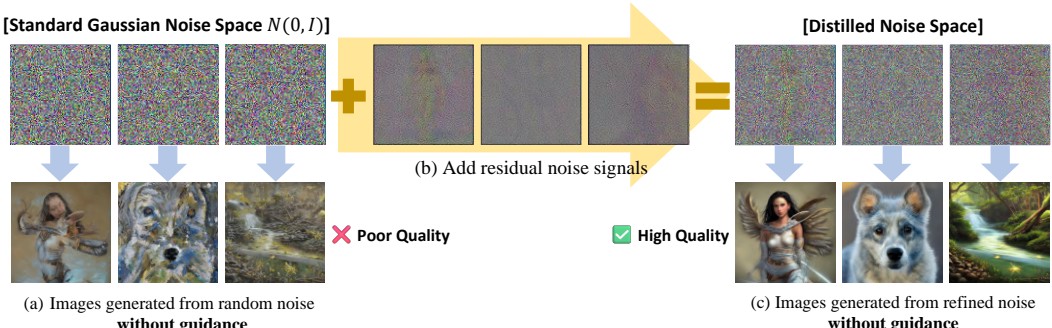

(b) Add residual noise signals

❌ Poor Quality          ✅ High Quality

(a) Images generated from random noise **without guidance**

(c) Images generated from refined noise **without guidance**

Figure 1: **Effectiveness of *NoiseRefine*.** Diffusion models often fail to generate high-quality images without guidance, such as classifier-free guidance (CFG) Ho & Salimans (2022), which doubles the inference cost. In this paper, we propose *NoiseRefine*, a novel approach to improve image quality by learning to map Gaussian noise space to guidance-distilled noise space. Images are generated using the same seed and prompt.

## ABSTRACT

Diffusion models have demonstrated remarkable image generation capabilities, but their performance heavily relies on sampling guidance such as classifier-free guidance (CFG). While sampling guidance significantly enhances image quality, it requires two forward passes at every denoising step, leading to substantial computational overhead. Existing approaches mitigate this cost through distillation, training a student network to learn the guided predictions. In contrast, we take a distinct approach by *refining the initial Gaussian noise*, a critical yet underexplored factor in the diffusion-based generation pipelines. We introduce a noise refinement framework, *NoiseRefine*, where a refining network is trained to minimize the difference between images generated by unguided sampling from the refined noise and those produced by guided sampling from the input Gaussian noise. This simple approach demonstrates that images from the refined noise alleviate artifacts and mitigate structural collapse, achieving significantly higher quality than those generated from pure Gaussian noise without modifying the diffusion model, thereby preserving its prior knowledge and compatibility with finetuned or timestep distilled variants. Beyond its practical benefits, we provide an in-depth analysis of refined noise, offering insights into its role in the denoising process and its interaction with guidance. Our findings suggest that structured noise initialization is key to efficient and high-fidelity image synthesis. Project page: cvlab-kaist.github.io/NoiseRefine.

---

*Equal contribution.

†Equally contributed as second authors.

‡Corresponding authors.

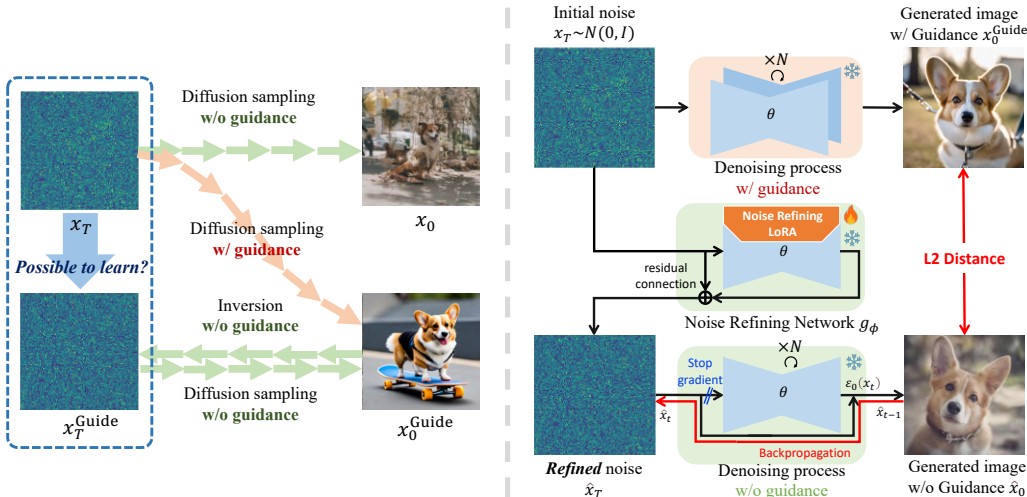

(a) Mapping between noise spaces       (b) Overview of training framework

Figure 2: **Motivation and training framework of *NoiseRefine*.** (a) Starting from an initial noise $x_T$, unguided sampling often produces low-quality images, necessitating sampling guidance such as CFG. In contrast, the inversion noise $x_T^{\text{Guide}}$, obtained by inverting guidance-generated images from the same $x_T$, can yield high-quality results even without guidance. This raises our central question: *can we learn to map $x_T$ into $\hat{x}_T$?* (b) Learning with a reconstruction loss between $x_T$ and $x_T^{\text{Guide}}$ may be suboptimal due to errors during inversion. Instead, our model learns to refine $x_T$ into $\hat{x}_T$, with the objective of matching unguided image $\hat{x}_0$ from refined noise $\hat{x}_T$ with guidance-generated image $x_0^{\text{Guide}}$ from initial noise $x_T$.

# 1 INTRODUCTION

In recent years, text-to-image (T2I) diffusion models (Rombach et al., 2022; Esser et al., 2024; Podell et al., 2023), which generate images conditioned on text prompts, have achieved remarkable advancements. These models produce visually appealing images that are both realistic and well-aligned with human perception. A central factor behind their effectiveness is the use of sampling guidance techniques (Dhariwal & Nichol, 2021; Ahn et al., 2024; Hong et al., 2023; Hong, 2024), most notably classifier-free guidance (CFG) (Ho & Salimans, 2022). While indispensable for high-quality synthesis, these methods require evaluating additional prediction (unconditional prediction in the case of CFG) at every denoising step, effectively doubling inference cost.

A common strategy to mitigate this overhead is *guidance distillation*, where a student network (Meng et al., 2023) or an adapter (Hsiao et al., 2024) is trained to approximate the guided predictions of the original model. However, such approaches often require modifications to the denoising network, which is prone to catastrophic forgetting (Kirkpatrick et al., 2017), and potentially incompatible with complementary techniques such as domain-specific fine-tuning (Ruiz et al., 2023) or timestep distillation Yin et al. (2024); Lin et al. (2024); Salimans & Ho (2022); Sauer et al. (2024b).

Recently, a growing line of work has explored the role of initial noise, suggesting that it can influence the final image structure to some extent (Singh et al., 2022; Wu & De la Torre, 2022; Mao et al., 2023a; Ban et al., 2024; Xu et al., 2024a; Qi et al., 2024; Guo et al., 2024; Eyring et al., 2024; Zhou et al., 2024; Mannering et al., 2025; Ma et al., 2025). Inspired by this, we ask: *Instead of distilling guidance into the denoising network, can we distill it into noise?* Diffusion inversion methods (Song et al., 2020a; Garibi et al., 2024) provide important clue. Ideally, a perfect inversion method would reconstruct a given image without requiring any guidance. Under this idealized assumption, a straightforward way to obtain the "guidance-free noise" target is to start from an initial Gaussian noise, generate a high-quality image using guidance, and then apply inversion to compute its corresponding noise. This resulting "inversion noise" should, in principle, reproduce a similar image without guidance. This conceptual idea is illustrated in Fig. 2(a).

However, we show that directly learning the mapping from Gaussian noise to inversion noise is suboptimal due to the accumulated reconstruction errors introduced by the inversion process (Sec. 3.2). To overcome this limitation, we shift the objective from the noise space to the image space and propose *NoiseRefine*, a novel method that refines Gaussian noise into informative and structured noise, enabling high-quality generation without guidance. As illustrated in Fig. 2 (b), a lightweight transformation network maps arbitrary Gaussian noise into the refined noise space, trained so that unguided samples closely match guided counterparts generated from the same seed. At inference, a single forward pass through this network substantially improves unguided generation quality, producing significantly more plausible images even without guidance, while preserving the original diffusion pipeline intact, in a prompt-learning-like manner (Zhou et al., 2022a) that avoids catastrophic forgetting from model fine-tuning (Kirkpatrick et al., 2017).

Beyond eliminating guidance, *NoiseRefine* offers several advantages. First, since it operates solely on the noise input, it can be directly applied to fine-tuned models in various domains (e.g. Anime) without retraining the refining network. Second, it remains fully compatible with timestep-distillation techniques (Meng et al., 2023; Hsiao et al., 2024; Zhou et al., 2025). Together, these properties make *NoiseRefine* a plug-and-play solution for enhancing base, fine-tuned, and timestep distilled models. We validate our approach on both class-conditional and widely used text-to-image diffusion models.

Our contributions can be summarized as follows:

- **Noise refinement for guidance-free generation**: To the best of our knowledge, this work is the first to explore refining initial noise in diffusion pipelines to achieve high-quality image generation without diffusion guidance.

- **Preserving the diffusion pipeline**: Our method does not modify the original diffusion model or pipeline, which ensures compatibility with LoRA modules in the original pipeline, generalizes well to fine-tuned models, and seamlessly integrates with existing timestep-distillation techniques.

- **Thorough analysis of refined noise in diffusion models**: We provide a detailed study on the role of refined noise in the denoising process, offering insights into their impact on generation quality.

## 2 RELATED WORK

**Diffusion guidance.** Classifier Guidance (CG) (Mao et al., 2023a) enhances fidelity by leveraging trained classifier gradients, albeit at the cost of diversity. CFG (Ho & Salimans, 2022) models an implicit classifier to achieve similar effects. Ahn et al. (Ahn et al., 2024) and Karras et al. (Karras et al., 2024) further generalize those guidance methods by intentionally generating lower-quality samples to guide the process toward improved outputs and other guidance techniques (Hong et al., 2023; Sadat et al., 2024; Hong, 2024) generate 'perturbed' samples in various ways. While effective, these methods double computational and memory costs by requiring degraded sample generation at each step, which is essential to their operation.

**Distillation of diffusion models.** Diffusion models are costly at inference due to guidance and iterative denoising. A line of work distills teacher models into lighter students (Salimans & Ho, 2022; Meng et al., 2023; Sauer et al., 2024b; Lin et al., 2024; Sauer et al., 2024a), targeting fewer steps (timestep distillation) (Salimans & Ho, 2022; Sauer et al., 2024b;b;a) or cheaper guidance (guidance distillation) (Meng et al., 2023), with extensions via adapters (Hsiao et al., 2024) or prompt distillation (Zhou et al., 2025). In contrast, while existing guidance distillation approaches transfer guidance signals into the student network, we distill guidance directly into the initial noise of diffusion models, making our method fully compatible with timestep distillation.

**Noise optimization.** Recent studies have explored improving the initial noise through optimization, reinitialization, or task-specific refinements (Samuel et al., 2024; Eyring et al., 2024; Mao et al., 2023b;a; Karunratanakul et al., 2024; Zhou et al., 2024; Guo et al., 2024; Mannering et al., 2025; Ma et al., 2025). Approaches include reward-model optimization (Eyring et al., 2024), bootstrap sampling for rare concepts (Samuel et al., 2024), patch databases for layout control (Mao et al., 2023b), iterative disentanglement (Guo et al., 2024), and systematic noise search (Ma et al., 2025). Recent work (Zhou et al., 2024) trains a noise transformation network using CFG and inversion, optimizing it in noise space with inversion noise as the supervision target. In contrast, we train the network with an image-space loss and focus on guidance-free generation rather than human preference alignment.

## 3 METHOD

In Sec 3.1, we analyze the differences between the original initial noise and the inversion noise obtained via guidance-generation followed by inversion. In Sec. 3.2, we address the errors introduced during the inversion and propose to learn in the image space rather than the noise space. Finally, Sec 3.3 presents our complete training framework, incorporating a multi-step score distillation loss that mitigates the cost of backpropagation.

### 3.1 DIFFERENCE BETWEEN INITIAL NOISE AND INVERSION NOISE

When an image generated with guidance is inverted back to its noise using an inversion method (Song et al., 2020a), the recovered "inversion noise" tends to reproduce a similar image even without guidance, as described in Fig. 2 (a). We investigate the relationship between the initial noise $x_T$ and the inversion noise $x_T^{\text{Guide}}$ of a guidance-generated image, as their differences may explain the gap in the quality of denoised outputs. To this end, we sample Gaussian noise $x_T \sim \mathcal{N}(0, I)$, generate a guided image $x_0^{\text{Guide}} = \text{Denoise}^{\text{Guide}}(x_T, c)$ using a text-to-image diffusion model (Rombach et al., 2022) with CFG and/or other guidance methods (Ho & Salimans, 2022; Ahn et al., 2024; Hong, 2024), and then apply an inversion method (Song et al., 2020a; Garibi et al., 2024; Meiri et al., 2023) to obtain the corresponding inversion noise $x_T^{\text{Guide}} := \text{Inversion}(x_0^{\text{Guide}})$. Both $x_0^{\text{Guide}}$ and $x_T^{\text{Guide}}$ depend on the condition $c$, which we omit for notational simplicity. This procedure yields paired samples $(x_T, x_T^{\text{Guide}})$ for subsequent analysis. Notation definitions are provided in Appendix B.1.

We generate 10K $\{x_T, x_T^{\text{Guide}}\}$ pairs via the aforementioned process with randomly selected prompts from the MS-COCO dataset (Lin et al., 2014) and Stable Diffusion 2.1 (Rombach et al., 2022). Comparing the pixel-wise absolute differences between $x_T$ and $x_T^{\text{Guide}}$ to those between random noise instances, Fig. 3 (a) shows that the differences in $\{x_T, x_T^{\text{Guide}}\}$ pairs are significantly smaller than those of 'Random' pairs. These differences are primarily concentrated in low-frequency components in the frequency domain, as shown in Fig. 3 (b), which plots the magnitude differences between Fourier-transformed noises. This indicates that the discrepancy between the initial noise and inversion noise is structured rather than random. Such structured, low-frequency deviations imply the existence of a consistent and potentially learnable mapping between the two noise distributions. If this mapping can be effectively modeled, it may provide a pathway to generating higher-quality samples without explicit guidance during sampling.

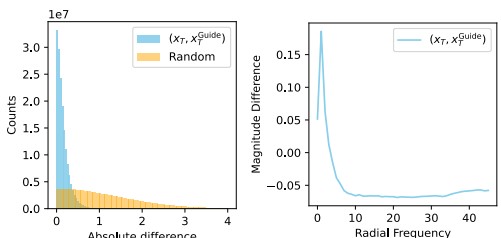

Figure 3: **Analysis of the relationship between $x_T$ and $x_T^{\text{Guide}}$.** (a) Histogram of pixel-wise absolute differences. Blue: pairs of Gaussian noise and corresponding inversion noise; Orange: pairs of random Gaussian noise. (b) Magnitude difference of Fourier components, showing that $x_T$ and $x_T^{\text{Guide}}$ mainly differ in low-frequency regions.

### 3.2 LEARNING IN IMAGE SPACE RATHER THAN NOISE SPACE

**Mitigating inversion error.** A straightforward approach would be to directly learn a mapping from the initial noise to the corresponding inversion noise. While this approach is conceptually feasible, inversion methods (Song et al., 2020a; Meiri et al., 2023; Garibi et al., 2024) have inherent limitations: they rely on approximations, and the ideal (true) inversion noise $x_T^{\text{Guide}\dagger}$, defined as the noise that would perfectly reconstruct the guided image, is not directly accessible in practice due to approximation errors in existing inversion methods. As a result, training on approximated inversion noise corrupted by inversion errors may limit performance (Fig. 4).

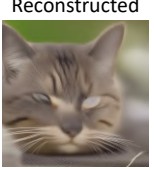

Original    Reconstructed

Figure 4: **Inversion error.** The right image is reconstructed from the inversion noise of the left one. 50 inversion steps were used.

In practice, we trained two noise refining networks on 10K guided images and 10K inverted noises from a fixed prompt "a photo of a corgi riding a skateboard." As shown in Fig. 5 top row, directly learning this mapping produces blurry reconstructions.

To sidestep this issue, we optimize the objective in the image space rather than directly in the noise space. The key idea is to reduce the distance between images generated with and without guidance, $d(x_0, x_0^{\text{Guide}})$, rather than directly reducing the distance between their corresponding noises, $d(x_T, x_T^{\text{Guide}})$. Here $d$ denotes a distance metric, instantiated as the L2 distance. We formally state this relationship in Proposition 1 and provide a proof in Appendix B.2.

**Proposition 1.** *Let $x_T$ be an initial noise, and suppose that $x_0$ is the image obtained through denoising. Assuming Lipschitz continuity of the denoising mapping $\epsilon_\theta^{(t)}$ with respect to the distance metric $d$, for every $x_T$ there exists a constant $\kappa > 0$ such that the following holds:*

$$d(x_T, x_T^{Guide\dagger}) < \kappa d(x_0, x_0^{Guide}).$$

This result suggests optimizing in the image space, since reducing the image discrepancy also bounds the discrepancy in the corresponding noise. In the following sections, we detail how to train the refining network, our architectural choice for the refining network, and how to mitigate the costly backpropagation through full denoising steps.

### 3.3 TRAINING FRAMEWORK

Fig. 2 (b) illustrates the training framework. Starting from Gaussian noise $x_T$ and a prompt $c$, a diffusion model generates a guided image $x_0^{\text{Guide}}$ using $N'$ denoising steps with guidance. Any diffusion guidance (Ho & Salimans, 2022; Ahn et al., 2024; Hong et al., 2023; Sadat et al., 2024; Hong, 2024; Karras et al., 2024) or their combination can be applied for distillation.

Our model, noise refining network $g_\phi(\cdot)$, refines the initial noise $x_T$ into the ***refined*** noise $\hat{x}_T$, which is then fed into the same diffusion pipeline to generate an image $\hat{x}_0$ using $N$ denoising steps *without guidance*. The training objective minimizes the L2 distance $d(\hat{x}_0, x_0^{\text{Guide}})$ between the unguided output $\hat{x}_0$ and the guided target $x_0^{\text{Guide}}$. Through this process, noise refining network learns to transform initial noise into refined noise, capturing the benefits of guidance without modifying the diffusion model. At inference time, we sample a Gaussian noise, apply a single refinement step once before denoising, and then perform the standard unguided denoising process starting from the refined noise. Note that both noise refining network and the original model also receive a prompt $c$, though omitted for simplicity.

**Noise refining network.** Although the architecture of noise refining network $g_\phi(\cdot)$ can be chosen flexibly, we adopt a lightweight LoRA (Hu et al., 2021) on pretrained diffusion models. This allows noise refining network to effectively leverage the diffusion model's rich knowledge of text and image information while enabling parameter-efficient fine-tuning and faster convergence. Moreover, instead of loading a new refining network, LoRA can be attached for refinement and then detached, reducing GPU memory usage and seamlessly integrating with the original diffusion pipeline.

Table 1: **Quantitative comparison between noise space loss vs. image space loss (ours).**

| Method | PickScore | HPSv2 | AES | IR | CLIPScore |
|---|---|---|---|---|---|
| Noise space | 17.97 | 0.087 | 4.079 | -2.269 | 18.39 |
| **Ours** | **21.62** | **0.258** | **5.296** | **0.190** | **36.43** |

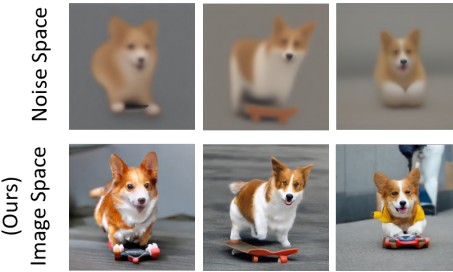

To better capture the structured differences between Gaussian and refined noise observed in Fig. 3, we introduce a residual connection in $g_\phi(\cdot)$, as shown in Fig. 2 (b), allowing the network to predict only the correction rather than the full refined noise.

Figure 5: **Sample images from models using noise space loss and image space loss.**

**Mitigating the cost of backpropagation on multiple denoising steps.** Widely used foundational diffusion models, such as the Stable Diffusion family, typically require 20–30 denoising steps to produce high-quality results. Although it is possible to naively apply our method, doing so would incur high computational costs due to backpropagation through the denoising network up to $N$ times, along with substantial GPU memory usage, making training inefficient. These constraints have been a major bottleneck, limiting many prior noise optimization methods (Eyring et al., 2024; Kim et al., 2024) to one- or few-step diffusion models (Lin et al., 2024; Sauer et al., 2024b).

To circumvent the backpropagation costs of the full-step diffusion model, we propose a novel approach, "multistep score distillation (MSD)", where we detach gradients through the denoising network during backpropagation, inspired by score distillation sampling (Poole et al., 2022).

Specifically, the typical denoising process is:

$$D_1(\ldots D_T(g_\phi(x_T))), \tag{1}$$

where $D_t(x)$ represents a single denoising step:

$$D_t(x) = a_t x_t + b_t \epsilon_\theta^{(t)}(x), \tag{2}$$

where $a_t$ and $b_t$ are coefficients derived from the DDIM sampler (Song et al., 2020a) and are formally defined in Appendix B.1. Then, the loss $\mathcal{L}_{\text{Denoise}}$, defined as the L2 loss between the denoised image and the target image $x_0^{\text{Guide}}$, is given by

$$\mathcal{L}_{\text{Denoise}}(g_\phi(x_T), \theta) := d\left(D_1\left(\ldots D_T(g_\phi(x_T))\right), x_0^{\text{Guide}}\right), \tag{3}$$

where $d$ represents the L2 distance.

In MSD, we perform the typical denoising process but detach the gradients on the denoising network $\epsilon_\theta$ at each step. Specifically:

$$\mathcal{L}_{\text{MSD}}(g_\phi(x_T), \theta) := d\left(F_1\left(\ldots F_T(g_\phi(x_T))\right), x_0^{\text{Guide}}\right), \tag{4}$$

where

$$F_t(x) = a_t x_t + b_t \text{ SG}(\epsilon_\theta^{(t)}(x)). \tag{5}$$

$\text{SG}(\cdot)$ denotes the stop-gradient (detach) operation.

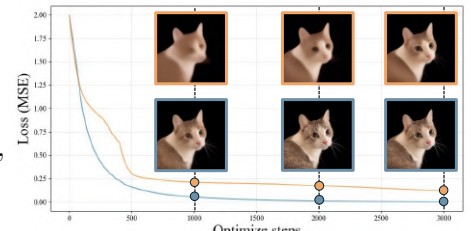

Figure 6: **Comparison of optimization results.** Orange: full-gradient MSE optimization; Blue: MSD loss optimization.

We conduct a pilot experiment to examine the effect of detaching gradients in the denoising process. Specifically, we optimize an initial Gaussian noise $x_T$ using the MSD loss $\mathcal{L}_{\text{MSD}}$ and the full-step gradient loss $\mathcal{L}_{\text{Denoise}}$ to make the denoised image close to the given target $x_0^{\text{Guide}}$, and compare the results in Fig. 6. As shown, detaching gradients leads to faster convergence and sharper images while significantly reducing computational costs. This improvement arises because omitting the denoiser Jacobian avoids unstable multi-step backpropagation and helps mitigate gradient explosion or vanishing that occurs when repeatedly backpropagating through the same denoiser, similar to long-horizon instability in recurrent networks. We further illustrate this stabilization effect in Fig. 35–36 in Appendix B.3.

In our training framework, the noise refining network $g_\phi(\cdot)$ is trained to minimize $\mathcal{L}_{\text{MSD}}(g_\phi(x_T), \theta)$ with respect to the refining network parameters $\phi$. We validate our approach, demonstrating that MSD closely approximates learning with the full-gradient loss $\mathcal{L}_{\text{Denoise}}(g_\phi(x_T), \theta)$. This is formalized in the following proposition, with a detailed proof and empirical evidence in Appendix B.2–B.3.

**Proposition 2.** *By approximating the gradients through multistep score distillation (MSD) using detached gradients at each step, we approximate the full-gradient objective with a mild assumption. In conclusion, the two gradients can be approximated as follows:*

$$\nabla_\phi \mathcal{L}_{Denoise}(g_\phi(x_T), \theta) \approx k \nabla_\phi \mathcal{L}_{MSD}(g_\phi(x_T), \theta), \tag{6}$$

*where $k \in (0, 1)$ is constant.*

## 4 EXPERIMENTS

In this section, to show the effectiveness and efficiency of the noise refining network, we present extensive qualitative and quantitative results. Following this, we demonstrate the advantages of our method which stem from preserving the diffusion pipeline intact, such as its generalizability to other fine-tuned diffusion models and compatibility with time-step distillation methods.

| Random noise w/o guidance | Random noise w/ guidance | **(ours)** *Refined* noise w/o guidance | Random noise w/o guidance | Random noise w/ guidance | **(ours)** *Refined* noise w/o guidance |

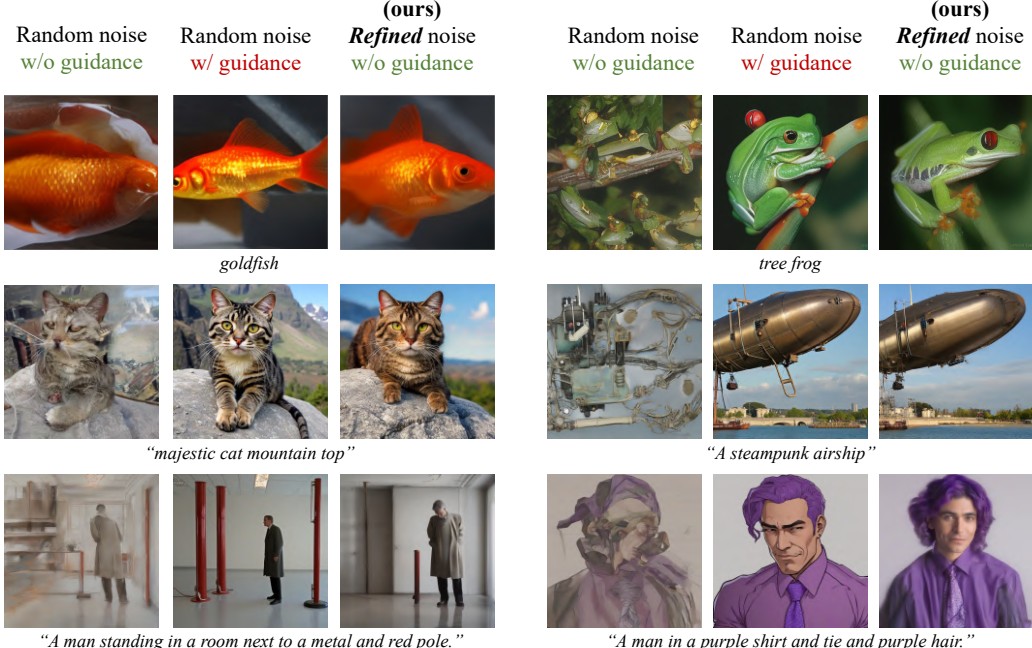

*goldfish*      *tree frog*

*"majestic cat mountain top"*      *"A steampunk airship"*

*"A man standing in a room next to a metal and red pole."*      *"A man in a purple shirt and tie and purple hair."*

Figure 7: **Qualitative results.** Samples generated (left) from Gaussian noise without guidance, (middle) from Gaussian noise with sampling guidance (Ho & Salimans, 2022; Ahn et al., 2024), and (right) from refined noise without guidance. The first row present results from SiT-XL/2, the next row from SD2.1, and the final row from SDXL.

## 4.1 SETUP

**Training setup.** To evaluate the effectiveness of *NoiseRefine*, we train the noise refining network on three models with distinct conditions, objectives, and architectures: a class-conditional flow-matching model (SiT-XL/2 (Ma et al., 2024)) and two text-to-image (T2I) diffusion models (Stable Diffusion (SD) 2.1 (Rombach et al., 2022) and SDXL (Podell et al., 2023)). For T2I models, prompts are sampled from MS COCO (Lin et al., 2014) and Pick-a-Pic (Kirstain et al., 2023) for training. Notably, our method does not require paired image datasets. The refining network is trained with classifier-free guidance (CFG) (Ho & Salimans, 2022) on the class-conditional model, and with both CFG and perturbed-attention guidance (PAG) (Ahn et al., 2024) on the T2I models. Further implementation details are provided in Appendix D.

**Evaluation setup.** For T2I models, we generate 30K images for evaluation from 30K unique prompts randomly sampled from the MS COCO 2014 validation set, disjoint from the training split. For SiT-XL/2, we use 50K samples with ImageNet conditions. All qualitative examples are drawn from these sets. For the main experiments with SiT-XL/2, we use the Euler sampler with 20 denoising steps. For SDXL and SD 2.1, we use the DDIM sampler with 20 denoising steps.

## 4.2 QUALITATIVE AND QUANTITATIVE EVALUATION

**Qualitative comparison.** Fig. 7 presents representative samples. Without guidance, Gaussian noise yields spatially incoherent images (1st, 4th columns), while refined noise produces consistently higher-quality results with plausible structure (3rd, 6th columns). This underscores the critical role of the initial noise and demonstrates that our refining network distills guidance signals into spatially informed noise, enabling consistent generations. Additional results on SiT-XL/2, SD 2.1, and SDXL are provided in Appendix E (Figs. 40–45). For completeness, we also provide results on SiT using the original evaluation protocol (250-step 2nd-order Heun sampler) in Appendix A.13.

**Quantitative comparison.** To evaluate image fidelity and diversity, we compute Fréchet Inception Distance (FID) (Heusel et al., 2017) and Inception Score (IS) (Salimans et al., 2016) as shown in Tab. 2. For each model, we compare four settings: (1) unguided sampling from Gaussian noise (our baseline), (2) guided sampling from Gaussian noise, (3) guidance-distilled sampling (Meng et al., 2023) from Gaussian noise, and (4) unguided sampling from refined noise with noise refining network. Refined noise consistently improves FID and IS over Gaussian noise, and achieves

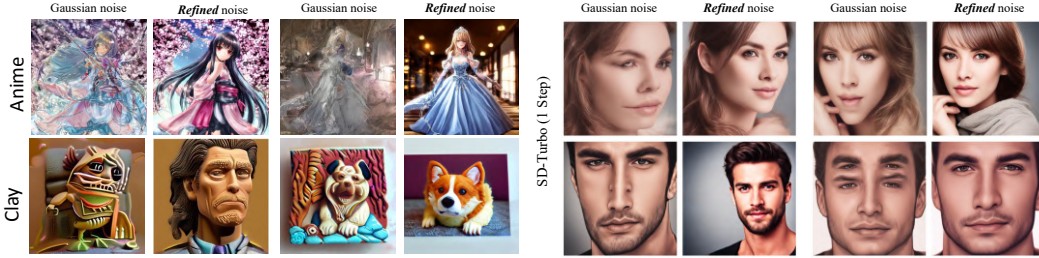

(a) Generalizability on fine-tuned model          (b) Generalizability on timestep-distilled model

Figure 8: **Generalizability and compatibility of refined noise.** (a) Results on fine-tuned models (animation and clay object domains) comparing Gaussian vs. refined noise. (b) Results on timestep-distilled models (SD-Turbo), showing that refined noise improves structural coherence and quality over Gaussian noise.

quality close to guided or guidance-distilled sampling with only a single refinement step. Details on implementation of each method are provided in Appendix D.

**Training cost.** Since our proposed method involves guided sampling during training, it introduces additional computational overhead compared to guidance distillation Meng et al. (2023). For a fair comparison, we provide a detailed training cost analysis in Appendix A.7, and report the performance of both methods over training steps in Fig. 26 (Appendix A.8).

**User study.** We conducted a user study to evaluate prompt adherence and image quality by comparing guided samples from Gaussian noise with unguided samples from refined noise. As shown in Tab. 3, participants preferred refined-noise samples and guided samples at similar rates. Additional details and comparisons with unguided Gaussian noise are provided in Appendix E.2.

**Ablation studies.** We provide additional ablations in Appendix C, including network architecture, number of denoising steps $N$, and other factors. We also report the results of SiT-XL/2 Ma et al. (2024) using the Heun sampler with 125 denoising steps, following the original SiT paper, in Appendix A.13.

Table 2: **Quantitative comparison of image quality.** 30K prompts from MS-COCO (Lin et al., 2014) validation dataset were used for evaluation. *Guidance Distil.* indicates guidance distillation (Meng et al., 2023).

| Model | Initial Noise | Sampling Guidance | FID ↓ | IS ↑ |
|---|---|---|---|---|
| SiT-XL/2 | Gaussian | ✗ | 18.43 | 40.00 |
| | Gaussian | ✓ | 14.20 | 63.99 |
| | Gaussian | ✗ (*Guidance Distil.*) | 12.12 | 58.90 |
| | *Refined* (Ours) | ✗ | 10.80 | 50.59 |
| SD2.1 | Gaussian | ✗ | 42.71 | 20.86 |
| | Gaussian | ✓ | 16.19 | 37.95 |
| | Gaussian | ✗ (*Guidance Distil.*) | 19.09 | 33.45 |
| | *Refined* (Ours) | ✗ | 14.62 | 34.90 |
| SDXL | Gaussian | ✗ | 63.28 | 17.64 |
| | Gaussian | ✓ | 21.20 | 34.60 |
| | Gaussian | ✗ (*Guidance Distil.*) | 18.57 | 37.51 |
| | *Refined* (Ours) | ✗ | 26.22 | 27.63 |

Table 3: **User study on image quality and prompt adherence.**

| Metric | Gaussian Noise w/ Guidance | Refined Noise (Ours) w/o Guidance |
|---|---|---|
| Image Quality | 46.04% | **53.96%** |
| Prompt Adherence | 48.24% | **51.76%** |

## 4.3 ADVANTAGES OF NOISE REFINEMENT

In this subsection, we highlight the advantages of noise refining for guidance-free generation. This approach preserves the diffusion pipeline, including the denoising network, maintaining the model's integrity. Our method can be viewed as a form of prompt learning (Zhou et al., 2022a), which prevents catastrophic forgetting (Kirkpatrick et al., 2017). Further discussion is available in A.1.

**Generalizability on different domains.** Guidance distillation (Meng et al., 2023) can remove the need for guidance in a base model. Yet, applying it to fine-tuned models necessitates a separate distillation step for each variant, making the process computationally expensive. In contrast, our noise refining network, trained on the base model, can be directly applied to fine-tuned models, enabling efficient adaptation across multiple domains. We present this by transferring our noise refining network, trained on Stable Diffusion 2.1, to a fine-tuned model in the animation and clay object domain. Fig. 8 (a) shows that our model effectively refines noise, eliminating the need for guidance across different domains. We also provide quantitative results for this zero-shot transfer of the noise refining network in Tab. 4, showing performance comparable to guided generation. Additional results are in Appendix E.3.

Table 4: **Quantitative results on generalization to finetuned models across different domains.**

| Domain | Initial Noise | Guidance | PickScore | HPSv2 | ImageReward | Aesthetic | CLIPScore |
|--------|--------------|----------|-----------|-------|-------------|-----------|-----------|
| Clay | Gaussian | ✗ | 17.95 | 0.18 | -1.69 | 5.32 | 18.31 |
|  | Gaussian | ✓ | 19.17 | 0.24 | -0.32 | 5.53 | 26.36 |
|  | *Refined* (Ours) | ✗ | 18.82 | 0.21 | -0.95 | 5.39 | 23.61 |
| Anime | Gaussian | ✗ | 16.47 | 0.17 | -1.59 | 5.21 | 22.37 |
|  | Gaussian | ✓ | 17.68 | 0.24 | 0.04 | 5.56 | 30.04 |
|  | *Refined* (Ours) | ✗ | 18.08 | 0.24 | -0.34 | 5.48 | 29.62 |

**Compatibility with timestep distillation models.** Our method integrates seamlessly with existing timestep distillation approaches (Luo et al., 2023a; Sauer et al., 2024a; Xu et al., 2024b; Luo et al., 2023b; Yin et al., 2024; Lin et al., 2024; Salimans & Ho, 2022) without requiring additional training, since it preserves the diffusion pipeline unchanged. We apply refined noise to SD-Turbo (Luo et al., 2023a) and evaluate its performance. Qualitative results are shown in Fig. 8 (b), and quantitative comparisons are reported in Table 5.

Table 5: **Quantitative results of refined noise on timestep-distilled model (SD-Turbo).**

| Noise | Denoising Step | PickScore | HPSv2 | AES | ImageReward | CLIPScore | FID ↓ | IS ↑ |
|-------|---------------|-----------|-------|-----|-------------|-----------|-------|------|
| Gaussian | 1 Step | 21.38 | 0.270 | 5.47 | 0.04 | 30.97 | 27.18 | 34.31 |
| Gaussian | 2 Step | 21.75 | 0.295 | **5.62** | 0.11 | 30.87 | 30.24 | 32.46 |
| *Refined* | 1 Step | **21.92** | **0.300** | 5.51 | **0.43** | **31.19** | **24.94** | **38.07** |

Compared to generation starting from Gaussian noise, our approach improves structural coherence and overall quality, highlighting the role of structured initial noise even in few-step models. Moreover, single-step inference with refined noise generally outperforms two-step inference from Gaussian noise in terms of numerical metrics.

## 5 DISCUSSION

In this section, we analyze what noise refining network learns and identify components in refined noise that support better generation quality.

**Low-frequency components aid denoising.** Analysis of the refining network's output shows that it primarily adds low-magnitude, low-frequency signals. In Fig. 9 (a), the difference between Gaussian and refined noise is concentrated in small values, unlike the difference between two Gaussian samples. Moreover, (b) indicates that noise refining network naturally produces low-frequency layouts without explicit constraints. This observation is consistent with Fig. 3, where the gap between inversion noise and Gaussian noise also lies mainly in the low-frequency range.

As illustrated in Fig. 10, these components are condition-dependent and serve as an initial *layout*, shaping object structures early in denoising and improving coherence. To further examine their role, we performed frequency decomposition (separating low- and high-frequency contributions; Appendix A.2) and cross-prompt experiments (testing robustness under mismatched prompts; Appendix A.3), which highlight the critical importance of low-frequency signals.

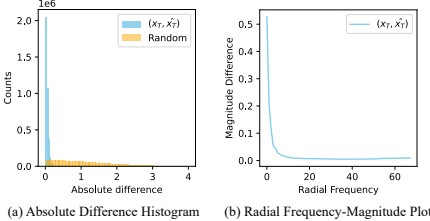

(a) Absolute Difference Histogram  (b) Radial Frequency-Magnitude Plot

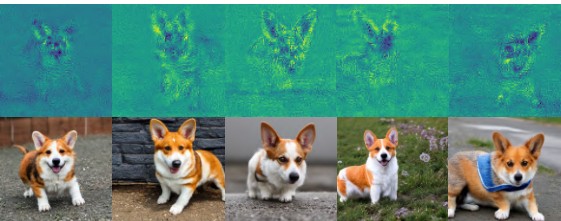

Figure 9: **Analysis of initial noise $x_T$ vs. refined noise $\hat{x}_T$.** (a) Histogram of absolute differences (vs. pairs of random Gaussian noise). (b) Fourier magnitude differences, showing variation mainly in low frequencies.

Figure 10: **Analysis of initial noise $x_T$ vs. refined noise $\hat{x}_T$.** The top row visualizes the absolute noise difference, and the bottom row shows the corresponding generated images. The added signal acts as a coarse structural layout for generation.

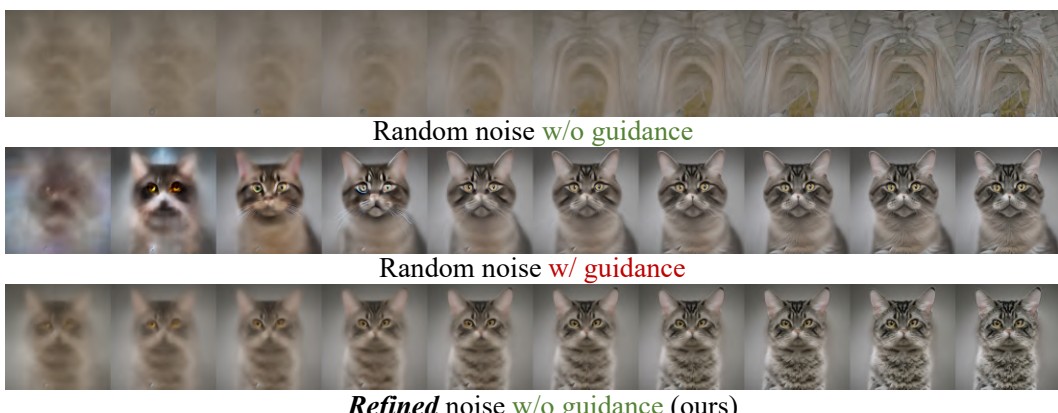

Figure 11: **Refined noise enables coherent trajectories.** From left to right, $x_0$ predictions are shown as $t$ goes from $T$ to 0. **Refined** noise yields a consistent trajectory by providing initial layout.

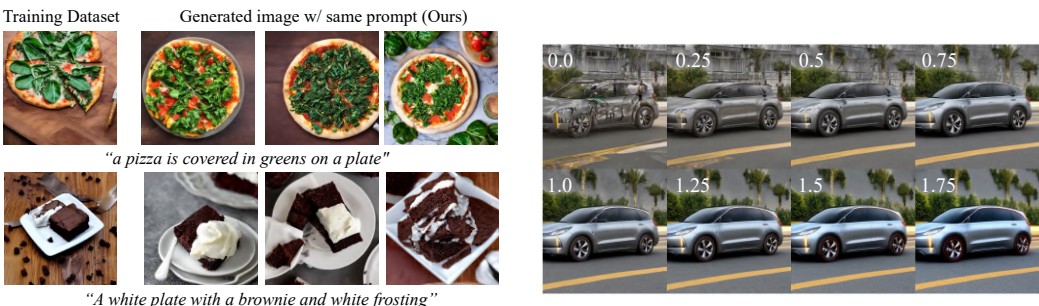

Figure 12: **Nearest generated images from training images.** From 10 samples per training image, the top-3 nearest neighbors are shown, indicating novelty beyond the training set.

Figure 13: **Controlling the strength of noise refinement.** Numbers in the top-left corner indicate the scaling factor of $g_\phi$.

**Consistent trajectory.** The third row of Fig. 11 shows that starting from refined noise, the model quickly forms plausible layouts in early steps, enabling it to focus on adding details during denoising. In contrast, the first row shows that Gaussian noise fails to establish a coherent structure early, leading to misplaced details and leaving ambiguous regions untouched throughout denoising. We also analyze the corresponding cross-attention maps in Fig. 20 of Appendix.

**Diversity and novelty.** Although refined noise provides an initial layout, results remain diverse across seeds, with IS (Salimans et al., 2016) scores surpassing those from Gaussian noise (Tab. 2). Nearest-neighbor retrieval (Fig. 12) confirms that the outputs are not simple replicas of training data but genuinely *novel* samples.

**Controllability.** The strength of guidance can be adjusted in two ways. In the training-free case, scaling the output of $g_\phi$, the residual between Gaussian and refined noise, controls the coherence of image structure (Fig. 13), analogous to tuning the guidance scale. In the training-based case, the model can be conditioned on an additional guidance-scale embedding, with results in Appendix A.4.

**Comparison to related works.** Our goal is to distill diffusion guidance into the noise space, which differs from prior work on noise reinitialization, search, or optimization. For completeness, we discuss related approaches and their objectives, and provide comparison experiments in Appendix A.9.

## 6 CONCLUSION

In this work, we propose **NoiseRefine**, a method that replaces costly guidance in diffusion sampling with a single noise refinement step. Our approach preserves the original diffusion pipeline, prevents catastrophic forgetting, and enables seamless integration with existing timestep distillation techniques (Meng et al., 2023; Sauer et al., 2024b) to enhance image quality and coherence. Additionally, we analyze the properties of refined noise and its role in denoising, providing insights into the influence of noise in diffusion models. We believe our work paves the way for leveraging expressive noise space in a training-based manner.

## 7 REPRODUCIBILITY STATEMENT

We provide detailed explanations and proofs of the theoretical results in Appendix B, and further describe the architecture, implementation, and experimental details in Appendix D. We will also release our code and model checkpoints to ensure reproducibility.

### ACKNOWLEDGMENTS

This research was supported by Institute of Information & communications Technology Planning & Evaluation (IITP) grant funded by the Korea government (MSIT) (RS-2019-II190075, RS-2024-00509279, RS-2025-II212068, RS-2023-00227592, RS-2025-02214479, RS-2024-00457882, RS-2025-25441838, RS-2025-25441838, RS-2025-02214479, RS-2025-02217259) and the Culture, Sports, and Tourism R&D Program through the Korea Creative Content Agency grant funded by the Ministry of Culture, Sports and Tourism (RS-2024-00345025, RS-2024-00333068, RS-2023-00222280, RS-2023-00266509), and National Research Foundation of Korea (RS-2024-00346597). We also gratefully acknowledge the provision of GPU compute resources by Hugging Face for this research.

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

## APPENDIX

In the Appendix, we provide discussions including the in-depth analysis on refined noise and its impact on denoising process (Section A), clarify the notations and formulations related to diffusion models used in the main paper and provide the proofs for our propositions (Section B), more ablation studies regarding noise refining network (Section C), implementation details and experimental settings (Section D), additional results including qualitative results, comparison with other methods, user study (Section E).

## CONTENTS

## A    ADDITIONAL DISCUSSIONS

In this section, we discuss the advantages of training noise refining network $g_\phi$ for guidance-free generation (Sec. A.1). In addition, we present our hypothesis on why refined noise eliminates the need for guidance methods, explaining it step by step (Sec. A.2). We further analyze the impact of initial noise and prompt on the generated image (Sec. A.3).

### A.1    EFFECTIVENESS OF PROMPT LEARNING

Why is learning noise mapping beneficial? A useful perspective comes from the success of prompt learning in large-scale models. Models such as CLIP (Radford et al., 2021), trained on web-scale datasets with billions of parameters, are difficult to fine-tune due to their sheer size and the risk of *disturbing well-learned representations* (Zhou et al., 2022b). Instead, prompt learning, which optimizes input prompts rather than model parameters, has emerged as an effective alternative (Zhou et al., 2022b;a; Jiang et al., 2020; Shin et al., 2020). In particular, conditional prompt learning methods like CoCoOp (Zhou et al., 2022a) generate prompts based on different inputs. Similarly, in our approach, noise prompts are learned based on Gaussian noise $x_T$ and the text prompt $c$, allowing for more efficient guidance.

In this context, restricting training to the noise space rather than modifying the entire denoising pipeline offers several advantages. As illustrated in Fig. 9 and Fig. 10, key low-frequency components in the noise space encode structural information such as image layout. This enables efficient learning with a relatively small dataset, without requiring modifications to the entire model. By contrast, full fine-tuning often leads to excessive computational costs and the risk of overfitting.

More importantly, unlike guidance distillation methods such as (Meng et al., 2023), our approach preserves the original model and prevents catastrophic forgetting (Kirkpatrick et al., 2017). This ensures that pretrained modules, such as DreamBooth (Ruiz et al., 2023) or LoRA (Hu et al., 2021), remain fully compatible. Fig. 14 illustrates this effect: when applying the Miranda Kerr LoRA, guidance distillation (Meng et al., 2023) alters identity characteristics, whereas our method preserves the original sample's identity while improving image quality. This demonstrates that our method maintains the integrity of the representation space, while guidance distillation compromises it.

Gaussian noise    ***Refined*** noise    Guidance Distill

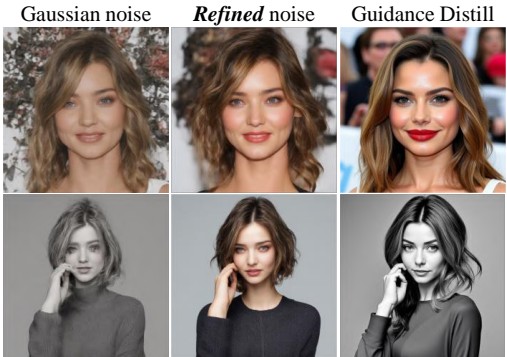

Figure 14: **Compatibility of LoRA for each method.** Results of applying the 'Miranda Kerr' LoRA, trained on SD 2.1. Distilled model exhibits different identity with 'Miranda Kerr'.

### A.2    WHY DOES REFINED NOISE HELP DENOISING?

To identify which refined noise components contribute to guidance-free generation, we first decompose the refined noise into multiple frequency components. In this study, we utilize a two-dimensional Fourier transform to break down both the refined noise and the initial noise into their respective frequency components. Each frequency component is represented by a frequency band, denoted as $(a, b)$, which corresponds to the frequency range from $a$ to $b$. Note that although we explored other decomposition methods, such as dividing the noise into patches, they did not yield interpretable results.

(a) denoised images according to the cutoff band

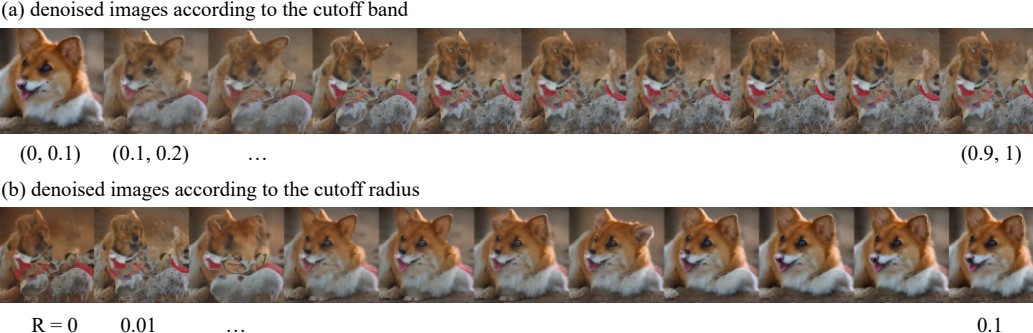

(0, 0.1)    (0.1, 0.2)    …                                                                  (0.9, 1)

(b) denoised images according to the cutoff radius

R = 0    0.01    …                                                                  0.1

Figure 15: **Visualization of denoised images according to the cutoff band.** Both refined and initial noise were transformed into the frequency domain using Fourier transforms. The frequency domain of the initial noise, normalized such that the maximum radius is 1. (a) The frequency divided into intervals of 0.1. For each interval, the corresponding frequency components were replaced with those from the refined noise, followed by denoising. The results show that only when the (0, 0.1) frequency band was replaced does an image generated by the refined noise emerge. (b) Visualization of denoised images by incrementally increasing the cutoff radius from 0 in steps of 0.01 and replacing the corresponding components of the initial noise with refined noise. The results demonstrate that images denoised using refined noise are obtained starting at a cutoff radius of 0.03.

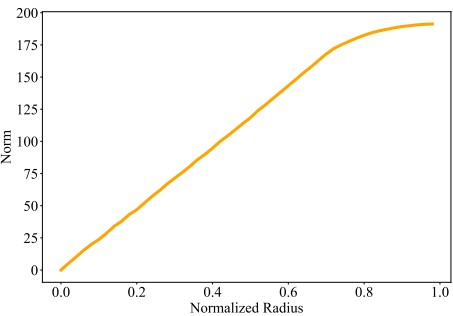

Figure 16: **Visualization of the norm based on the frequency-filtered radius of refined noise.** This visualization demonstrates the increase in norm as the cutoff radius in the frequency domain is expanded. The refined noise was transformed into the frequency domain using a Fourier transform, and the norm corresponding to each cutoff radius was calculated and plotted.

**Low-frequency components matter.** Using 2D Fourier transforms, we transform both refined and initial noise into the frequency domain. The initial and refined noise frequency domain is normalized into $(0, 1)$. We synthesize a new noise signal by replacing specific frequency bands of the initial noise with the corresponding bands from the refined noise. Fig. 15 (a) presents the generated images corresponding to different frequency bands, demonstrating that the low-frequency components of the refined noise predominantly influence the generation process. In Fig. 15 (b), images are generated by varying the band length within the low-frequency region. The results indicate that, despite the low magnitude of the low-frequency components, which can be confirmed through Fig. 16, they are sufficient to reconstruct the image effectively.

**Diffusion models can generate images using only low-frequency components.** In Fig. 17, we examine how well diffusion models can denoise when specific frequency bands of refined noise are retained, and the values of the remaining bands are set to zero (using ideal high/low pass filters). The top row shows the results of applying a 2D Fourier transform to the refined noise, normalizing the FFT frequency domain into $(0, 1)$, and sequentially retaining lower frequency bands, such as $(0, 0), (0, 0.1), (0, 0.2), ..., (0, 1)$, while setting the remaining bands to zero. These noise inputs are then denoised without CFG (Ho & Salimans, 2022). The figure demonstrates that the diffusion model begins forming a recognizable corgi shape even when only the lower 50% of frequency bands

Frequency band to keep, otherwise zero

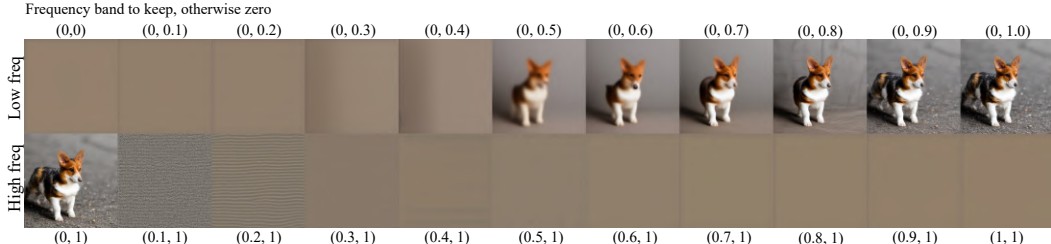

Figure 17: **Denoised images using only low(top) / high(bottom) frequency components.** Diffusion models can generate the overall structure of the image using only the low-frequency bands of the refined noise. We use DDIM (Song et al., 2020a) with 20 steps for denoising without CFG, and the prompt was *"a photo of a corgi"*.

of the refined noise are present. In contrast, noise containing only high-frequency bands fails to generate coherent images.

Frequency band to keep, otherwise reinitialize

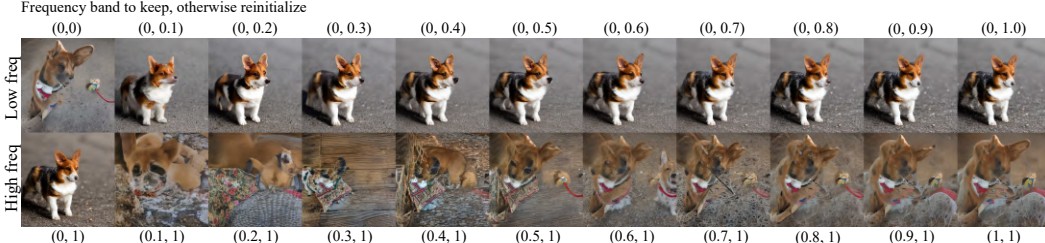

Figure 18: **Denoised images using only low (top) / high (bottom) frequency components with reinitialization.** We use DDIM (Song et al., 2020a) with 20 steps for denoising without CFG, and the prompt was *"a photo of a corgi"*.

**High-frequency components contribute details.** Here, we use the same noise decomposition process of refined noise as Fig. 17 but following (Geng et al., 2025), we reinitialize the frequency components that were set to zero with corresponding components from standard Gaussian noise, then denoise again. The results, shown in Fig. 18, indicate that when all frequency components are present, the diffusion model can generate clear and complete images. Randomly reinitialized high-frequency components appear to add details onto the structure formed by the low-frequency components. While refined noise retaining only the lower 10%–20% of frequencies can still reconstruct the original image when the rest is reinitialized, noise retaining only the high-frequency components fails to do so. This suggests that low-frequency components alone carry the significant information needed for image generation.

In Fig. 19, each row visualizes images generated with only the lower 5%, 10%, 20%, and 30% (from the top rows to last rows) frequency components of the refined noise, while the bottom row shows images generated with only the upper 5%, 10%, 20%, and 30% frequency components. These results confirm that low-frequency components encode the overall layout and structure, whereas high-frequency components lack meaningful information.

From these observations, we infer that the poor quality of unguided diffusion model outputs is due to their failure to form appropriate low-frequency components during denoising. High-frequency details added on poorly formed layouts result in artifacts that are perceived as unnatural.

**How do guidance methods form plausible initial layouts?** As highlighted in (Ahn et al., 2024), classifier-free guidance (CFG) (Ho & Salimans, 2022) enhances the difference between conditional and unconditional predictions at each step, amplifying "signals that can only be generated with the condition" (e.g., features like the eyes or nose of a corgi in *"a photo of a corgi"*). This effectively strengthens salient features corresponding to low-frequency components in the early denoising steps. From this, we deduce that guidance methods (Ahn et al., 2024; Ho & Salimans, 2022; Hong et al.,

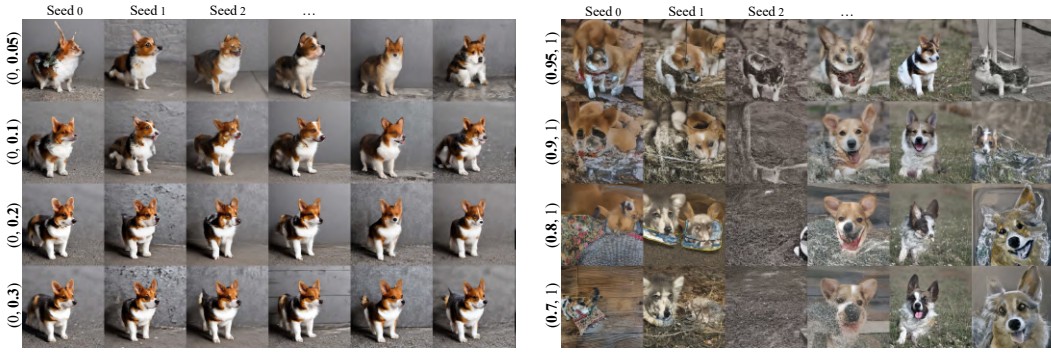

(a) Low-frequency components of refined noise          (b) High-frequency components of refined noise

Figure 19: **Different denoised images using only low(a) / high(b) frequency components for different seeds.** Here we use 8 different seeds. From the top rows, it visualizes 8 images using only the lower (a) / higher (b) 5%, 10%, 20%, and 30% (from the top to the last rows) frequency components of the refined noise.

2023) add appropriate low-frequency components during inference, aiding the formation of high-quality layouts.

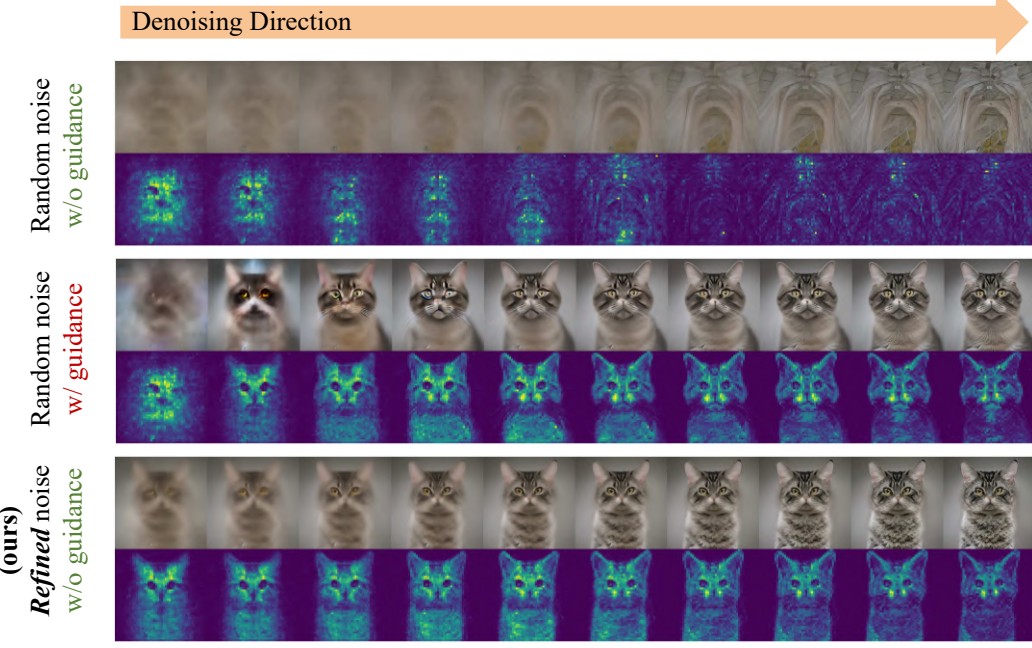

Figure 20: **Visualization of 11th layer cross attention map.** Token corresponding to 'cat' is used for visualization among the prompt 'a photo of a cat'. For each case, first row shows $x_0$ prediction at each timestep and second row shows cross attention map at the timestep. When guidance is not used, failure to create meaningful attention map across all timestep is notable, leading to completely broken generation. However when guidance or our refined noise is used, meaningful cross attention map is observed, leading to successful generation.

**How does noise refining network form low-frequency layouts?** Interestingly, noise refining network naturally forms low-frequency layouts even though our training framework does not explicitly enforce learning them as can be seen in Fig. 9. To understand this, we analyze cross-attention maps across denoising steps. Fig. 20 visualizes these maps at different timesteps. Gaussian noise fails to form meaningful cross-attention maps in early steps due to its near-zero signal-to-noise ratio (SNR), which is expected. However, this failure persists in later steps, indicating an inability to form well-aligned layouts (Fig. 20 first row).

Several studies (Chefer et al., 2023; Guo et al., 2024; Mao et al., 2023b) has shown that reducing noisy artifacts in cross-attention maps and aligning them with object regions during inference improves performance. This suggests that the failure of cross-attention maps to align is a key reason for the diffusion model's inability to create coherent layouts. When using CFG (Ho & Salimans, 2022) (second row) or refined noise (third row), the cross-attention maps align well with the prompt, resulting in better outputs. Notably, cross-attention maps for refined noise exhibit accurate object shapes from the very first step, implying that the diffusion model can form plausible layouts from the beginning of the denoising process. This is further supported by $x_0$ predictions of Fig. 20 at each denoising step.

**Implications for guidance-free generation.** Without guidance methods or noise refiners aiding the formation of low-frequency layouts, diffusion models fail to create plausible initial layouts. Random low-frequency components lead to artifacts that are perceived as unnatural. An interesting avenue for future research would be identifying why diffusion models struggle to form low-frequency components without guidance and developing training techniques to eliminate the need for guidance during the training stage.

### A.3 IMPACT OF AN INITIAL NOISE AND PROMPT ON THE GENERATED IMAGE

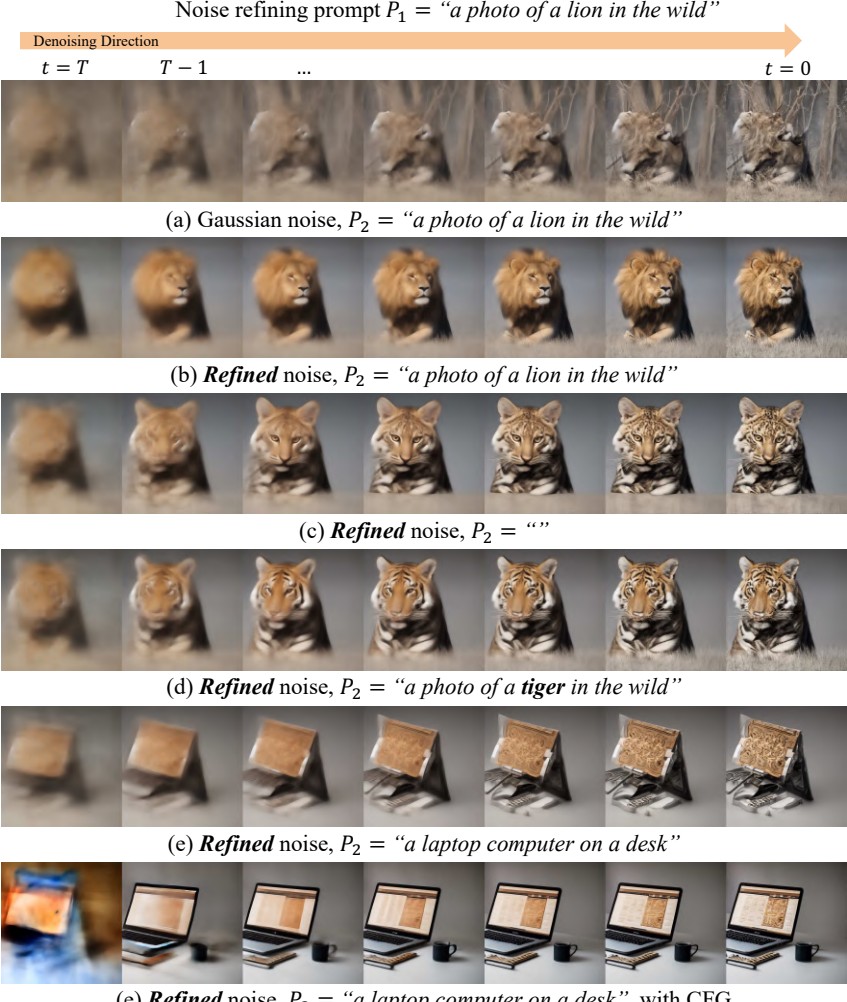

Figure 21: **Visualization of denoised image using different prompt for noise refinement $\epsilon_\theta$ and denoising $g_\phi$.**

We previously demonstrated how refined noise affects initial layouts and how guidance and refined noise contribute to forming these layouts effectively. In this section, we investigate how the 'layout' and the prompt influence the final generated image during the denoising process. Specifically, we explore what happens when the prompt used to generate the initial layout ($P_1$, one of the inputs to noise refining network $g_\phi$) differs from the prompt used during denoising ($P_2$, one of the inputs to the denoising network $\epsilon_\theta$ in the Guidance-Free T2I Pipeline shown in Fig. 38). Does the model prioritize one prompt over the other? Or does it attempt to harmonize both? We investigate this question through the results shown in Fig. 21.

- Fig. 21 (a) visualizes the predicted $x_0$ term in Eq. 15 during the denoising process when no layout is provided (starting from Gaussian noise). The leftmost image corresponds to the predicted $x_0$ at $t = T$, and subsequent images are visualized every three steps. Due to the noisy and ambiguous nature of the initial layout of Gaussian noise, the diffusion model fails to form a coherent lion layout from the initial structure. Instead, it partially adds features such as fur, mane, nose, or mouth, resulting in poor perceptual quality.

- In contrast, (b) shows that in the case of $P_1 = P_2$, refined noise effectively forms the lion layout from the beginning. The diffusion model accurately places the overall lion shape, including its mane, eyes, nose, and mouth, in appropriate positions during the denoising process.

- (c) shows the results when the denoising prompt $P_2$ is set to an empty prompt (null prompt). Despite this, the model successfully generates a feline animal based solely on unconditional generation, as the layout sufficiently captures the overall structure of the object. This can be interpreted as the information embedded in the ***refined*** noise.

- (d) demonstrates the case where the denoising prompt $P_2$ is set to a prompt similar to the initial layout prompt (*"a photo of a tiger in the wild"*). When a similar prompt is used, the image retains the layout provided by the refined noise while also adhering to the prompt.

- In (e), $P_2$ is set to an entirely independent prompt (*"a laptop computer on a desk"*). Here, the model fails to generate a coherent image corresponding to the layout or the prompt. The diffusion model attempts to form a laptop on the existing lion or feline layout but fails to align with the laptop prompt, leading to failure.

- Finally, (f) shows that applying CFG (Ho & Salimans, 2022) in the settings of (e) allows the diffusion model to disregard the initial layout and generate a laptop. This partially explains why CFG consistently produces high-quality images. Randomly generated initial noise is unlikely to align with the prompt (as shown in (a)), and CFG helps the model ignore such initial noise and generate images consistent with the given prompt.

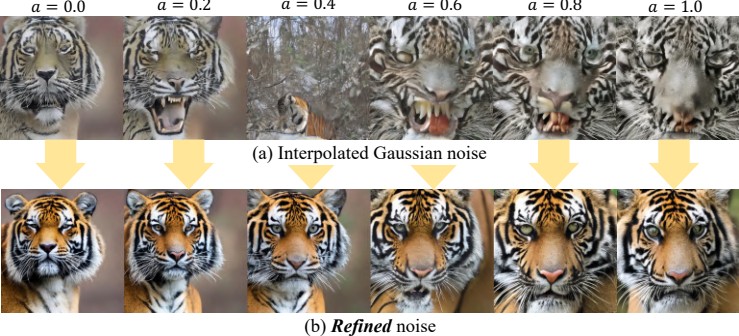

(a) Interpolated Gaussian noise

(b) ***Refined*** noise

Figure 22: **Images from interpolated refined Gaussian noise.**

**Interpolation between refined noise.** To evaluate whether noise refining network effectively learns noise mapping, we follow (Song et al., 2020a;b) to perform spherical interpolation on initial noise samples, generating multiple interpolated noises. We then refine each interpolated noise using noise refining network and verify that the refined noises effectively interpolate natural images. In Fig. 22, (a) shows the images denoised by the diffusion model without any guidance method,

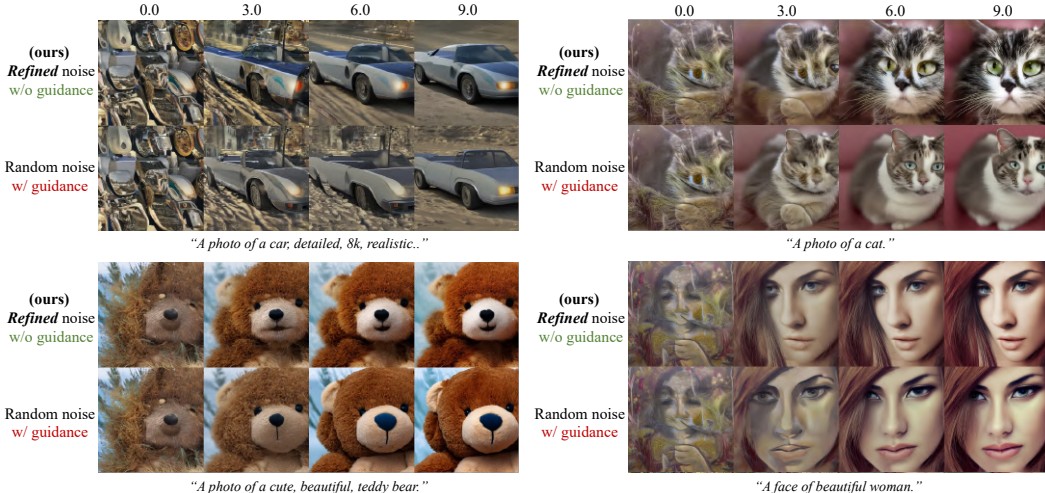

Figure 23: **Qualitative results of the training-based approach for controlling guidance strength.** Numbers above each column indicate the guidance scale provided as input to the model. Our results show that *NoiseRefine* effectively learns the controllability of classifier-free guidance, preserving the expected variations in image characteristics as the guidance scale changes.

starting from spherical interpolations of two random Gaussian noises. Specifically, each interpolated noise is obtained by performing slerp($x_{T_1}$, $x_{T_2}$, $a$) for various interpolation ratios $a$, where slerp performs spherical interpolation between two Gaussian noise at a ratio of $a$.

Fig. 22 (b) shows the results of denoising the refined versions of these interpolated noises without guidance. The results demonstrate that the refined noises effectively interpolate between the two images. This indicates that noise refining network does not simply memorize specific low-frequency signals while ignoring the input noise. Instead, it effectively learns a mapping from a Gaussian noise space to a guidance-free noise space where semantic interpolation between guidance-free images is possible.

## A.4 CONTROLLABILITY

The guidance strength can be controlled by scaling the output of the noise refining network, which provides a training-free control mechanism. We further show on SD2.1 that controllability can also be achieved through a training-based approach by incorporating the guidance scale as an additional input to the refiner, following (Meng et al., 2023; Luo et al., 2023a). The model architecture and dataset remain unchanged, except for adding a small linear projection layer for the guidance scale. Specifically, following the conditioning design in (Meng et al., 2023), we inject the guidance-scale embedding into the existing timestep embedding rather than introducing a new conditioning branch. The scale is encoded with a sinusoidal embedding and projected through a linear layer to match the timestep-embedding dimension, keeping the architectural change minimal and the training pipeline nearly identical. The model is trained for one epoch. Fig. 23 shows qualitative results, and Fig. 24 reports quantitative results.

## A.5 GUIDED SAMPLING WITH REFINED NOISE

Our noise refining network improves image quality not only in unguided sampling but also when guidance is applied. Unlike random Gaussian noise, refined noise (Fig. 10) preserves structural cues and provides a consistent "initial layout", reducing artifacts such as extra limbs and enhancing overall coherence (Fig. 25).

We also provide quantitative results in Tab. 6, using MS-COCO prompts under the same settings as the main quantitative results in Tab. 2, following the evaluation configurations detailed in Appendix D.2.2. This shows gains in prompt alignment and human preference.

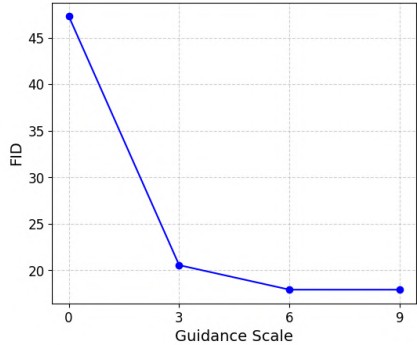

Figure 24: **Quantitative results of training-based approach to control guidance strength.** The metrics were computed using 5K prompts from the MS COCO 2014 validation set (Lin et al., 2014).

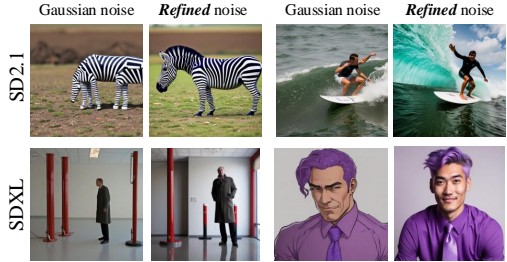

Figure 25: **Effectiveness of *refined* noise in *guided sampling*.**

### A.6 INFERENCE TIME ANALYSIS

We report comparison results of inference time for each method in Tab. 7. Inference time is computed by averaging time per image across 30K images generated with the inference step of 20 and a batch size of 1 on RTX 3090.

### A.7 TRAINING COST ANALYSIS

For training cost analysis based on NFE (Number of Function Evaluation), we assume unit cost for Refine ($R$), Backpropagation ($B$), Denoising ($D$), and VAE ($V$) operations. Following Appendix D, we denote $N'$ as the number of denoising steps used during inference (including dataset generation) and $N$ as number of denoising steps used during training. In our SD2.1 setting, we used $M = 50K$ samples, $E = 6$ epochs, $N' = 20$, and $N = 10$.

Based on these parameters, the total training cost of our method amounts to 5.65M steps, which is only 0.054% of the original training cost of SD2.1 (Tab. 8).

### A.8 COMPARISON OF TRAINING EFFICIENCY WITH GUIDANCE DISTILLATION

Although our primary aim is not to develop a more efficient guidance distillation procedure but rather to explore whether diffusion guidance can be distilled into noise instead of the network, we nevertheless provide a training-efficiency comparison with conventional guidance distillation for reference. To this end, we present the FID curves over steps for training noise refining network and a guidance-distilled model on SiT-XL/2 in Fig. 26.

Note that, aside from the dataset generation phase, training noise refining network imposes a higher per-step computational cost: our method applies a loss after denoising for $N$ steps, whereas standard guidance distillation computes its loss after a single denoising step. Thus, in theory, the GPU cost per step differs by approximately a factor of $N$. Because the two models were trained on different hardware environments, we present step-wise FID curves rather than wall-clock comparisons.

| Model | Noise | PickScore ↑ | HPSv2 ↑ | AES ↑ | IR ↑ | CLIPScore ↑ | FID ↓ | IS ↑ |
|---|---|---|---|---|---|---|---|---|
| SD 2.1 | Gaussian | 21.70 | 0.280 | 5.530 | 0.294 | 30.72 | **18.74** | 32.55 |
|  | Refined | **21.93** | **0.324** | **5.602** | **0.448** | **30.99** | 22.94 | **34.37** |
| SDXL | Gaussian | 22.02 | 0.280 | 5.706 | 0.717 | 30.77 | **21.02** | 34.60 |
|  | Refined | **22.48** | **0.289** | **5.720** | **0.977** | **31.36** | 22.34 | **35.23** |

Table 6: **Quality improvement of refined noise in *guided* sampling.**

| Model | Initial Noise | Guidance | Inference Time ↓ |
|---|---|---|---|
| SD2.1 | Gaussian | ✗ | **1.357s** |
|  | Refined | ✗ | 1.504s |
|  | Gaussian | ✓ | 2.589s |
| SDXL | Gaussian | ✗ | **3.218s** |
|  | Refined | ✗ | 3.323s |
|  | Gaussian | ✓ | 5.525s |

Table 7: **Quantitative comparison of image quality and computational cost.** 30K prompts from MS-COCO (Lin et al., 2014) validation dataset were used for evaluation.

As shown in the figure, training noise refining network converges somewhat more slowly, but it reaches the same FID to guidance distillation at around 2K steps

## A.9   COMPARISON WITH OTHER NOISE OPTIMIZATION/REFINEMENT WORKS

Our primary goal is to learning noise space where diffusion guidance is distilled. This objective fundamentally differs from prior work on noise optimization or refinement. As a result, direct comparisons are not entirely fair. Several other studies pursue distinct objectives, such as layout synthesis (Mao et al., 2023b), rare concept generation (Samuel et al., 2024), or prompt alignment (Guo et al., 2024). Also, method of (Eyring et al., 2024) is restricted to timestep distilled (one-step) diffusion models, where comparisons with multi-step models are infeasible due to memory constraints. Nevertheless, our approach is related in terms of improved noise initialization, and thus partial comparisons can still be informative. Thus, we present some comparisons to provide useful insights for the research community.

### A.9.1   INITNO

We compare image quality with (Guo et al., 2024) using a subset of the Attend-and-Excite prompt dataset and Stable Diffusion 2.1, optimizing only the initial noise. Since there are no corresponding ground truth images for the Attend-and-Excite prompts, we evaluate both human preference metrics and prompt alignment scores. As shown in Tab. 9, our method outperforms in all metrics, especially in the setting guidance is not used.

### A.9.2   PAHI

PAHI (Kim et al., 2024) exists under the category of noise manipulation. To the best of our knowledge, this work is the first in its focus on learning the noise space itself, rather than optimizing or selecting. Therefore, we compare our proposed approach with this methodology PAHI (Prompt Adaptive Human preference Inversion) (Kim et al., 2024) in this section.

There are several key differences between the two approaches. First, the tasks being addressed are distinct. While PAHI (Kim et al., 2024) aims at generating outputs aligned with human preferences, our objective is to replace conventional guidance mechanisms entirely. Second, our method offers much greater flexibility. PAHI (Kim et al., 2024) assumes that sampling from certain $\mathcal{N}(\mu, \Sigma)$ instead of a standard normal Gaussian distribution is more beneficial and predict $\mu$ and $\Sigma$. However, this assumption lacks a strong theoretical foundation. In contrast, our approach aims to learn a gaussian-free noise space without imposing such constraints. Additionally, while PAHI (Kim et al., 2024) is limited to few-step models due to the computational overhead of backpropagation, our approach leverages MSD loss, enabling the use of full-step models without modification.

| Stage | Formula | # of NFE |
|---|---|---|
| Dataset Generation | $(2N' \cdot D + 1 \cdot V) \cdot M$ | 2,050K |
| Post-Training | $(1 \cdot R + N \cdot D + 1 \cdot B) \cdot M \cdot E$ | 3,600K |
| Total Training Cost | Dataset Generation + Post-Training | 5,650K |
| Original SD2.1 Training | Batch Size (2048) * Total Step (1.69M) * $(1 \cdot D + 1 \cdot V + 1 \cdot B)$ | 10.383B |

Table 8: **Training cost analysis based on NFE (Number of Function Evaluation).**

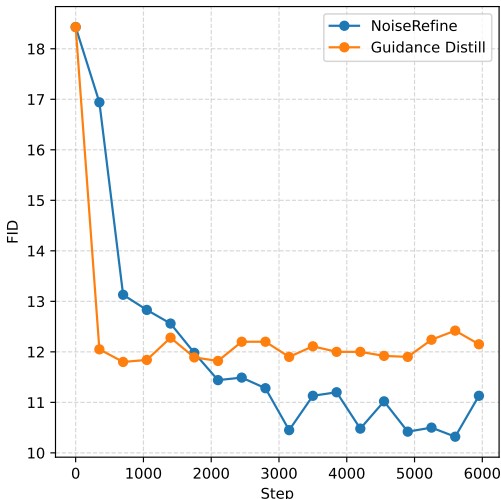

Figure 26: **Comparison of training efficiency with guidance distillation on SiT-XL/2 (Ma et al., 2024).**

Although the official code for PAHI (Kim et al., 2024) is unavailable, we adhere to the guidelines presented in their paper as possible and compare with our method. Specifically, we compare noise refining network with the setup that samples noise from $\mathcal{N}(\mu, \Sigma)$ where $\mu$ and $\Sigma$ is predicted by MLP for a given prompt. Both models are trained with filtered 20K MS COCO(Lin et al., 2014) dataset for 25K steps using two RTX 3090 GPUs. Example qualitative results of employing MLP are presented in Fig. 27, and quantitative comparisons are shown in Tab.10. Across both evaluations, noise refining network outperforms the other setup by a significant margin, showing the effectiveness of our proposed method.

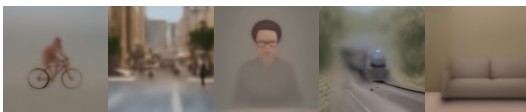

Figure 27: **Qualitative results when employing a shallow 2-layer MLP for estimating Gaussian parameters, as proposed by (Kim et al., 2024).** The results are significantly blurry, indicating that the simple approach of predicting $\mu$ and $\Sigma$ under the assumption that the optimal noise lies within $\mathcal{N}(\mu, \Sigma)$ performs poorly.

| Method | FID |
|---|---|
| MLP (Kim et al., 2024) estimating Gaussian parameters | 217.30 |
| **Noise refining network** | 13.74 |

Table 10: **Quantitative results when employing a shallow 2-layer MLP for estimating Gaussian parameters, as proposed by (Kim et al., 2024).**

### A.10 ROBUSTNESS TO THE NUMBER OF DENOISING STEPS AND SAMPLERS

Since noise refining network is trained with a fixed sampler (DDIM (Song et al., 2020a)) and denoising steps (10), concerns arise regarding its performance when using different samplers or denoising steps. To examine the impact of varying samplers and denoising steps, we conduct experiments com-

| Guidance | Initial Noise | PickScore | HPSv2 | AES | ImageReward | CLIPScore |
|:---:|:---:|:---:|:---:|:---:|:---:|:---:|
| ✗ | Gaussian | 19.78 | 0.174 | 5.073 | -1.684 | 24.95 |
| ✗ | InitNo | 19.80 | 0.176 | 5.071 | -1.666 | 25.02 |
| ✗ | Refined (Ours) | **21.14** | **0.241** | **5.389** | **-0.307** | **30.31** |
| ✓ | Gaussian | 21.67 | 0.260 | 5.525 | 0.368 | 32.25 |
| ✓ | InitNo | 21.68 | 0.261 | 5.524 | 0.376 | 32.26 |
| ✓ | Refined (Ours) | **21.83** | **0.276** | **5.571** | **0.533** | **32.51** |

Table 9: **Comparison with different noise initialization methods under guided (top) and unguided (bottom) settings.**

paring qualitative results across diverse configurations. For comparison, we select DPM++ SDE (Lu et al., 2022), DPM++ 2M (Lu et al., 2022), and EDM (Karras et al., 2022), using the prompt *"a photo of a cat"*. The results, presented in Fig. 28, show that our refined noise consistently produces reliable outputs regardless of the denoising timestep or sampler. This demonstrates the robustness of noise refining network across diverse samplers and denoising step configurations.

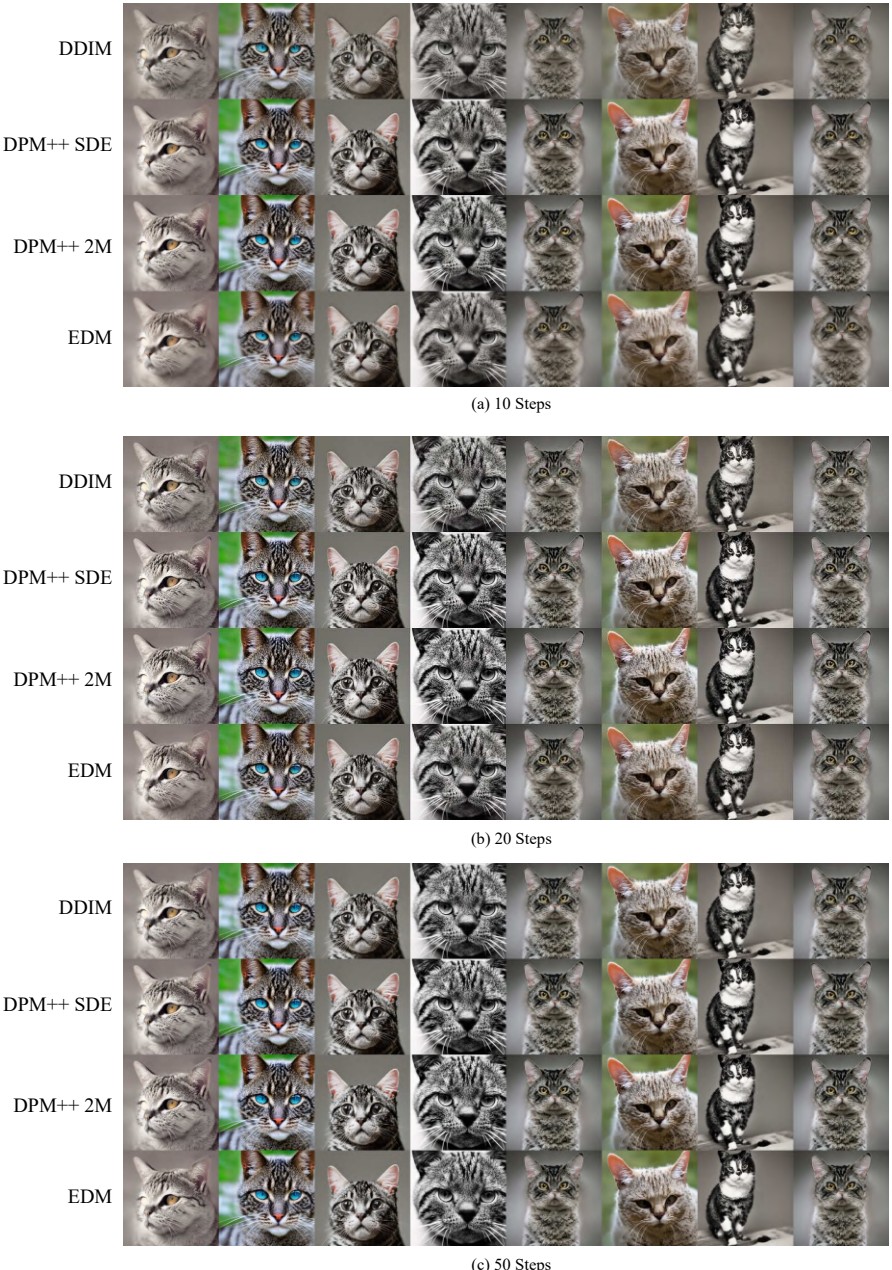

Figure 28: **Inference results on our refined noise in various denoising steps and sampler settings.** (a), (b), and (c) present inference results employing different samplers at denoising steps of 10, 20, and 50, respectively. The consistency observed across these results highlights the robustness of our refined noise to variations in both denoising steps and samplers.

A.11   ANALYSIS OF DDIM INVERSION

Following Tab.2 in the DDIM (Song et al., 2020a), we evaluate reconstruction quality under different numbers of DDIM inversion steps. Using 100 COCO (Lin et al., 2014) prompts, we first generate images with CFG (Ho & Salimans, 2022) scale 7.0 and PAG (Ahn et al., 2024) scale 3.0, and then perform inversion and reconstruction using the same number of DDIM steps. Qualitative and quantitative results are reported in Fig. 29 and Tab. 11, respectively.

The results show that simply increasing the number of inversion steps does not necessarily improve reconstruction quality. Instead, performance stabilizes within a moderate range, typically around 50–200 steps. This observation is consistent with prior findings of ReNoise (Garibi et al., 2024), whose Fig. 8 also indicates that more steps of DDIM Inversion do not always lead to better reconstructions.

Due to the inherent approximation nature of DDIM inversion, even within this favorable step range, perfect reconstructions remain challenging. Fine-grained details are often lost compared to the original images, highlighting the intrinsic limitations of the DDIM inversion.

| DDIM Inversion Step | 10 | 20 | 50 | 199 | 200 | 500 | 999 |
|---|---|---|---|---|---|---|---|
| Avg. Error | 85727.84 | 58266.11 | 24523.32 | 20565.77 | 25935.91 | 46407.59 | 63493.84 |
| Avg. PSNR | 12.16 | 15.82 | 23.24 | 24.83 | 22.93 | 17.71 | 14.84 |

Table 11: **Quantitative reconstruction error analysis across different DDIM inversion steps.**

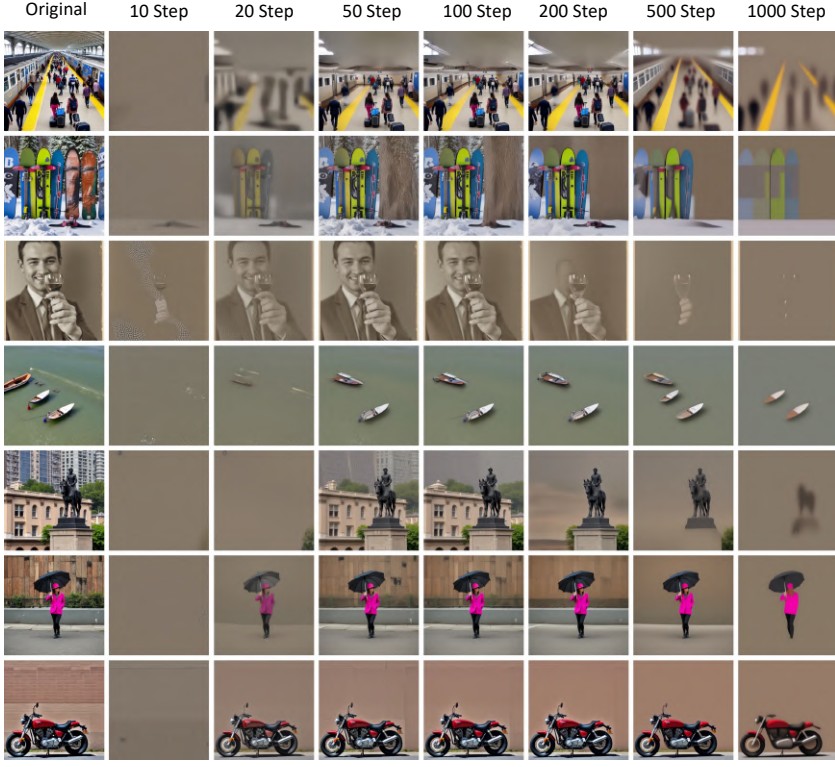

Figure 29: **Qualitative reconstruction error analysis across different DDIM inversion steps.**

## A.12 COMPARISONS WITH DIRECT TRAINING WITH CFG-GENERATED DATA.

An alternative to our approach is to directly train or fine-tune the diffusion model on CFG-generated images, effectively learning to map the initial noise $\mathbf{x}_T$ to the guided output $\mathbf{x}_0^{\text{guide}}$. To explore this, we fine-tune the SiT-XL/2 model on 10K images generated using classifier-free guidance (CFG) with guidance scale $w = 4.0$ and 20-step Euler sampling—the same dataset used in Tab. 2 (SiT-XL/2 row). The results are shown in Tab. 12.

| Model | Initial Noise | Sampling Guidance | Training Dataset | FID $\downarrow$ | IS $\uparrow$ |
|---|---|---|---|---|---|
| | Gaussian | ✗ | Original | 18.43 | 40.00 |
| SiT-XL/2 | Gaussian | ✗ | CFG-Generated | 12.31 | 58.59 |
| | Gaussian | ✗ (*Guidance Distil.*) | Original | 11.90 | 59.14 |
| | ***Refined*** (Ours) | ✗ | CFG-Generated | 10.42 | 50.39 |

Table 12: **Comparison with direct mapping from noise to guided images.**

Interestingly, we observe that this direct mapping yields noticeable quality improvements. However, its performance remains slightly worse than both guidance distillation and our proposed noise refining method in terms of FID. Furthermore, both guidance distillation and fine-tuning on CFG-generated images share the risk of catastrophic forgetting and lack flexibility across domains or sampler configurations. In contrast, ***NoiseRefine*** preserves the pretrained diffusion model throughout the denoising process. It modifies only the initial noise and thus remains compatible with domain-finetuned backbones, few-step or single-step samplers, and alternative denoising strategies without retraining the base model (see Sec. 4.3, Appendix A.10). This model-preserving property is a key design goal of our method: to distill guidance into the initial condition without altering the diffusion pipeline. If one allows modifying the denoising model itself, existing guidance distillation methods already provide a more direct and stable solution than fine-tuning on generated samples.

Nevertheless, these results raise several interesting research questions. What specific properties of CFG-generated images reduce the need for guidance at inference time? Can such properties be characterized and used to identify or filter natural images that are inherently more amenable to guidance-free generation? Exploring these questions in depth is an interesting direction for future research.

### A.13 COMPARISON UNDER THE SAME SETTINGS AS THE SiT PAPER

In the main experiment, we reported SiT-XL/2 results using a CFG scale of 4.0, as higher guidance scales often yield slightly worse FID but produce qualitatively superior samples.

For clearer comparison with the original SiT results, we use a CFG scale of 1.5, expand the evaluation to the full 1K ImageNet classes, and adopt the evaluation protocol of the SiT paper (Ma et al., 2024). Specifically, we follow their setup by using the second-order Heun sampler with 250 NFE (equivalently, 125 denoising steps) for all procedures, including dataset generation, training, and evaluation. Due to limited computational budget, this comparison experiment was trained for 5 epochs, whereas the main experiment used 8 epochs.

We report quantitative results in Tab.13 and qualitative results in Fig.30. Both evaluations show that *NoiseRefine* produces refined noise that achieves image quality comparable to that of CFG samples under the SiT evaluation setting.

| Model | Initial Noise | Sampling Guidance | FID ↓ | IS ↑ |
|---|---|---|---|---|
| | Gaussian | ✗ | 9.35 | 126.06 |
| SiT-XL/2 | Gaussian | ✓ | 2.15 | 258.09 |
| | *Refined* (Ours) | ✗ | 4.50 | 173.48 |

Table 13: **Quantitative results of SiT-XL/2 with settings of the SiT paper (Ma et al., 2024).** The reference values used for comparison were sourced from the original paper.

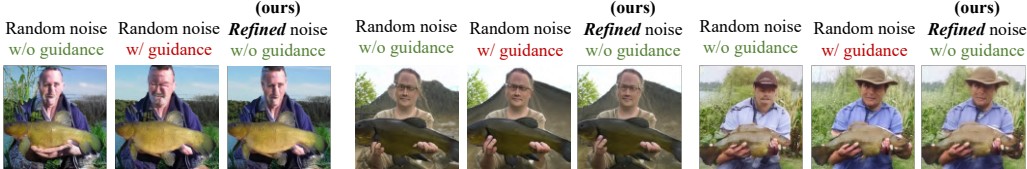

Figure 30: **Qualitative results of SiT-XL/2 with settings of the SiT paper (Ma et al., 2024).**

# B    THEORETICAL BACKGROUND

## B.1    PRELIMINARIES

**Denoising Diffusion Probabilistic Models (DDPM).**    DDPM Ho et al. (2020) defines a forward process that derives $x_t$ by adding Gaussian noise to the image $x_{t-1}$ according to the variance schedule, and a reverse process that samples $x_{t-1}$ from $x_t$, both as a Markovian chain. The forward process is defined as

$$q(x_t|x_{t-1}) = \mathcal{N}\left(x_t; \sqrt{\frac{\alpha_t}{\alpha_{t-1}}}\, x_{t-1}, \left(1 - \frac{\alpha_t}{\alpha_{t-1}}\right)\mathbf{I}\right), \tag{7}$$

$$q(x_t|x_0) = \mathcal{N}(x_t; \sqrt{\alpha_t}\, x_0, (1 - \alpha_t)\mathbf{I}), \tag{8}$$

with noise rate at timestep $t$ as $1 - \alpha_t/\alpha_{t-1}$, where $\alpha_t$ denotes noise scaling factors up to time step $t$. The reverse process is defined below.

$$p_\theta(x_{t-1}|x_t) = \mathcal{N}\left(x_{t-1}; \mu_\theta^{(t)}(x_t), \sigma_t^2\mathbf{I}\right). \tag{9}$$

To reparameterize the equation using

$$x_t = \sqrt{\alpha_t}x_0 + \sqrt{1 - \alpha_t}\,\epsilon \quad \text{for } \epsilon \sim \mathcal{N}(0, \mathbf{I}), \tag{10}$$

and $\epsilon_\theta$, which is a function approximator for predicting $\epsilon$ from $x_t$, the inference process becomes

$$x_{t-1} = \frac{1}{\sqrt{\frac{\alpha_t}{\alpha_{t-1}}}}\left(x_t - \frac{1 - \frac{\alpha_t}{\alpha_{t-1}}}{\sqrt{1 - \alpha_t}}\,\epsilon_\theta^{(t)}(x_t)\right) + \sigma_t z, \tag{11}$$

Where $z \sim \mathcal{N}(0, \mathbf{I})$ and $\sigma_t^2$ denotes the variance of Gaussian trainsitions .The objective of DDPM is defined as

$$L_{\text{simple}}(\theta) = \mathbb{E}_{t,x_0,\epsilon}\left[\|\epsilon - \epsilon_\theta^{(t)}(x_t)\|^2\right], \tag{12}$$

where the L2 loss between the actual noise $\epsilon$ added during training and the noise prediction $\epsilon_\theta(x_t, t)$ for uniformly sampled $t \in \{1, ..., T\}$.

**Denoising Diffusion Implicit Models (DDIM).**    DDIM Song et al. (2020a) consider the following inference distributions:

$$q_\sigma(x_{1:T}|x_0) := q_\sigma(x_T|x_0)\prod_{t=2}^{T} q_\sigma(x_{t-1}|x_t, x_0). \tag{13}$$

with a mean function as below.

$$q_\sigma(x_{t-1}|x_t, x_0) = \mathcal{N}\left(\sqrt{\alpha_{t-1}}x_0 + \sqrt{1 - \alpha_{t-1} - \sigma_t^2} \cdot \frac{x_t - \sqrt{\alpha_t}x_0}{\sqrt{1 - \alpha_t}}, \sigma_t^2\mathbf{I}\right). \tag{14}$$

Distinctively from DDPM, the forward process is Non-Markovian since each $x_t$ could depend on both $x_{t-1}$ and $x_0$. Reparameterizing with $\epsilon_\theta$, we can sample $x_{t-1}$ from $x_t$ through an equation:

$$x_{t-1} = \sqrt{\alpha_{t-1}}\underbrace{\left(\frac{x_t - \sqrt{1 - \alpha_t}\,\epsilon_\theta^{(t)}(x_t)}{\sqrt{\alpha_t}}\right)}_{\text{predicted } x_0} + \sqrt{1 - \alpha_{t-1} - \sigma_t^2} \cdot \epsilon_\theta^{(t)}(x_t) + \sigma_t\epsilon_t \tag{15}$$

$$= a_t x_t + b_t \epsilon_\theta^{(t)}(x),$$

where $\epsilon_t \sim \mathcal{N}(0, \mathbf{I})$ and $a_t = \sqrt{\alpha_{t-1}}/\sqrt{\alpha_t}, \quad b_t = \sqrt{1 - \alpha_{t-1}} - a_t\sqrt{1 - \alpha_t}$.

The objective of DDIM is the same as that of DDPM:

$$L_{\text{DDIM}}(\theta) = \mathbb{E}_{t,x_0,\epsilon}\left[\|\epsilon - \epsilon_\theta^{(t)}(x_t)\|^2\right]. \tag{16}$$

**Denoising and inversion process.** We denote the denoising process as $\text{Denoise}(x_T)$. When using the DDIM sampler (Song et al., 2020a), the denoising process is defined as:

$$\text{Denoise}(x_T) := D_1\left(\ldots D_T(g_\phi(x_T))\right), \tag{17}$$

where each step $D_t$ is given by:

$$D_t(x) := a_t x_t + b_t \epsilon_\theta^{(t)}(x). \tag{18}$$

The guided denoising process, denoted as $\text{Denoise}^{\text{Guide}}(x_T, c)$, follows the same steps as Eq. 17, but replaces $\epsilon_\theta^{(t)}(x)$ with guided scores, such as the classifier-free guided score $\epsilon_\theta^{\text{CFG}}(x_t, c)$ (Ho & Salimans, 2022), the perturbed-attention guided score $\epsilon_\theta^{\text{PAG}}(x_t)$ (Ahn et al., 2024), or a combination of both ($\epsilon_\theta^{\text{CFG,PAG}}(x_t)$). These guided scores are defined in Eqs. 31 and 32.

While we utilize the DDIM scheduler in this work, any other diffusion scheduler (Ho et al., 2020; Song et al., 2020a; Karras et al., 2022) can be used by appropriately modifying $a_t$ and $b_t$.

For the inversion process $\text{Inversion}(x_0, c)$, we follow the method in (Garibi et al., 2024) to obtain the initial noise $x_T$, which can be denoised back to the given image $x_0$ without employing any guidance methods (Ho & Salimans, 2022; Ahn et al., 2024) during inversion.

## B.2 DERIVATIONS

**Proposition 1.** Let $x_T$ be an initial noise, and suppose that $x_0$ is the image obtained through denoising. Assuming Lipschitz continuity with distance metric $d$, for every $x_T$, there exists a constant $\kappa > 0$ such that the following holds:

$$d(x_T, x_T^{\text{Guide}\dagger}) < \kappa d(x_0, x_0^{\text{Guide}}).$$

*proofs.* The Lipschitz condition is expressed as follows:

$$d(\epsilon_\theta^{(t)}(x), \epsilon_\theta^{(t)}(y)) \le L_t d(x, y), \tag{19}$$

where $L_t$ is constant dependent on $t$, $x$ and $y$ are arbitrary inputs to $\epsilon_\theta^{(t)}$. DDIM step in terms of $x_t$ can be expressed as follows:

$$x_{t-1} = \sqrt{\frac{\alpha_{t-1}}{\alpha_t}} x_t + \left(\sqrt{1 - \alpha_{t-1}} - \sqrt{\frac{\alpha_{t-1}(1 - \alpha_t)}{\alpha_t}}\right) \epsilon_\theta^{(t)}(x_t). \tag{20}$$

Eq. 20 can be expressed in terms of $x_t^{\text{Guide}\dagger}$ which is denoised from $x_T^{\text{Guide}\dagger}$. With those equations, we can get the following equation,

$$x_{t-1} - x_{t-1}^{\text{Guide}\dagger} = \sqrt{\frac{\alpha_{t-1}}{\alpha_t}}(x_t - x_t^{\text{Guide}\dagger}) + \left(\sqrt{1 - \alpha_{t-1}} - \sqrt{\frac{\alpha_{t-1}(1 - \alpha_t)}{\alpha_t}}\right)(\epsilon_\theta^{(t)}(x_t) - \epsilon_\theta^{(t)}(x_t^{\text{Guide}\dagger}))$$

$$= \sqrt{\frac{\alpha_{t-1}}{\alpha_t}}(x_t - x_t^{\text{Guide}\dagger}) - \gamma_t(\epsilon_\theta^{(t)}(x_t) - \epsilon_\theta^{(t)}(x_t^{\text{Guide}\dagger})),$$

where $\gamma_t = \left(\sqrt{\alpha_{t-1}(1 - \alpha_t)/\alpha_t} - \sqrt{1 - \alpha_{t-1}}\right) > 0$. If the distance metric $d$ have translation invariance, the equation can be expressed as follows with Eq. 19:

$$d(x_{t-1}, x_{t-1}^{\text{Guide}\dagger}) \le \sqrt{\frac{\alpha_t}{\alpha_{t-1}}}(1 + \gamma_t L_t)d(x_t, x_t^{\text{Guide}\dagger}). \tag{21}$$

Recursively organizing Eq. 21 for $t = T, T - 1, \ldots, 1$, it can be expressed as follows:

$$d(x_T, x_T^{\text{Guide}\dagger}) \le \left(\prod_{t=1}^{T}(1 + \gamma_t L_t)\right)\sqrt{\frac{\alpha_T}{\alpha_0}}d(x_0, x_0^{\text{Guide}\dagger}). \tag{22}$$

Since $\alpha_T$ is close to 0, using $d(x_0, x_0^{\text{Guide}\dagger})$ is sufficient to directly learn $x_T^{\text{Guide}\dagger}$ if $d(x_0, x_0^{\text{Guide}\dagger})$ is small enough.

**Proposition 2.** By approximating the gradients through Multistep Score Distillation (MSD) using detached gradients at each step, we approximate the full-gradient objective with a mild assumption. In conclusion, the two gradients can be approximated as follows:

$$\nabla_\phi \mathcal{L}_{\text{Denoise}}(g_\phi(x_T), \theta) \approx k\nabla_\phi \mathcal{L}_{\text{MSD}}(g_\phi(x_T); \theta), \tag{23}$$

where $k \in (0, 1)$ is constant.

***Warmup.*** We begin by recalling the typical denoising process in DDIM sampling:

$$D_1 \circ D_2 \circ \cdots \circ D_T(x_T), \tag{24}$$

where $D_t(x) = a_t x + b_t \epsilon_\theta^{(t)}(x)$ denotes a single denoising step, and $a_t$, $b_t$ are DDIM-derived coefficients (Song et al., 2020a).

The final generated image $\bar{x}_0$, obtained by applying the full denoising trajectory to the refined noise, is:

$$\bar{x}_0 = D_1 \circ D_2 \circ \cdots \circ D_T(g_\phi(x_T)), \tag{25}$$

where $g_\phi(x_T)$ is the refined noise output from the refining network $g_\phi$.

From Eq. 15, the denoised image from the full DDIM denoising process $\bar{x}_0$ is:

$$\bar{x}_0 = \sqrt{\frac{\alpha_0}{\alpha_T}} g_\phi(x_T) - \sum_{t=1}^{T} \sqrt{\frac{\alpha_0}{\alpha_{t-1}}} \gamma_t \epsilon_\theta^{(t)}(x_t). \tag{26}$$

This expression is derived by recursively applying the DDIM update rule (Eq. 20) from $x_T$ to $x_0$, unfolding the full denoising trajectory step by step. The result is a closed-form expression for the cumulative DDIM trajectory, expressing $\hat{x}_0$ as a function of the refined noise $g_\phi(x_T)$ and intermediate model predictions.

We define the denoising loss as follows:

$$\mathcal{L}_{\text{Denoise}}(g_\phi(x_T); \theta) := d\left(\bar{x}_0, x_0^{\text{Guide}}\right) \tag{27}$$

where $d(\cdot, \cdot)$ denotes the L2 distance.

***proofs.*** Since the only difference between the two losses is the stop gradient in the diffusion model and all other components are identical, it suffices, by the chain rule, to show that the gradient of $F_1(F_2(\ldots F_T(g_\phi(x_T)))$ with respect to $\phi$ is proportional to the gradient of $\text{Denoise}(g_\phi(x_T))$ with respect to $\phi$. The derivation proceeds as follows:

$$\begin{aligned}
\nabla_\phi \text{Denoise}(g_\phi(x_T)) &= \nabla_\phi \left( \sqrt{\frac{\alpha_0}{\alpha_T}} g_\phi(x_T) - \sum_{t=1}^{T} \sqrt{\frac{\alpha_0}{\alpha_{t-1}}} \gamma_t \epsilon_\theta^{(t)}(x_t) \right) \\
&= \left( \sqrt{\frac{\alpha_0}{\alpha_T}} I - \sum_{t=1}^{T} \gamma_t \sqrt{\frac{\alpha_0}{\alpha_{t-1}}} \frac{\partial \epsilon_\theta^{(t)}(x_t)}{\partial x_t} \frac{\partial x_t}{\partial g_\phi(x_T)} \right) \frac{\partial g_\phi(x_T)}{\partial \phi}.
\end{aligned} \tag{28}$$

As detailed in B.3, the term $\partial \epsilon_\theta^{(t)}(x_k)/\partial x_k$ can be approximated as being proportional to the identity matrix. Additionally, the term $\partial x_k/\partial g_\phi(x_T)$ can be expressed in terms of $\partial \epsilon_\theta^{(t)}(x_k)/\partial x_k$. Then, each component of $\partial \epsilon_\theta^{(t)}(x_k)/\partial x_k$ can be approximated by the identity matrix. Consequently, $(\partial \epsilon_\theta^{(t)}(x_k)/\partial x_k)(\partial x_k/\partial g_\phi(x_T))$ becomes proportional to the identity matrix. Denoting the proportionality constant as $\eta_t := \left( \frac{\partial \epsilon_\theta^{(t)}(x_t)}{\partial x_t} \cdot \frac{\partial x_t}{\partial g_\phi(x_T)} \right)$, Eq. 28 is simplified as follows:

$$\begin{aligned}
Eq.\ 28 &= \left( \sqrt{\frac{\alpha_0}{\alpha_T}} - \sum_{t=1}^{T} \sqrt{\frac{\alpha_0}{\alpha_{t-1}}} \gamma_t \eta_t \right) \frac{\partial g_\phi(x_T)}{\partial \phi} \\
&= \left( 1 - \sqrt{\alpha_T} \sum_{t=1}^{T} \frac{1}{\sqrt{\alpha_{t-1}}} \gamma_t \eta_t \right) \sqrt{\frac{\alpha_0}{\alpha_T}} \frac{\partial g_\phi(x_T)}{\partial \phi} \\
&= \left( 1 - \sqrt{\alpha_T} \sum_{t=1}^{T} \frac{1}{\sqrt{\alpha_{t-1}}} \gamma_t \eta_t \right) \nabla_\phi F_1(F_2(\ldots F_T(g_\phi(x_T)))).
\end{aligned} \tag{29}$$

### B.3 DIFFUSION MODEL JACOBIAN APPROXIMATION

**Why is the approximation possible?** In this subsection, we present experimental results demonstrating that the Jacobian of the diffusion model $\epsilon_\theta^t$ with respect to the input $x_t$ can be approximated as proportional to the identity matrix. Fig. 31 illustrates the Jacobian $\partial \epsilon_\theta^t / \partial x_t$. We observe that the Jacobian behaves like the identity matrix regardless of timestep, except when $t$ is significantly small. To quantify this observation, we plot the distributions of the Jacobian's diagonal and off-diagonal elements across timesteps in Fig. 32 (log scale). The off-diagonal elements are consistently much smaller and concentrated near zero, while the diagonal elements remain significantly larger, confirming the strong diagonal dominance of the Jacobian. Fig. 33 shows the same analysis for DiT-XL/2 Peebles & Xie (2023), where we observe a similar pattern: the off-diagonal values stay close to zero across timesteps, whereas the diagonal values remain substantially larger. This demonstrates that the identity-matrix-like Jacobian structure holds not only for Stable Diffusion 2.1 but also for transformer-based models such as DiT.

In such cases, the deviation does not affect our primary analysis. According to the results of **Proposition 1**, the timestep-dependent constant $\frac{1}{\sqrt{\alpha_{t-1}}}\gamma_t$ multiplied to each Jacobian term $\eta_t$ is expressed as follows:

$$\frac{1}{\sqrt{\alpha_{t-1}}}\gamma_t = \sqrt{\frac{1-\alpha_t}{\alpha_t}} - \sqrt{\frac{1-\alpha_{t-1}}{\alpha_{t-1}}}. \tag{30}$$

This value can be numerically determined based on the scheduling, and in the case of DDIM (Song et al., 2020a), it is presented in Fig. 34. The graph shows that the constant decreases toward zero as t approaches 0.

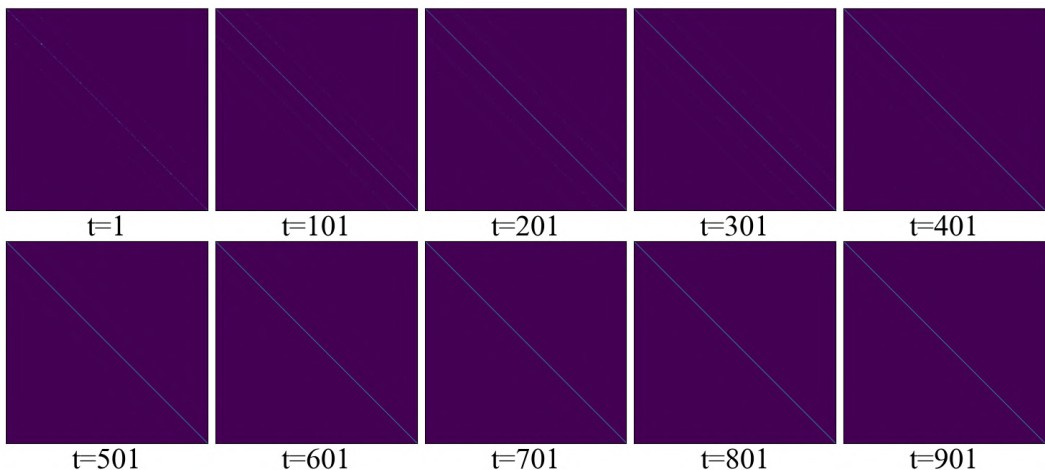

Figure 31: **Visualization of Jacobian of a denoising network.** Starting from $T = 1000$, we performed denoising over 10 steps and plotted the Jacobian heatmap at each timestep. We extracted a $500 \times 500$ section from the full Jacobian matrix for visualization. Each plot demonstrates that the Jacobian is close to the identity matrix.

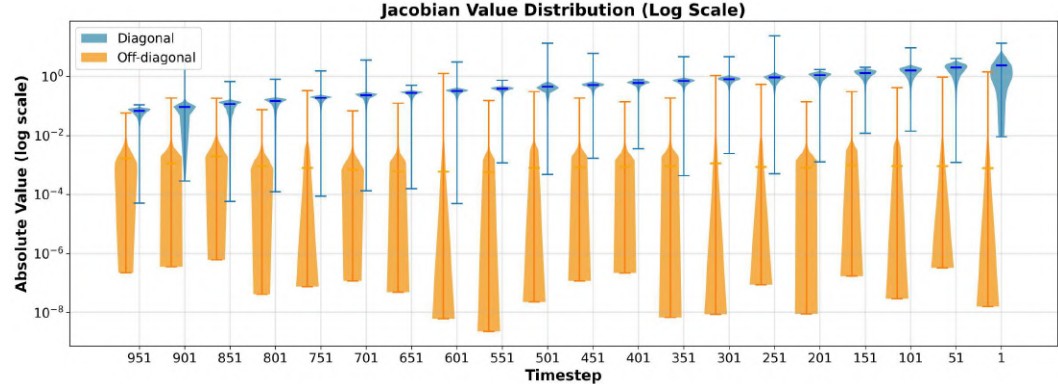

Figure 32: **Distribution of Jacobian elements in Stable Diffusion 2.1 (log scale).** Scatter plot of diagonal and off-diagonal Jacobian magnitudes across timesteps. The off-diagonal elements remain close to zero, while the diagonal elements are significantly larger, demonstrating strong diagonal dominance of the Jacobian.

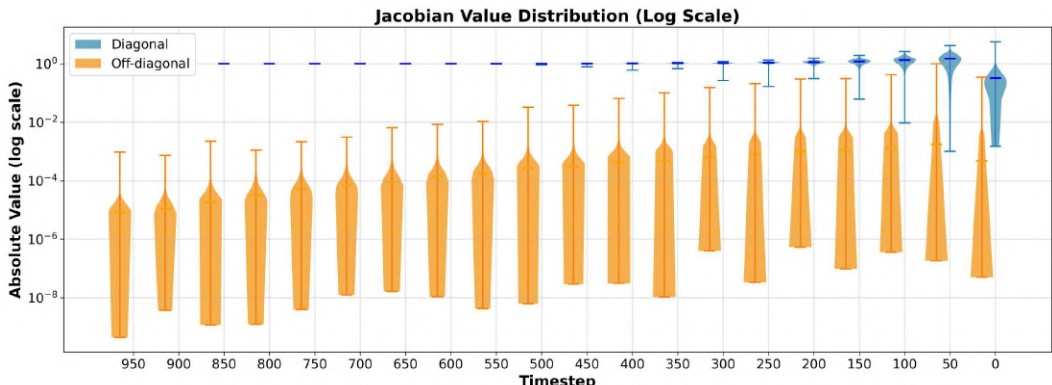

Figure 33: **Distribution of Jacobian elements in DiT/XL-2 (log scale).** Violin plot of diagonal and off-diagonal Jacobian values across timesteps for the DiT/XL-2 Peebles & Xie (2023). Despite its transformer structure and larger receptive field, the off-diagonal values remain close to zero, showing an identity-like Jacobian similar to Stable Diffusion.

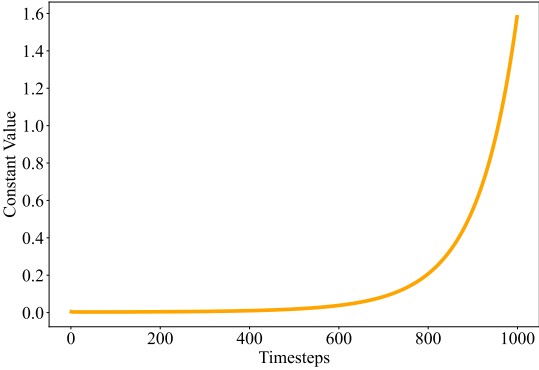

Figure 34: **Visualization of constant values over timesteps.** Visualization of the time-dependent constant value $\frac{\gamma_t}{\sqrt{\alpha_{t-1}}}$ corresponding to Eq. 30 across different timesteps. The results numerically demonstrate that for small timesteps, where the Jacobian deviates from the identity matrix, the multiplied constant values are sufficiently close to zero.

**Why does the approximation *enhance* performance?**   The above analysis explains why the Jacobian can be approximated by the identity matrix, but it does not address why this approximation empirically improves optimization, yielding faster convergence and higher-quality results. Although the Jacobian is "close enough" to identity, it is not perfectly identity, especially at small timesteps. When full gradients are backpropagated through multiple denoising steps, these small off-diagonal components accumulate across steps and induce optimization instability, similar to exploding and vanishing gradients in recurrent networks.

To illustrate this effect, we perform a simple toy experiment in which we directly optimize an initial Gaussian noise $x_T$ so that its denoised output $\hat{x}_0$ matches a given target image. Fig. 35 shows the MSE loss $||\hat{x}_0 - x_{\text{target}}||^2$ and the gradient norm during optimization. Using the full-step gradient leads to unstable behavior: the loss fails to converge, and the gradient norm becomes large and highly erratic. We also visualize the gradient norm $||\partial\mathcal{L}/\partial x_t||^2$ at each denoising step (Fig. 36 top), and observe that for some iterations the gradients become progressively larger as $t$ approaches 1, clear evidence of gradient explosion.

By skipping the Jacobian $\partial\epsilon_\theta^t/\partial x_t$ at each step, MSD avoids this long-horizon accumulation and yields a far more stable optimization process. As shown in Fig. 36 (bottom), the gradient norms remain well-behaved and stable throughout optimization. This behavior aligns with prior findings in score distillation sampling, and further demonstrates that gradient skipping acts as an effective regularizer that prevents instability arising from multi-step backpropagation.

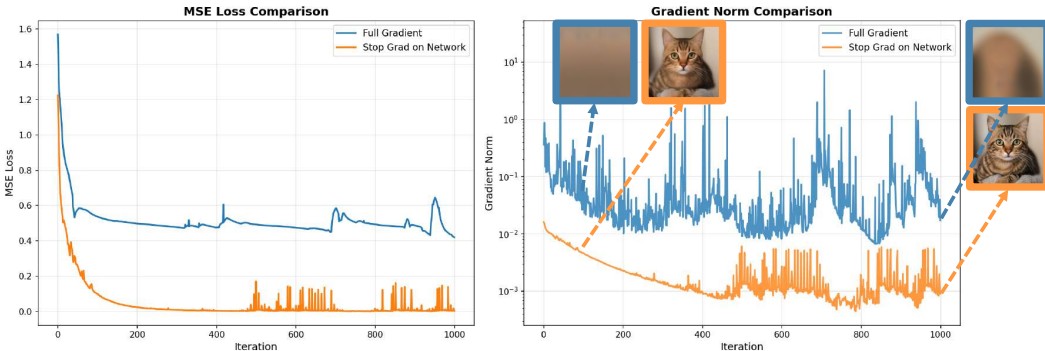

Figure 35: **Optimization instability of full-step gradients.** We optimize the initial noise so that the denoised output $\hat{x}_0$ matches a target image and plot the MSE loss (left) and gradient norm (right) over iterations. The full-step gradient exhibits unstable dynamics, with the loss failing to converge and the gradient norm becoming large and erratic. In contrast, the MSD approximation maintains stable gradients and converges reliably, demonstrating that skipping the Jacobian $\partial\epsilon_\theta^t/\partial x_t$ effectively prevents long-horizon instability.

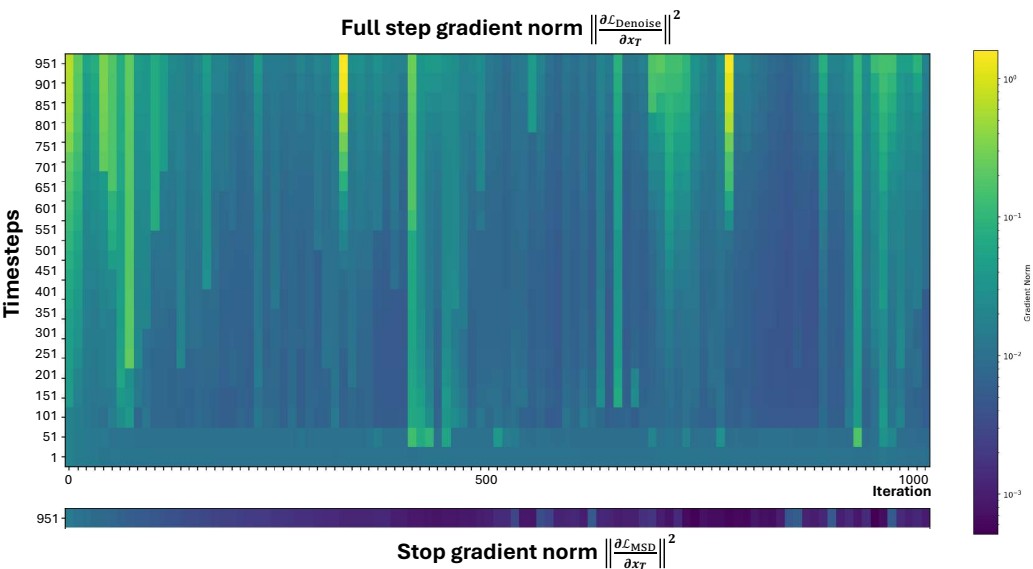

Figure 36: **Gradient explosion across denoising steps when using full-step gradients.** We visualize the gradient norm $||\partial\mathcal{L}/\partial x_t||^2$ at each denoising step during optimization. With full-step backpropagation, the gradients become progressively larger as $t$ approaches 1, and in several iterations the norm reaches values close to 1, indicating clear gradient explosion. In contrast, when applying MSD (gradient skipping), the gradients remain small and stable. This confirms that multi-step Jacobian accumulation is the primary source of instability, and that skipping the Jacobian $\partial\epsilon_\theta^t/\partial x_t$ effectively prevents this issue.

## C MORE ABLATION STUDIES

### C.1 NOISE REFINING NETWORK

To effectively leverage pretrained knowledge, we attach LoRA layers to the original model when training noise refining network. To evaluate the effectiveness of LoRA (Hu et al., 2021), we conduct an ablation by training the refining network using the same original Stable Diffusion 2.1 UNet architecture, but from scratch. We use the filtered MS COCO dataset from both datasets and train the models for 25K steps on two RTX 3090 GPUs, keeping all other experimental configurations identical. As shown in Tab. 14 and Fig. 37, the LoRA-based approach achieves faster convergence and significantly lower FID at the same iteration, demonstrating its efficiency in training. These results indicate that leveraging pretrained knowledge leads to superior performance compared to training from scratch.

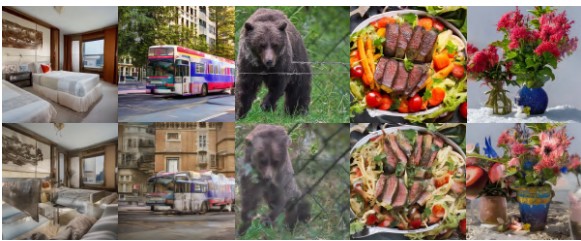

Figure 37: **Qualitative comparison with noise refining network (top) and UNet trained from scratch (bottom).**

| Model | FID |
|---|---|
| From scratch | 37.87 |
| **Pretrained (Ours)** | **13.74** |

Table 14: **Quantitative comparison with noise refining network using pretrained UNet + LoRA and UNet trained from scratch.**

| Parameter | | FID (Heusel et al., 2017) ↓ | IS (Salimans et al., 2016) ↑ |
|---|---|---|---|
| # of steps | 5 | 13.74 | 30.80 |
| | 10 | **13.36** | **32.81** |

Table 15: **Ablation study on the number of denoising steps.**

### C.2 NUMBER OF DENOISING STEPS

We analyze the impact of denoising steps by comparing $N = 5$ and $N = 10$, reporting FID (Heusel et al., 2017) and IS (Salimans et al., 2016) in Tab. 15. The results show improved performance with more steps, but high step counts ($N \geq 10$) incur prohibitive backpropagation costs, highlighting the need for MSD to mitigate computational overhead.

# D IMPLEMENTATION AND EXPERIMENTAL DETAILS

## D.1 IMPLEMENTATION DETAILS

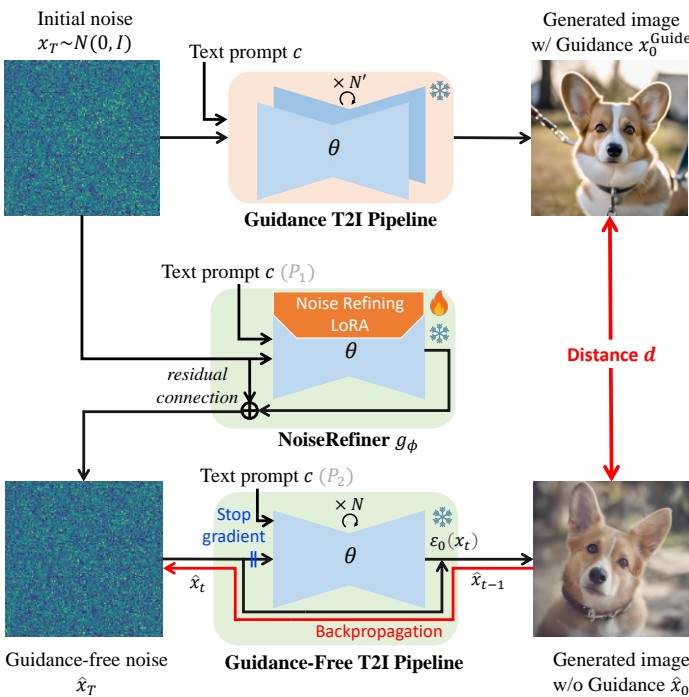

Figure 38: **Training framework with annotations.** We provide an annotated illustration of the training framework to clarify the notation in the following discussion.

**More details of our framework.** Most experiments are conducted with text-to-image diffusion models (Rombach et al., 2022; Podell et al., 2023), so we provide implementation details for these models here. Although our framework *NoiseRefine* can be generalized from pixel-level to latent-level diffusion models, in our experiments we use MSE loss in latent space for $d(x_0^{\text{Guide}}, \hat{x}_0)$.

We provide our training framework in Fig. 38. It consists of three parts: Guidance T2I Pipeline takes Gaussian noise $x_T \sim \mathcal{N}(0, \mathbf{I})$ and condition (text prompt) $c$ as inputs and generates an image $x_0^{\text{Guide}}$ with guidance methods (Ho & Salimans, 2022; Ahn et al., 2024; Hong et al., 2023; Sadat et al., 2024; Hong, 2024; Karras et al., 2024). The noise refining network $g_\phi$ refines Gaussian noise $x_T$. Guidance-Free T2I Pipeline takes refined noise $\hat{x}_T = g_\phi(x_T)$ and condition (text prompt) $c$ and generates an image $\hat{x}_0$ without guidance. For Guidance T2I Pipeline, with the denoising network $\epsilon_\theta$, we can use the guided score $\epsilon_\theta^{\text{CFG}}(x_t, c)$ for CFG (Ho & Salimans, 2022) or $\epsilon_\theta^{\text{PAG}}(x_t, c)$ for PAG (Ahn et al., 2024) in denoising process as below:

$$\epsilon_\theta^{\text{CFG}}(x_t, c) = \epsilon_\theta(x_t, c) + w(\epsilon_\theta(x_t, c) - \epsilon_\theta(x_t)), \tag{31}$$

$$\epsilon_\theta^{\text{PAG}}(x_t) = \epsilon_\theta(x_t) + s(\epsilon_\theta(x_t) - \hat{\epsilon}_\theta(x_t)), \tag{32}$$

where $w$ and $s$ denote the guidance scale of CFG (Ho & Salimans, 2022) and PAG (Ahn et al., 2024), and $c$ is for the condition. Note that the perturbed score $\hat{\epsilon}_\theta$ is from perturbing the forward process of the denoising network $\epsilon_\theta$ (Ahn et al., 2024). With the denoising step $N' = 20$, we can get the guided image $x_0^{\text{Guide}}$. Our noise refining network refines Gaussian noise $x_T$ with $g_\phi$ at timestep $t = T$, which is from the reverse step of DDIM (Song et al., 2020a) in Eq. 15. The output of noise refining network $g_\phi$ is denoted as $\hat{x}_T = g_\phi(x_T)$ and becomes the input of Guidance-Free T2I Pipeline. In this pipeline, $\hat{x}_T$ is denoised into $\hat{x}_0$ without guidance using $N$ denoising steps.

**Architecture details.** For noise refining network $g_\phi$, we use Stable Diffusion 2.1 (Rombach et al., 2022) with LoRA (Hu et al., 2021) rank of 128, applied to all attention, convolutional, and feed-

forward layers. We use DDIM (Song et al., 2020a) scheduler with the same settings as the pre-trained model. For noise refinement, we use an input timestep $T = 999$, and the default denoising step $N$ is set to 10. In Stable Diffusion XL (Lin et al., 2024), we use the same configs of Stable Diffusion 2.1 except LoRA rank which is set to 256.

## D.2 EXPERIMENTAL DETAILS

### D.2.1 TRAINING SETUP

Note that our model requires only text prompts for training, eliminating the need for real images, as we leverage self-generated images from the model we aim to train using guidance methods.

For Stable Diffusion 2.1 (Rombach et al., 2022), we train our model on 20K MS COCO prompts, 30K Pick-a-Pic prompts, using CFG scale 7.0 and PAG scale 3.0 for all generated images.

For Stable Diffusion XL (Podell et al., 2023), we train our model on 55K MS COCO prompts, 36K Pick-a-Pic prompts, and 90K LAION prompts, using the same CFG and PAG scales as for Stable Diffusion 2.1.

For SiT-XL/2 (Ma et al., 2024), we train our model on 100 classes of ImageNet (Krizhevsky et al., 2012) (class 1 to class 100) using CFG scale 4.0. Total dataset consists of 100K images, 1K images for each class. For dataset generation, we employ Euler sampler with 20 denoising steps.

For SD2.1 and SDXL, we generated images for all datasets with guidance and retained only the top-$N$ samples ranked by AES (Schuhmann, 2022) scores, where $N$ denotes the reported dataset size. For SiT-XL/2, no filtering was applied.

For the ablation study on the number of denoising steps, we primarily use SD2.1.

### D.2.2 EVALUATION SETUP

The datasets used are described in Sec. 4.1. For guided sampling, we use the same guidance scale as in the training of the noise refining network across all models. For guidance-distilled sampling with a distilled denoising network (Meng et al., 2023), since no official implementation is available, we follow Eq. 3 in their paper for reimplementation. The same dataset and guidance scale are used for training the distilled model.

For all experiments using SiT-XL/2 except A.13, we generate 50K samples using random initial noise and the Euler sampler with 20 denoising steps.

# E ADDITIONAL RESULTS

## E.1 ADDITIONAL QUALITATIVE RESULTS

We present our additional qualitative results of SD2.1 on Fig. 41, 42, 43, 44 and results of SDXL on Fig. 45. Results show that the performance of using refined noise by noise refining network is comparable to that of using guidance on random Gaussian noise. All the results are selected from images used in Tab. 2.

## E.2 USER STUDY

**Gaussian noise *vs* refined noise.** We conducted a user study to evaluate prompt adherence and image quality by comparing images generated from random Gaussian noise and our refined noise. The images are generated using randomly sampled MS COCO validation prompts, as shown in Tab. 2. The results are presented in Tab. 16. The study demonstrates that our method outperformed the baseline in all human evaluation criteria. A total of 26 participants anonymously evaluated 20 pairs of images, each pair consisting of an image generated using initial Gaussian noise and our refined noise from noise refining network. The percentage was calculated by dividing the total number of selections for each option by the total number of responses, following the same methodology as in Tab. 16.

Participants were provided with the following instructions for each pair of images:

1. Which image has better overall quality? (left/right)
2. Which image more faithfully reflects the given prompt? (left/right)

| Metric | Gaussian Noise | Refined Noise (Ours) |
|---|---|---|
| Image Quality | 3.08% | **96.92%** |
| Prompt Adherence | 6.73% | **93.27%** |

Table 16: **User study on the image quality and prompt adherence of generated images.**

**Guided sampling *vs* refined noise.** Tab. 17 shows the results of user study, confirming noise refining network's comparable to results starting from Gaussian initial noise without guidance. 45 participants compared 30 image pairs generated with guidance and our method (refined noise without guidance), using generated images for evaluation in Tab. 2, and evaluated visual appealing and prompt alignment. The instructions for the survey are the same as the above.

| Metric | Gaussian Noise + Guided Sampling | Refined Noise (Ours) |
|---|---|---|
| Image Quality | 46.04% | **53.96%** |
| Prompt Adherence | 48.24% | **51.76%** |

Table 17: **User study on the image quality and prompt adherence of generated images.**

## E.3 GENERALIZATION ON OTHER DOMAINS

Fig. 39 presents additional qualitative results across different domains, including anime and clay. The prompts used for generation are provided in Tab. 18.

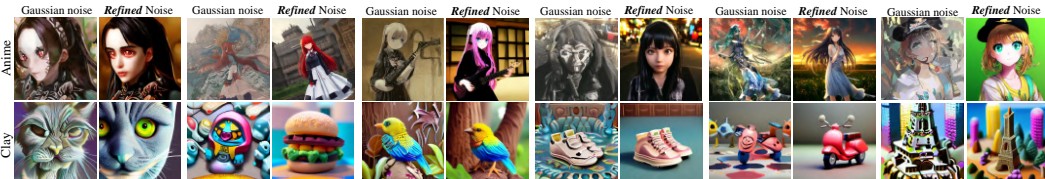

Figure 39: **Additional qualtitative results of generalization on other domains.**

| |
|---|
| (masterpiece, best quality, ultra-detailed, best shadow), (detailed background,dark fantasy), (beautiful detailed face), high contrast, (best illumination, an extremely delicate and beautiful), (cinematic light), colorful, hyper detail, dramatic light, intricate details, (1 girl, solo,black hair, sharp face,low twintails,red eyes) |
| (masterpiece,best quality), 1girl, long hair, red hair, solo, dress, red eyes, looking at viewer, long sleeves, standing, building, white dress, gloves, hair ornament, black jacket, smile, floating hair, dutch angle, closed mouth, looking away, outdoors |
| (masterpiece,best quality), 1girl, solo, black skirt, blue eyes, electric guitar, guitar, headphones, holding, holding plectrum, instrument, long hair, music, one side up, pink hair, playing guiter, pleated skirt, black shirt, indoors |
| (masterpiece, best quality, ultra detailed:1.3), perfect composition, anime, illustration 4k, (extremely detailed, hyper detailed), raw, hdr, 8k textures, extreme detail, hight detailed skin texture, high sharpness, 1girl, (detailed eyes:1.3), petite, on the street, in public, night street, night lights |
| (masterpiece, best quality, ultra detailed:1.3), A beautiful, anime-style female character with long flowing hair, wearing a flowing summer dress, standing in a field of flowers at sunset, soft pastel colors, detailed facial features |
| 1girl, aqua eyes, baseball cap, blonde hair, closed mouth, earrings, green background, hat, hoop earrings, jewelry, looking at viewer, shirt, short hair, simple background, solo, upper body, yellow shirt, (waifu, anime, exceptional, best aesthetic, new, newest, best quality, masterpiece, extremely detailed:1.2) |
| clayitization, A portrait of a black cat with piercing green eyes, Ultra-detailed, 3d, octane render, intricate details |
| clayitization, a photo of a cheese burger, ultra detailed, 3d, octane render, intricate details |
| clayitization, colorful tropical bird perched on branch, ultra detailed, 3d render, smooth clay textures, vibrant palette, 3d, octane render, soft lighting, realistic textures |
| clayitization, stylish pair of sneakers, detailed textures, 3d, vibrant colors, realistic clay appearance, octane render |
| clayitization, classic red Vespa scooter, highly detailed, glossy clay finish, 3d model, studio lighting, octane render |
| clayitization, a photo of the Eiffel Tower, ultra detailed, intricate architectural details, 3d, octane render |

Table 18: **Example prompts used for domain generalization experiment (Fig 39).**

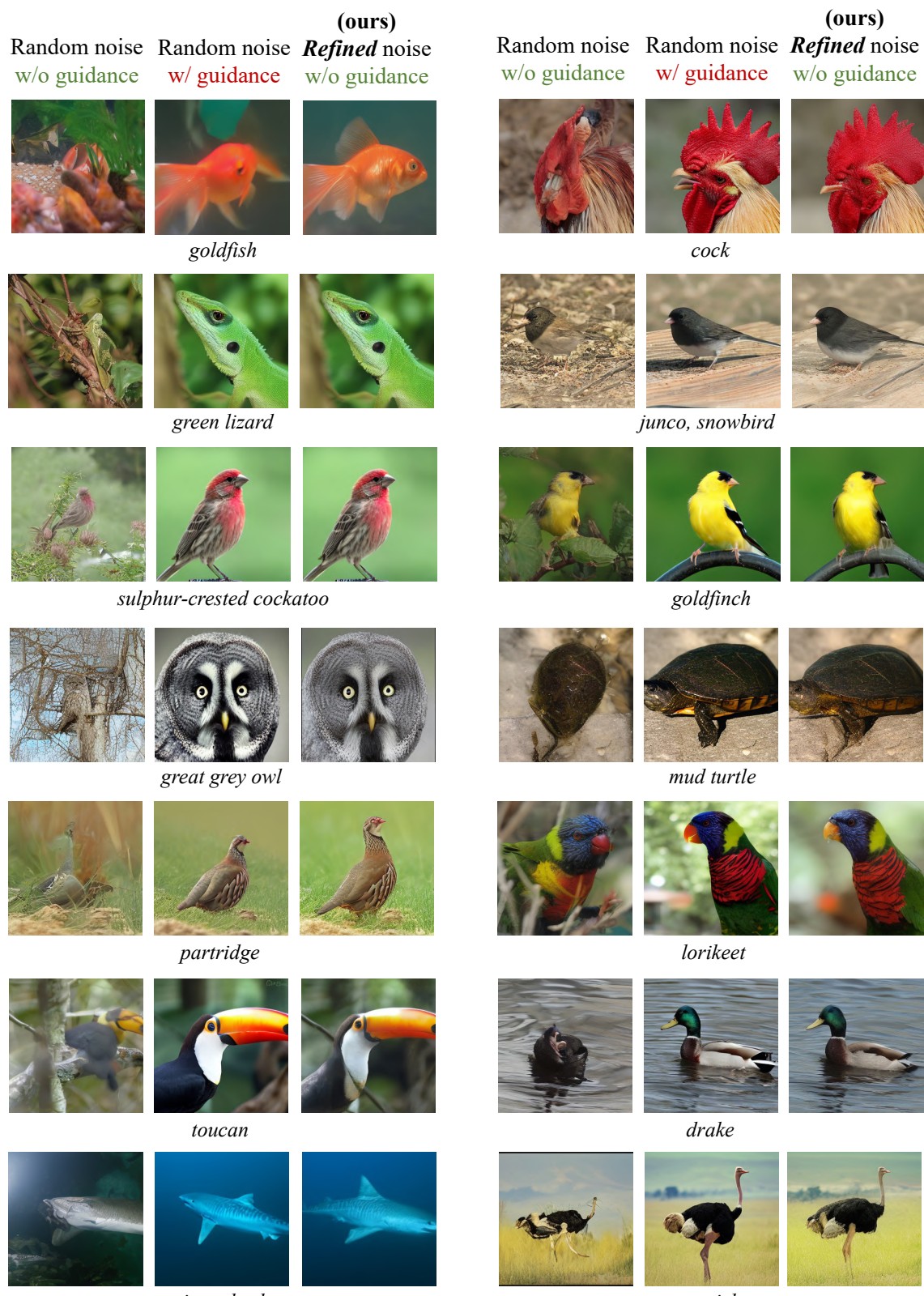

Figure 40: **Additional qualitative results on SiT-XL/2.**

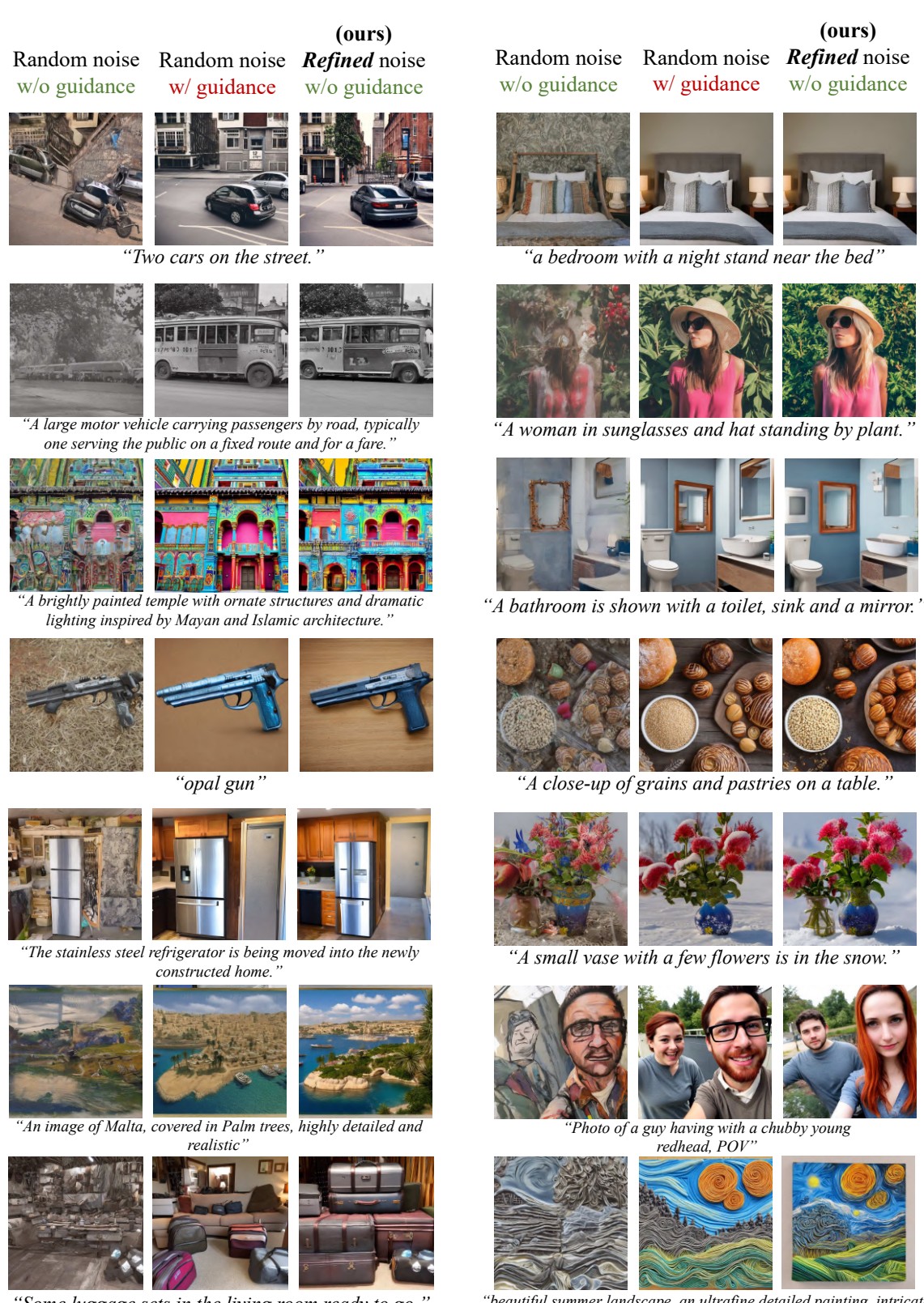

Figure 41: **Additional qualitative results on SD2.1.**

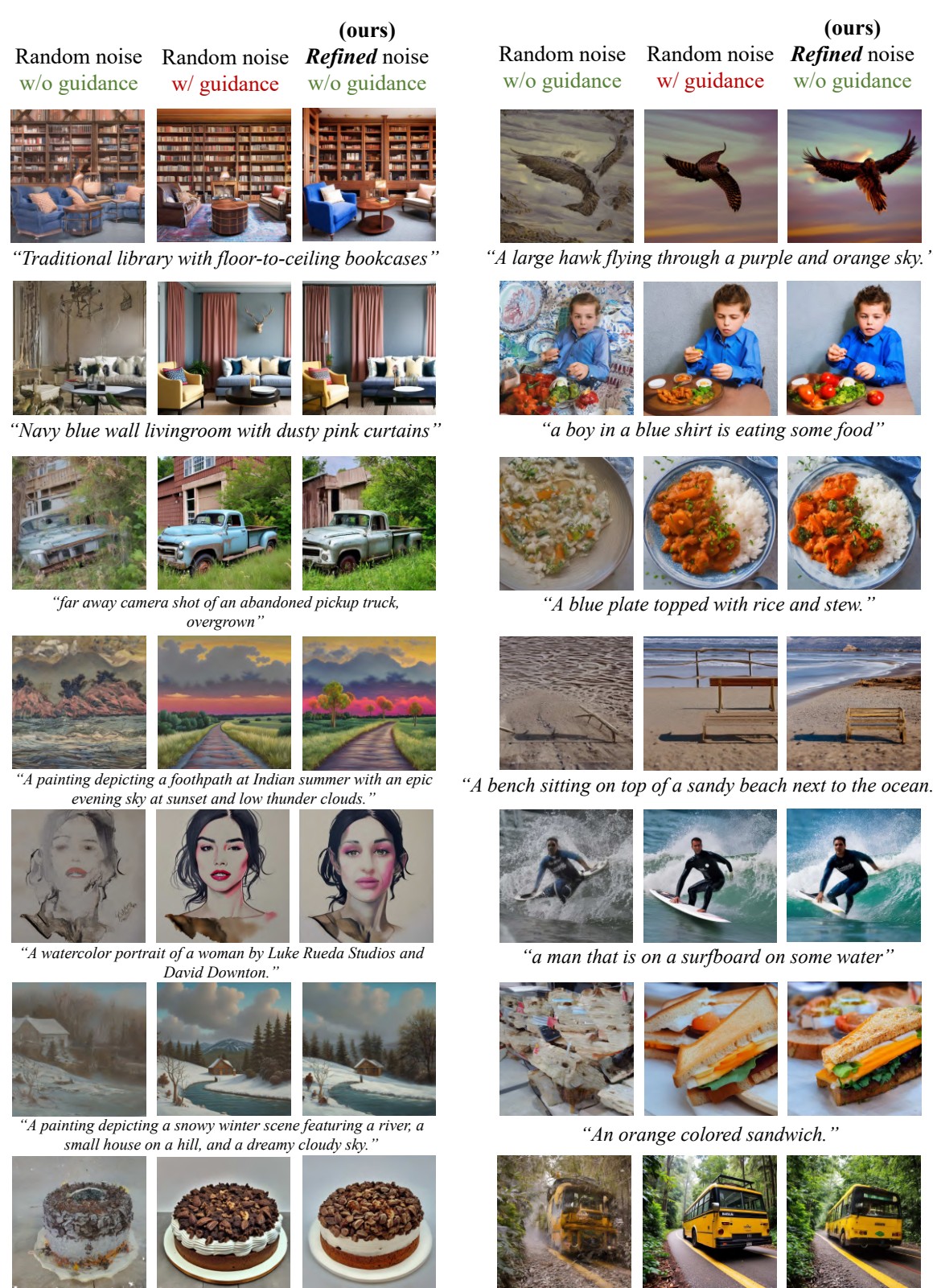

Figure 42: **Additional qualitative results on SD2.1.**

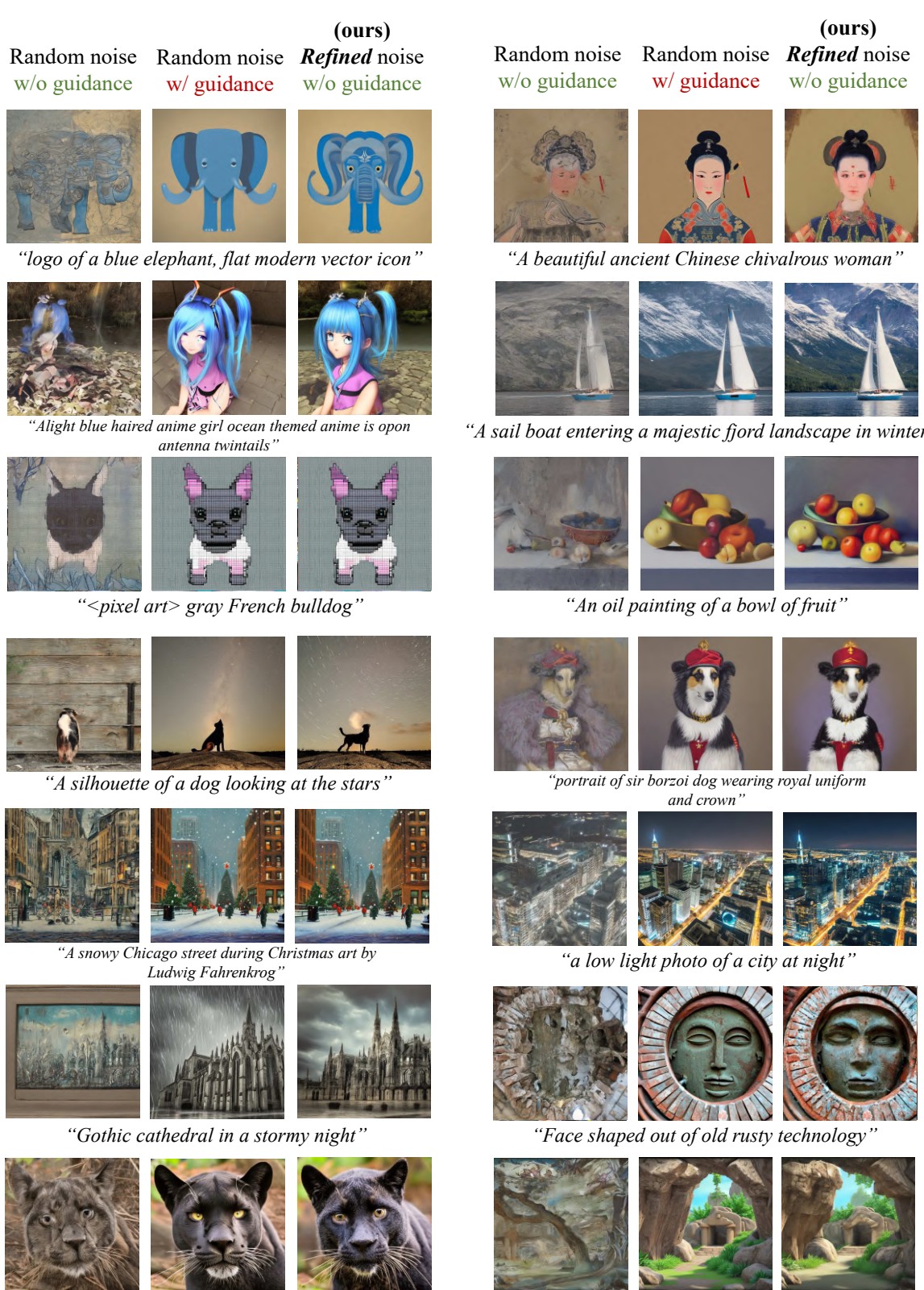

Figure 43: **Additional qualitative results on SD2.1.**

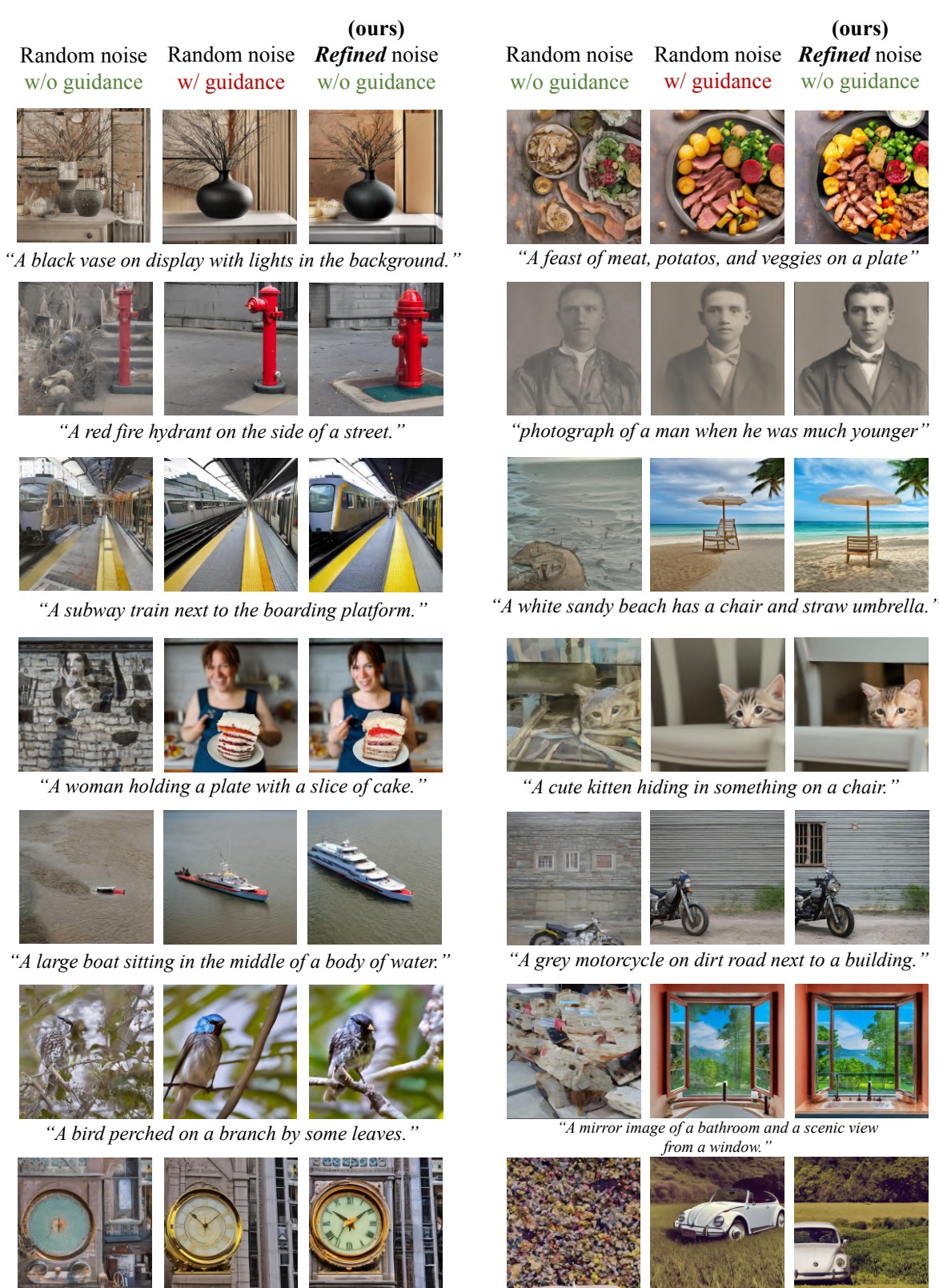

Figure 44: **Additional qualitative results on SD2.1.**

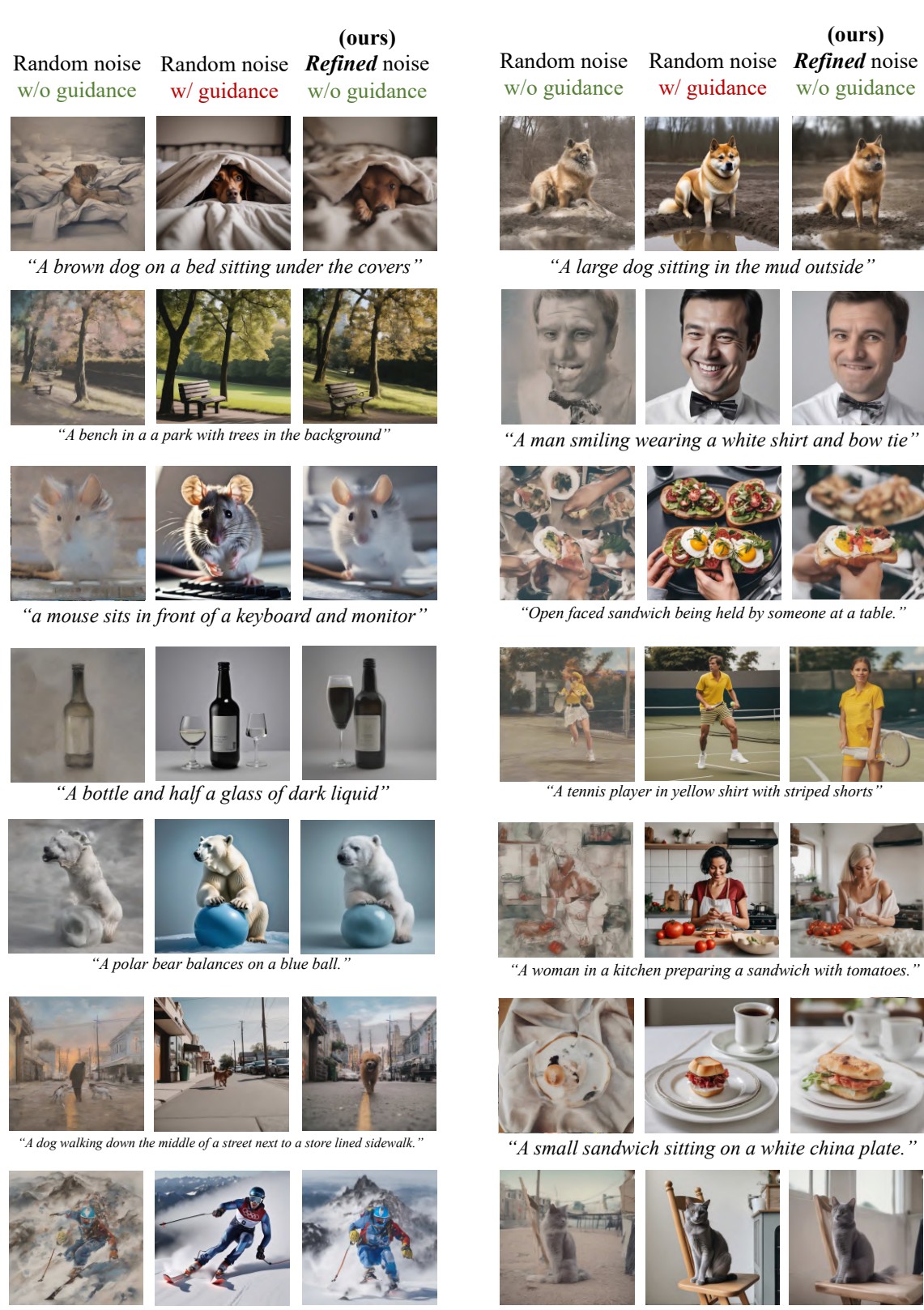

Figure 45: **Additional qualitative results on SDXL.**

## F  LLM USAGE DISCLOSURE

During the preparation of this paper, the authors made limited use of large language models (LLMs) for polishing the writing, grammar refinement and LaTeX formatting. LLMs were not used for generating research ideas, designing or conducting experiments, analyzing results, or formulating conclusions. All scientific content and contributions are entirely the responsibility of the authors, and any LLM-assisted text was carefully reviewed and revised before inclusion.

