# OpenReview forum: "A Noise is Worth Diffusion Guidance"
_ICLR.cc/2026/Conference — ICLR 2026 Poster_

### Official Review · Reviewer_AzoV · 2025-10-25

**Soundness:** 1
**Presentation:** 2
**Contribution:** 2
**Rating:** 2
**Confidence:** 4

**Summary:**

This paper introduces NoiseRefine, a new method to eliminate the computational overhead of classifier-free guidance (CFG) during diffusion model inference. Instead of applying guidance at each sampling step, NoiseRefine employs a refining network to perform a one-time transformation on the initial noise vector. This network, created by fine-tuning a pre-trained diffusion model with LoRA, maps random noise to a "guidance noise". By using this refined noise as the starting point, an off-the-shelf diffusion model can generate high-quality, guided images without the need for iterative guidance, effectively halving the inference cost.

**Strengths:**

The paper introduces a new approach to guidance distillation. The core idea of refining the initial noise vector instead of performing iterative guidance is interesting. The work is well-motivated by the clear and practical goal of reducing the inference cost of classifier-free guidance, a significant bottleneck in modern diffusion models.

**Weaknesses:**

Though the idea of adopting noise refinement in distilling guidance is interesting, the manuscript in its current form has several major issues related to contradictory motivation, methodological clarity, and experimental evaluation that weaken its conclusions. My primary concerns are detailed below.

### Contradictory Motivation and Methodological Justification.
The introduction motivates the method by showing that "guidance noise" (inverted from a high-quality guided image) can be successfully mapped back to a similar high-quality image by the diffusion model (Figure 2). This suggests that such a guidance noise target is stable and learnable. However, in Section 3.2, the authors argue against directly learning this mapping. They claim that inversion errors make this approach infeasible, leading to low-quality reconstructions (Figure 4). This directly contradicts the initial motivation. If the inversion error is as severe as Figure 4 suggests, the premise illustrated in Figure 2, which underpins the entire approach, is undermined. Could the authors please clarify this discrepancy?

### Ambiguity in the Proposed Loss Function and Training.
The discussion around the loss functions ($L_{denoise}$ v.s. $L_{MSD}$) is confusing and raises questions.
**a. Training Cost**: The paper claims that training with the full-gradient loss $L_{denoise}$ is significantly more expensive than with the proposed score-matching approximation $L_{MSD}$. Could the authors elaborate on this? Since the diffusion model's weights are frozen during the refiner's training, the primary cost should be the forward/backward passes of the refining network itself. It's not immediately clear why the diffusion model incurs expensive training cost.
**b. Approximation Outperforming the Target**: In Figure 6, the proposed $L_{MSD}$ loss not only converges to a lower value but also produces visually superior results compared to the full-gradient objective. This is counter-intuitive. If $L_{MSD}$ is a theoretical approximation, why does it empirically outperform the objective it is meant to approximate? This suggests either the approximation is not just an approximation but introduces a beneficial regularization, or there is an issue with the implementation or interpretation of the $L_{denoise}$ objective.

### Experimental Evaluation Weaknesses
1. Missing cost analysis in the main paper.
The paper is motivated by reducing the high cost of guidance sampling and claims the commonly adopted guidance distillation is computationally expensive, yet it fails to demonstrate the efficiency gain or provide a quantitative comparison of its own training costs against related methods. Please provide a direct comparison of the training cost (e.g., in GPU hours) of NoiseRefine against the key guidance distillation baselines mentioned.  The inference process is not clearly described. Is the noise refined once before the sampling process begins, or at every step? The primary contribution is claimed to be efficiency, so a detailed breakdown of the inference speed-up (e.g., wall-clock time, FLOPs) compared to the standard CFG baseline is essential.
2.  Under-reported and puzzling baseline results.
**a. CFG baseline**: The CFG performance for SiT-XL/2 in Table 2 is substantially worse than what is reported in the original paper. This weakens the claimed improvements of NoiseRefine.
**b. Approximation surpassing full method**: It is highly unexpected that NoiseRefine, which is trained to approximate CFG, would achieve better FID scores than full CFG itself. This result is counter-intuitive and may point to a suboptimal implementation of the CFG baseline or an unfair comparison setup.
3. Incomplete baselines and flawed user study.
**a. Missing baselines**: For conditional image generation, the comparison is incomplete. Methods like auto-guidance, which also aim to reduce guidance costs with a lightweight model, are highly relevant and should be included as a baseline.
**b. User study design**: The user study is flawed. It compares images generated without any guidance to images generated with NoiseRefine. A meaningful comparison must be between the standard guidance method (CFG) and the proposed method (NoiseRefine) to evaluate if the approximation preserves perceptual quality.
4. Questionable application and generalizability.
**a. Experiments on SD-Turbo**: The experiments on SD-Turbo are conceptually questionable. NoiseRefine is designed to alleviate the cost of iterative guidance. In a one-step model like SD-Turbo, there is no iterative cost to reduce. Instead, NoiseRefine adds the overhead of an extra network pass, making it inherently less efficient than the native one-step model. Furthermore, could the authors explain the surprising result in Table 4 where 2-step sampling yields a worse FID than 1-step sampling?
**b. Generalizability**: The claim of superior generalizability in Figure 8 is supported only by limited qualitative examples. To make this claim convincing, quantitative metrics evaluating performance on out-of-distribution prompts or styles are needed.

While the core idea of refining initial noise is interesting, the manuscript requires a major revision. Addressing the concerns regarding baseline performance, cost analysis, and experimental design is critical to substantiating the contributions of this work.

**Questions:**

All my concerns and questions are listed above.

---

> ### Author Response · Authors · 2025-11-26
>
> **Dear reviewer AzoV,**
>
> We sincerely appreciate the reviewer’s careful reading of our manuscript and the constructive feedback. Below, we provide responses to each point.
>
> ## **W1: Motivation and methodological justification**
>
> ### **[Comment W1] The motivation for using inversion noise appears inconsistent with the later claim that inversion errors make this approach unreliable**
>
> > The introduction motivates the method by showing that "guidance noise" … can be successfully mapped back to a similar high-quality image by the diffusion model (Figure 2). … However, in Section 3.2, the authors argue against directly learning this mapping. They claim that inversion errors make this approach infeasible, leading to low-quality reconstructions (Figure 4). This directly contradicts the initial motivation. … Could the authors please clarify this discrepancy?
> >
>
> ### **[Response]**
>
> Thank you for pointing this out. We appreciate the opportunity to clarify this potential source of confusion.
>
> Our intention in the introduction was to present a *conceptual* motivation: **if** a perfect inversion method existed, then a “ideal guidance-free noise target” would also exist, and a model could directly learn to map an initial Gaussian noise to this ideal target. Figure 2(a) illustrates this *idealized thought experiment*.
>
> However, as we discuss in Section 3.2 and as shown in Figure 4, real inversion methods [1] are imperfect and introduce non-negligible errors. These errors make direct learning of the inversion-based noise target infeasible and lead to degraded reconstructions. This is why, despite the conceptual appeal of noise-target learning, the practical results (Fig. 5) reveal its limitations.
>
> To address this, our method ultimately follows the pathway in Figure 2(b), which regresses directly to the guided image and avoids reliance on inversion quality. This design choice is entirely consistent with the empirical observations in Section 3.2.
>
> We acknowledge that the current writing in the introduction may have unintentionally implied that Figure 2(a) reflects an achievable real-world pipeline. We have revised the introduction to explicitly state that Figure 2(a) serves only as an idealized motivation, while the practical method aligns with Figure 2(b), which performs robustly without requiring a perfect inversion.
>
> [1] Denoising Diffusion Implicit Models, Jiaming Song, Chenlin Meng, Stefano Ermon

---

> ### Author Response · Authors · 2025-11-26
>
> ## **W2. Ambiguity of proposed loss function and training**
>
> ### **[Comment W2] a. Why is** $L_\text{Denoise}$ **more expensive than** $L_\text{MSD}$?
>
> ### **[Response]**
>
> It is true that the diffusion model’s weights are frozen. However, in the training framework (Fig. 2(b)), producing the unguided image from the refined noise requires running $N$ denoising steps (the bottom branch in Fig. 2(b)). Although the forward pass itself is identical to standard sampling, the key difference is that, under the full-gradient objective, **gradients must be propagated through the entire denoising trajectory**, not just through the refining network. This leads to substantial additional computation.
>
> Specifically, this introduces two sources of overhead:
>
> 1. **Backpropagation through all** $N$ **denoising steps:**
>
>     Even though the diffusion model’s weights are frozen, the loss $L_\text{Denoise}$ still requires computing gradients through every denoising step in the trajectory. This means running the backward pass $N$ times through the denoising network, making the full-gradient objective **X times slower** than our approach, which stops gradients at each denoising step and avoids propagating through the diffusion model.
>
> 2. **Memory for storing or recomputing activations:**
>
>     The forward pass requires retaining the intermediate states of the denoising trajectory. While gradient checkpointing can reduce VRAM usage, it introduces additional compute overhead.
>
>
> In contrast, $L_\text{MSD}$ avoids backpropagating through the $N$-step denoising process entirely, resulting in **noticeably lower computational and memory cost** during training.
>
> ---
>
> ### **[Comment W2] b. Why can the approximation outperform the full objective?**
>
> ### **[Response]**
>
> Thank you for pointing this out. We agree that it may seem counter-intuitive that the approximation outperforms the full-gradient objective. In principle, the original formulation—with gradients propagated through the entire denoising trajectory—should be superior.
>
> However, in practice, the full-gradient objective suffers from **optimization instability**. Because the loss must backpropagate through multiple consecutive denoising steps (e.g., 10 steps), it effectively constructs a deep recurrent computation graph over the same denoising network. Such repeated Jacobian multiplications are highly susceptible to gradient explosion or vanishing, a phenomenon analogous to the well-known instability in training recurrent networks during long-horizon backpropagation. As a result, the optimization becomes noisy and difficult to stabilize, leading to suboptimal convergence. During the rebuttal period, we analyzed the gradients under the full-graident objective and observed clear explosion and vanishing patterns, as shown in Fig. 34 in Appendix B.3.
>
> To address this optimization instability, our approximation skips the gradients through the denoising steps, following the insight introduced by Score Distillation Sampling (SDS) [2]. SDS also omits the denoising network Jacobian term in its objective and explicitly reports that doing so not only avoids the prohibitive computational cost but also yields an *effective* and more stable gradient signal:
>
> > “In practice, the U-Net Jacobian term is expensive to compute… We found that omitting the U-Net Jacobian term leads to an effective gradient for optimizing DIPs with diffusion models.” — [2]
> >
>
> Our empirical observations mirror this behavior. We hypothesize that approximating the U-Net Jacobian as an identity matrix implicitly regularizes the training signal by avoiding unstable multi-step Jacobian products, resulting in better convergence—as reflected in Fig. 6 and Fig. 33. This aligns with the reviwer’s suggestion that the approximation introduces a beneficial regularization effect.
>
> We have added this clarification to the main paper, and we provide an additional analysis of the regularization effect that arises from avoiding long-horizon unstable backpropagation in Appendix Sec. B.3, “Diffusion model Jacobian approximation.”
>
> [2] DreamFusion: Text-to-3D using 2D Diffusion, Ben Poole, Ajay Jain, Jonathan T. Barron, Ben Mildenhall

---

> ### Author Response · Authors · 2025-11-26
>
> ## **W3. Experimental evaluation**
>
> ### **[Comment W3-1] a. Direct comparison with guidance distillation is needed**
>
> > “The paper is motivated by reducing the high cost of guidance sampling and claims the commonly adopted guidance distillation is computationally expensive, yet it fails to demonstrate the efficiency gain or provide a quantitative comparison …  provide a direct comparison of the training cost … against the key guidance distillation baselines mentioned.”
> >
>
> ### **[Response]**
>
> **Clarification of claim.**
>
> Our paper is *not* intended to claim that NoiseRefine provides lower training cost compared to existing guidance distillation methods. As stated in our contributions (Lines 115–124), our primary goal is to demonstrate that **guidance information can be distilled into the initial noise itself**, and to provide a thorough analysis of this phenomenon. The paper’s main contribution lies in revealing and characterizing this alternative form of guidance distillation, rather than proposing a computationally cheaper training procedure.
>
> We agree that the phrase in the introduction (“making training costly…”) could be misinterpreted as a claim that existing guidance distillation has higher per-model training cost. This was not our intention, and we have removed the phrase to avoid confusion.
>
> **Comparison to guidance distillation baselines.**
>
> Since our work does not claim superior training efficiency, we believe that a direct cost comparison is not crucial for evaluating the scientific contribution. However, we acknowledge the reviewer’s concern and include a comparison of the two methods’ performance over training steps in Fig. 26 in Appendix A.8. While the noise refining network converges more slowly, the results show that it consistently improves FID throughout training. Although our method requires relatively high training cost, this overhead arises naturally from optimizing the initial noise, as it requires operating on the full diffusion pipeline rather than a single step. As discussed above, our contribution is to demonstrate that this form of guidance distillation is feasible and effective despite the inherent cost structure of noise-based optimization.
>
> ---
>
> ### **[Comment W3-1] b. Clarification on the inference process**
>
> > Is the noise refined once before the sampling process begins, or at every step?
> >
>
> ### **[Response]**
>
> **Inference procedure.**
>
> The noise is refined **once** before the denoising process begins. This single refinement step is the core idea of our “noise refinement”: $g_\phi$ transforms the initial Gaussian noise into a refined noise, and the subsequent denoising is performed entirely without guidance and without any further refinement.
>
> This behavior is described at several points in the manuscript (Lines 89–98, 116, 222), but for clarity we have now added an explicit description of the inference process at Line 226 in the manuscript. We appreciate the reviewer’s comment, which helped us clarify this point in the manuscript.
>
> **Inference speed comparison.**
>
> A detailed wall-clock comparison is provided in Appendix A.6 (Table 6), showing that our method achieves substantially faster inference than the CFG baseline while using the same number of denoising steps. Since refinement requires only a single forward pass of a lightweight module, the overhead is negligible, and the overall computational cost is close to that of unguided sampling.
>
> For the reviewer’s convenience, we also include the numbers directly in the rebuttal (copied from Appendix A.6).
>
> **Table: Quantitative comparison of image quality and computational cost.** 30K prompts from MS-COCO (Lin et al., 2014) validation dataset were used for evaluation.
> | **Model** | **Initial Noise** | **Guidance** | **Inference Time ↓** |
> |----------|-------------------|--------------|-----------------------|
> | **SD2.1** | Gaussian | ✗ | 1.357s |
> |          | Refined | ✗  | 1.504s |
> |          | Gaussian | ✓ | 2.589s |
> | **SDXL** | Gaussian | ✗ | 3.218s |
> |          | Refined | ✗ | 3.323s |
> |          | Gaussian  | ✓ | 5.525s |

---

> ### Author Response · Authors · 2025-11-26
>
> ### **[Comment W3-2] a. CFG baseline appears worse than in the original paper**
>
> ### **[Response]**
>
> **Clarification on sampling settings.**
>
> We understand the reviewer’s concern. The discrepancy in FID mainly arises from **(i) differences in samplers and denoising steps** and **(ii) differences in guidance scale**.
>
> Specifically, SiT [3] uses the Euler–Maruyama sampler with **250 denoising steps**, following the DiT evaluation protocol. Euler–Maruyama is an SDE solver and is known to produce higher-quality samples and lower FID scores. In contrast, due to our limited GPU budget, we use the Euler sampler (an ODE solver) with only 20 denoising steps for faster evaluation, which generally results in higher FID.
>
> Additionally, our evaluation uses a 100-class subset of ImageNet, consistent with the training setup of the noise refining network. For completeness, all training and evaluation details are documented in Appendix D.2.
>
> For full transparency, we will release the exact code used to generate samples and compute FID.
>
> **Guidance scale.**
>
> We use a guidance scale of $w = 4.0$, whereas the original SiT paper only reports FID at $w =1.5$. This is because FID is known to misalign with human perceptual quality, and many diffusion/flow-matching papers (including SiT) use **different guidance scales for FID reporting and for qualitative samples** (e.g., FID at $w =1.5$, visuals at $w =4.0$).
>
> Since our aim is not to outperform CFG in FID but to produce high-quality images, we follow the perceptual-quality setting used in SiT and train our model with ($w = 4.0$). **Because the original paper does not provide FID at this guidance scale, we compute FID-50K at ($w = 4.0$) ourselves** and report it in Table 3. All comparisons use the **same initial noises, same class subset, and identical seeds** to ensure fairness.
>
> [3] SiT: Exploring Flow and Diffusion-based Generative Models with Scalable Interpolant Transformers, Nanye Ma, Mark Goldstein, Michael S. Albergo, Nicholas M. Boffi, Eric Vanden-Eijnden, Saining Xie
>
> ---
>
> ### **[Comment W3-2] b. Approximation surpassing the guided sampling in FID**
>
> > It is highly unexpected that NoiseRefine, which is trained to approximate CFG, would achieve better FID scores than full CFG itself. This result is counter-intuitive and may point to a suboptimal implementation of the CFG baseline or an unfair comparison setup.
> >
>
> ### **[Response]**
>
> We clarified above that the discrepancy may originates from **different evaluation settings** and **the well-known limitations of FID** as a metric.
>
> While the teacher model (CFG) is the target for distillation, it is well documented that distilled diffusion models can outperform their teacher in FID and IS. This behavior is visible in the reported results of prior work, such as Table 1 in [4] and Table 2 in [5], where the student model achieves lower FID than the teacher under the same guidance scale.
>
> [4] On Distillation of Guided Diffusion Models, Chenlin Meng, Robin Rombach, Ruiqi Gao, Diederik P. Kingma, Stefano Ermon, Jonathan Ho, Tim Salimans
>
> [5] Plug-and-Play Diffusion Distillation, Yi-Ting Hsiao, Siavash Khodadadeh, Kevin Duarte, Wei-An Lin, Hui Qu, Mingi Kwon, Ratheesh Kalarot

---

> ### Author Response · Authors · 2025-11-26
>
> ### **[Comment W3-3] a. AutoGuidance should be included as a relevant baseline**
>
> > For conditional image generation, the comparison is incomplete. Methods like auto-guidance, which also aim to reduce guidance costs with a lightweight model, are highly relevant and should be included as a baseline.
> >
>
> ### **[Response]**
>
> We appreciate the reviewer’s comment. However, Autoguidance is not an appropriate baseline in our setting. Autoguidance is itself **a sampling-time guidance mechanism**, not a method designed to reduce guidance cost. Importantly, its guidance signal can also be distilled by our framework, just like any other guidance method. Therefore, Autoguidance does not serve as a meaningful baseline for evaluating our approach, whose primary goal is to show that *any* guidance mechanism can be distilled into the initial noise.
>
> The Autoguidance paper introduces three types of “bad” models for generating guidance signals. Among these, **only** the variant that uses a smaller model provides any inference-time cost reduction. The other two variants—using an under-trained model with the same architecture or using a different EMA version of the main model—have **the same computational cost as CFG**. Even in the small-model case, Autoguidance still requires **two forward passes per denoising step**, making it more expensive than unguided sampling. Moreover, Autoguidance requires loading an **additional diffusion model** into GPU memory, which is more memory-intensive than CFG (which uses a single shared model for conditional and unconditional branches). For these reasons, Autoguidance is generally not used as a practical cost-reduction technique; its efficiency benefit applies only under a specific configuration.
>
> Most importantly, Autoguidance is simply **one particular form of guidance**, and our approach is general enough to distill *any* such mechanism into the initial noise. In our main experiments, we demonstrate that Perturbed-Attention Guidance can be successfully distilled by our noise-refining network on both Stable Diffusion 2.1 and SDXL (Tab. 2 and Fig. 8). Likewise, recent work such as *Align Your Flow* [6] also distills Autoguidance signals into few-step models, reinforcing that Autoguidance is a *guidance source* rather than a cost-reduction baseline.
>
> [6] Align Your Flow: Scaling Continuous-Time Flow Map Distillation Amirmojtaba Sabour, Sanja Fidler, Karsten Kreis
>
> ---
>
> ### **[Comment W3-3] b. User study should compare CFG vs. NoiseRefine**
>
> ### **[Response]**
>
> Thank you for pointing this out. We agree that comparing NoiseRefine directly with standard guidance (CFG) is a more appropriate way to assess whether the approximation preserves perceptual quality.
>
> Actually, this comparison is included in Appendix E.2. Our original intent in the main text was to illustrate that a **single noise refinement step** can already make unguided samples perceptually much more favorable, which is why that version of the user study was highlighted.
>
> However, we agree that the CFG comparison is more meaningful in this context. We have therefore moved the **NoiseRefine vs. Guided Sampling** user study from Appendix into the main paper. The results show that refined-noise samples are **equally preferred** to guided samples, indicating that NoiseRefine preserves perceptual quality.

---

> ### Author Response · Authors · 2025-11-26
>
> ### **[Comment W3-4] a. Conceptual concerns regarding the SD-Turbo experiment**
>
> > The experiments on SD-Turbo are conceptually questionable. NoiseRefine is designed to alleviate the cost of iterative guidance. In a one-step model like SD-Turbo, there is no iterative cost to reduce. Instead, NoiseRefine adds the overhead of an extra network pass, making it inherently less efficient than the native one-step model. Furthermore, could the authors explain the surprising result in Table 4 where 2-step sampling yields a worse FID than 1-step sampling?
> >
>
> ### **[Response]**
>
> **Clarification of the goal of the SD-Turbo experiments.**
>
> Thank you for raising this point. We agree that SD-Turbo [6] does not involve iterative guidance, and therefore, NoiseRefine is not intended to provide any cost reduction in this setting. That is not the goal of our experiment.
>
> Our purpose in evaluating SD-Turbo is to show that the role of initial noise remains meaningful even in few-step or one-step generative models, where iterative sampling is absent. We make this point in the main paper: “*Compared to generation starting from Gaussian noise, our approach improves structural coherence and overall quality, highlighting the role of structured initial noise even in few-step models.”*
> Initial noise may be even more important in these models because there are fewer opportunities to correct errors during sampling.
>
> In addition, we find that a noise-refining network trained on multi-step diffusion models can still improve perceptual quality in SD-Turbo, even though it is not trained for SD-Turbo. For this reason, the SD-Turbo experiment is not presented as an efficiency benchmark. Instead, it demonstrates that the phenomenon we study—namely, that guidance information can be encoded in the initial noise—also holds in few-step models, where the refined noise serves as a strong starting point for generation.
>
> **On the observation that 1-step FID is better than 2-step FID.**
>
> As noted above, **FID is not aligned with perceptual quality**, and this behavior is well documented in prior works. In fact, the original SD-Turbo paper [6] does **not** report FID for 2-step or 4-step sampling at all. They provide only qualitative results for these settings, and Appendix B explicitly states:
>
> > “For the evaluation results presented in Figures 5 to 7, we employ human evaluation and do not rely on commonly used metrics for quality assessment of generative models such as FID and CLIP-score, since these have been shown to capture more fine-grained aspects like aesthetics and scene composition only insufficiently.” — [7]
> >
>
> For this reason, we additionally report **human-aligned preference metrics**, including PickScore and HPSv2. These metrics show that **2-step sampling is slightly preferred over 1-step sampling**, which matches the visual quality observed in our qualitative results.
>
> [7] Adversarial Diffusion Distillation, Axel Sauer, Dominik Lorenz, Andreas Blattmann, Robin Rombach
>
> ---
>
> ### **[Comment W3-4] Limited evidence for generalizability**
>
> > The claim of superior generalizability in Figure 8 is supported only by limited qualitative examples. To make this claim convincing, quantitative metrics evaluating performance on out-of-distribution prompts or styles are needed.
> >
>
> ### **[Response]**
>
> **Adding quantitative results.**
>
> Thank you for pointing this out. To strengthen the claim in Figure 8, we added quantitative human-preference metrics for the two domains shown there. Specifically, as described in Table 4, we curated ten prompts per domain, expanded each prompt using an LLM (GPT-5.1) to obtain 1000 total prompts, and generated multiple samples using random noise paired with these prompts. The results show that, even without any additional training for each domain, our refined noise yields performance comparable to guided sampling on domain-specific fine-tuned models. This supports the generalizability of our approach beyond the examples shown in Figure 8.
>
> ---
>
> We truly appreciate the reviewer’s constructive feedback, which helped us strengthen the paper. We did our best to address all concerns raised by the reviewer. If you have any further questions or remaining concerns, please feel free to let us know. We would be happy to clarify and further improve the work.

---

> ### Comment · Reviewer_AzoV · 2025-11-26
>
> Thank you for the detailed replies. Your clarifications have addressed many of my initial questions. However, I have a few remaining concerns regarding the paper's presentation and experiments that I believe need to be addressed to strengthen the work.
>
> ### The optimization process for the noise refining network remains unclear.
>
> Since the gradients from the pre-trained diffusion model are detached, the training mechanism for the refiner is non-trivial. I suggest provide a brief but formal introduction to Score Distillation Sampling (SDS), which you cite as inspiration. This will provide the necessary context for readers to better understand the functioning of training objective. For example, give a clear description of how the gradients w.r.t $\phi$ is computed and what approximations have been made.
>
> Moreover, In appendix, Proposition 2, the proof is difficult to follow due to omitted intermediate steps.
>
> ### Experiment
>
> - Regarding W.1
>
> A large reversion error could be a direct consequence of small number of reverse sampling steps (e.g. 50 steps used in Figure 5). As discussed in DDIM, Table2, the reconstruction error is 0.0023 with 50 sampling steps, while further reducing to 0.0001 with 500 sampling steps. However, this issue, which is key to the motivation of the proposed method, is not further discussed in the main paper. For example, is it possible to learn a noise-to-noise mapping when utilizing more sampling steps during the reversion process?
>
> - Regarding W3-2.a
>
> For a method in this field to be considered a valid contribution, **It's important to provide a more straightforward, fair, and **apple-to-apple** comparison with baseline methods thus clearly positions the method's contribution within the literature.** However, current Table 2 in the main paper is insufficient to achieve this. While FID has known drawbacks, if it is used as a metric for comparison, the comparison must be fair. The argument to de-emphasize metrics does not justify an unfair comparison.
>
> While I understand the authors may have limited computational resources, it's feasible and widely accepted to evaluate the method on smaller benchmarks like CIFAR10 and ImageNet-64. As an example, Progressive Distillation[1] was evaluated on CIFAR10 and another guidance distillation method [2], highly relevant to your work, reported results on ImageNet 64x64.
>
> - W3.2: Approximation surpassing the guided sampling in FID
>
> The claim that the proposed approximation surpasses the original guided sampling in terms of FID requires a more rigorous explanation. Attributing this phenomenon to "the flaw of FID" is an insufficient explanation. Besides, the authors suggest this discrepancy could result from different sampling settings. To make this argument credible, I would suggest the authors explicitly display the sampling settings used for the baseline methods in the main paper.
>
> Minor to the updated submission:
> 1. To facilitate the review process, I recommend highlight all modifications in the updated submission.
> 2. The vertical spacing between the caption and the main text in lines 79-80 should be increased.
>
> [1] Salimans, Tim, and Jonathan Ho. "Progressive distillation for fast sampling of diffusion models." arXiv preprint arXiv:2202.00512 (2022).
> [2] Meng, Chenlin, et al. "On distillation of guided diffusion models." Proceedings of the IEEE/CVF conference on computer vision and pattern recognition. 2023.

---

> ### Author Response · Authors · 2025-12-03
>
> ### **[Comment] Precise inversion using more inversion steps**
>
> ### **[Response]**
>
> Thank you for raising this important point. We agree that reconstruction errors in DDIM inversion can be heavily influenced by the number of reverse sampling steps, as also discussed in Table 2 of DDIM. To directly address this concern, we conducted an additional set of experiments (now reported in Appendix A.11), following the evaluation protocol of DDIM. Specifically, we varied the number of DDIM inversion steps and measured reconstruction quality under 10–1000 steps.
>
> Our results show that simply increasing the number of inversion steps does not continually reduce the reconstruction error. As detailed in Fig. 29 and Tab. 11, performance stabilizes within a moderate range (typically 50–200 steps), and further increasing the number of steps does not lead to substantially improved reconstructions. This trend is consistent with the observations in ReNoise where Fig. 8 similarly shows that additional DDIM inversion steps do not necessarily yield better fidelity.
>
> These findings suggest that the reconstruction inaccuracies we study do not stem solely from an insufficient number of inversion steps. Rather, they reflect the inherent approximation nature of DDIM inversion itself. Even when using a large number of steps, fine-grained details cannot be perfectly recovered, indicating fundamental limitations in deterministic inversions of diffusion trajectories. Our appendix analysis was added precisely to clarify this point and strengthen the motivation behind the proposed approach.
>
> ---
>
> ###  **[Comment] Experiments under the same setting as the original SiT paper**
>
> ###  **[Response]**
>
> We note that our choice of $w = 4.0$ reflects its superior perceptual quality, which is also the setting used for qualitative results in the original SiT paper. In contrast, $w = 1.5$ is rarely used in practice.
>
> However, following the reviewer’s suggestion, we conducted new experiments using the original SiT quantitative settings: a 2nd-order ODE solver, 250 NFEs, and guidance scale $w = 1.5$. We report qualitative results in Fig. 30 and quantitative results in Tab. 13 (Appendix A.13).
>
> Due to limited time and GPU budget during the rebuttal period, we trained the models for approximately 60% of the total steps used in Tab. 2. Nonetheless, the results show a clear improvement over the unguided baseline and maintain a reasonable gap to CFG, consistent with reviewer 86wZ’s earlier observation.
>
> We also clarified the sampler type and denoising steps in the main paper and provided a reference to the corresponding experiments (Appendix A.13) that folllow the original paper’s settings.
>
> ---
>
> ### **[Comment] Readability of MSD Details and Derivation**
>
> ###  **[Response]**
>
> To improve clarity and accessibility, we revised Appendix B.2 by adding a **Warm-up** subsection. This section formally defines the full denoising trajectory, introduces consistent notation, and clarifies how the terms relate to one another. By providing this structured buildup, we aim to make the logic and approximations in Proposition 2 easier to follow. We hope this updated presentation will help readers better understand the derivation and motivation behind Multistep Score Distillation (MSD).
>
> ---
>
> ### **[Comment] Highlighting of edits and caption spacing**
>
> ###  **[Response]**
>
> Thank you for your helpful suggestions. As you recommended, we highlighted all additions and changes in green to make them easier to identify. We also adjusted the vertical spacing between the caption and the main text. We sincerely appreciate your constructive feedback, which helped us improve the clarity and presentation of the paper.

---

### Official Review · Reviewer_a4W6 · 2025-10-27

**Soundness:** 3
**Presentation:** 4
**Contribution:** 3
**Rating:** 8
**Confidence:** 3

**Summary:**

This paper proposes _NoiseRefine_, a guidance distillation method that learns to refine the initial Gaussian noise into one that can generate high-quality images without classifier-free guidance. When performing inversion _without_ guidance on an image generated _with_ guidance, the resulting "inversion noise" is generally non-Gaussian, yet it can still recreate the image without guidance. Based on this observation, the authors propose training a lightweight refinement network—a LoRA of the pretrained diffusion model—to learn the mapping between the original Gaussian noise and the inversion noise. This effectively distills guidance into the noise.

To achieve this, the method introduces three main contributions:
- **Image-space supervision**: Training is supervised in image space, which is shown to be more robust to inversion errors.
- **Residual connection**: Instead of learning the full mapping, the refinement network is parameterized to learn the residual.
- **Backpropagation**: Similar to score distillation sampling (SDS), the gradient is not propagated through the network during training. This has been shown to lead to faster convergence and higher-quality results.

To validate the approach, the paper ablates the method on several conditional image diffusion models and thoroughly analyzes what the refinement network learns, which seems to be primarily adding low-frequency structures to the noise. Additionally, the authors compare their method with other guidance distillation approaches and argue that distilling guidance into the noise is more efficient and generalizes better to other fine-tuned models.

**Strengths:**

The paper tackles an important topic in the diffusion community, which is related to guided generation. The main strength of this paper is in the idea it proposes. To the best of my knowledge, this is the first work to look into guidance distillation into the noise input, which seems like a simple yet brilliant idea.

The execution of the method is also in itself very well done: the authors validate each contribution that make the method work using small experiments. The manuscript is easy to read and the method clearly explained.

Furthermore, the appendix is very exhaustive and provides many additional details and experiments that makes the submission very strong.

**Weaknesses:**

No major weaknesses, the appendix addresses many of my concerns already.

**Questions:**

Again, not many questions, as the exhaustive appendix addressed most of my interrogations already.

- The user study is purely done against no guidance in the main paper, and against with guidance in the appendix. Have the authors tried to compare the method with other guidance distillation methods?

---

> ### Author Response · Authors · 2025-11-27
>
> **Thank you for the positive and encouraging assessment. We appreciate your comments on the novelty of distilling guidance into the noise input, the clarity of the method, and the thoroughness of our analysis and appendix.**
>
> Although our experiments provide quantitative comparisons with other guidance distillation methods for reference, our primary goal is to understand whether guidance can be distilled into the noise itself and to evaluate how closely such a model can approach the performance of guided sampling. In line with this objective, our user study was designed to focus on assessing how well noise-level distillation can reproduce the behavior of guided sampling.
>
> If you have any additional questions or suggestions, please feel free to let us know—**we are always happy to further improve the work.**

---

### Official Review · Reviewer_ScWm · 2025-10-30

**Soundness:** 3
**Presentation:** 3
**Contribution:** 3
**Rating:** 4
**Confidence:** 4

**Summary:**

This paper propose a novel and orthogonal approach that "distills" the effect of guidance into the initial noise vector rather than into the diffusion model itself. The core idea is to train a lightweight "noise refining network" (implemented as a parameter-efficient LoRA on the diffusion model's UNet) that maps a standard Gaussian noise vector ($x_T$) to a "refined" noise vector ($\hat{x}_T$). Specifically, this paper propose Multistep Score Distillation (MSD), a technique that stops the gradient during backpropagation through the denoising network at each step.

**Strengths:**

1. This idea is novel and plug-and-play.
2. It's orthogonal to other acceleration methods.

**Weaknesses:**

1. I'd like to know if optimizing Eq4 is difficult and costly, since it requires go through the network many times.
2. The result after refinement was still not good enough, significantly worse than the result generated by CFG.
3. For me, it's difficult to understand why noise-to-noise mapping has generalization properties. The paper lacks discussion and analysis on this aspect. For example, are the generated results of training samples has better quality than validation samples? For refined noise, can it be combined with CFG to obtain better results? If so, how does it compare to the results obtained using CFG with initial noise?

**Questions:**

Can we directly learn the mapping from the initial XT to X0_guide using a bridge model initialized from the original diffusion network? For example, DDBM or I2ISB. What are the advantages of MSD compared to this approach?

---

> ### Author Response · Authors · 2025-11-27
>
> **Dear reviewer ScWm,**
>
> We sincerely thank the reviewer for the clear and constructive feedback. Your comments greatly helped us refine the presentation and strengthen the discussion of the work. Below, we respond to each point in detail.
>
> ---
>
> ### **[Comment W1] Training cost of Eq. 4**
>
> > I'd like to know if optimizing Eq4 is difficult and costly, since it requires go through the network many times.
> >
>
> ### **[Response]**
>
> **Cost of training.**
>
> Eq. 4 requires $N$ denoising steps, so optimizing it indeed needs $N$ forward passes through the diffusion model. This cost is inherent to all noise-optimization methods, since the refining network must operate on the input of the entire denoising trajectory rather than a single step. Prior works [1,2] also incur this cost and therefore many of them are restrict themselves to few-step models ([2])
>
> However, our method avoids the main bottleneck: **we skip the gradients through the denoising network**. As a result, we do *not* perform the expensive $N$-step backpropagation that a full-gradient objective would require. This makes our training  more efficient compared to methods that must propagate gradients through the entire diffusion pipeline.
>
> We provide a detailed analysis of training cost in Tab. 8 of Appendix A.7.
> Furthermore, during the rebuttal period, we added the training-time FID curve of NoiseRefine (Fig. 26). We also include a comparison with a representative guidance-distillation baseline [3]. Although the noise-refining network converges somewhat more slowly, it reaches a comparable FID to guidance distillation at around 2K steps and continues to improve afterward.
>
> [1] Optimizing Diffusion Noise Can Serve As Universal Motion Priors, Korrawe Karunratanakul, Konpat Preechakul, Emre Aksan, Thabo Beeler, Supasorn Suwajanakorn, Siyu Tang
>
> [2] ReNO: Enhancing One-step Text-to-Image Models through Reward-based Noise Optimization, Luca Eyring, Shyamgopal Karthik, Karsten Roth, Alexey Dosovitskiy, Zeynep Akata
>
> [3] On Distillation of Guided Diffusion Models, Chenlin Meng, Robin Rombach, Ruiqi Gao, Diederik P. Kingma, Stefano Ermon, Jonathan Ho, Tim Salimans
>
> ---
>
> ### **[Comment W2] Insufficient quality compared to CFG**
>
> > The result after refinement was still not good enough, significantly worse than the result generated by CFG.
> >
>
> ### **[Response]**
>
> We would greatly appreciate it if you could clarify which aspect appeared significantly worse, as this would help us understand your concern more precisely.
>
> Based on the quantitative results in Tab. 3 and the qualitative examples in Fig. 7, the refined results are slightly below guided sampling but remain broadly comparable. We also include extensive additional examples in Appendix Fig. 33–38, which show consistent trends across diverse prompts.
>
> Also, we would like to clarify that our goal is not to match or compete with guided sampling in terms of ultimate generation quality. Rather, the main contribution of our work is to demonstrate that guidance information can be distilled into the initial noise, enabling diffusion sampling without any inference-time guidance (or guidance-distilled denoising steps) to produce plausible results. Our intention is to show that such guidance-equivalent noise refinement is feasible, and to provide a detailed analysis of *why* it works—such as the strong role of low-frequency components, the behavior of cross-attention maps, and other findings presented in Fig. 9–12 (Sec. 5) and Appendix A.1–A.2. We hope the reviewers find this perspective useful and that our analytical contributions are appreciated.

---

> ### Author Response · Authors · 2025-11-27
>
> ### **[Comment W3] Why is noise-to-noise generalization possible?**
>
> > For me, it's difficult to understand why noise-to-noise mapping has generalization properties. The paper lacks discussion and analysis on this aspect. For example, are the generated results of training samples has better quality than validation samples? For refined noise, can it be combined with CFG to obtain better results (CFG + refined)? If so, how does it compare to the results obtained using CFG with initial noise (CFG  Gaussian)?
> >
>
> ### **[Response]**
>
> Thank you for the thoughtful questions. We clarify the generalization behavior and the synergy between refined noise and CFG below.
>
> **Generalization mechanism of the noise refining network.**
>
> The mapping learned by the refining network is not a pure noise-to-noise transformation; it is closer to a **noise-to-layout** mapping. A diffusion model gradually forms coarse spatial structures from the initial noise during the early denoising steps, meaning that the noise-to-image mapping is inherently learnable and generalizes even though the model cannot observe every possible noise sample during training.
>
> Analogously, the refining network observes the initial noise and learns to adjust it so that the *unguided* denoising trajectory produces an image that matches the corresponding *guided* sample. Through training, the refining network discovers which refined noise patterns reliably lead to guided-like outcomes. Our analyses (Fig. 10 in the main paper and Fig. 20 in the appendix) show that the refining network consistently injects **low-frequency structural information**, effectively shaping the coarse layout determined by the original Gaussian noise—similar to the role of inversion noise (Fig. 3 and Fig. 9).
>
> As a result, the refining network learns a **generalizable mapping between Gaussian noise and a guidance-informed initial layout**, rather than memorizing pixel-wise noise patterns. We provide a more detailed explanation of this mechanism in Appendix A.2 (“Why does refined noise help denoising?”).
>
> **Can guided sampling further improve refined noise?**
>
> **T**his is a very interesting question. When applying CFG on top of refined noise, we observe further improvements in structural coherence and overall visual quality compared to applying CFG on standard Gaussian noise. For example, refined-noise samples frequently remove structural artifacts such as extra limbs and produce more stable compositions (see Fig. 25 and the quantitative results in Table 6, Appendix A.5).
>
> This indicates that even under CFG, the **initial noise plays a critical role**, and that refined noise provides complementary benefits. Intuitively, because the refining network is trained across many guided samples, it learns a *generalizable* and *averaged* form of layout guidance—whereas CFG alone sometimes produces overly strong or unstable corrections. The refined noise therefore offers a better starting point for guided sampling as well.
>
> We discuss these observations in more detail in Appendix A.5 (“Guided sampling with refined noise”).

---

> ### Author Response · Authors · 2025-11-27
>
> ### **[Comment Q1]**
>
> > Can we directly learn the mapping from the initial $x_T$ to $x_0^\text{guide}$ using a bridge model initialized from the original diffusion network? For example, DDBM or I2ISB. What are the advantages of MSD compared to this approach?
> >
>
> ### **[Response]**
>
> Thank you for the insightful question. Learning a direct mapping from ($x_T$) to ($x_0^{\text{guide}}$) using a bridge model (e.g., DDBM or I2ISB) is indeed an interesting idea. If we interpret correctly, fixing one end of the bridge to Gaussian noise ($x_T$) effectively reduces the formulation to training a standard diffusion model—i.e., learning a score model that maps noise to data. In that case, this would amount to **fine-tuning the original diffusion model using guided samples** as supervision. If this was not the reviewer’s intention, we would be happy to receive clarification.
>
> This direction is conceptually intriguing, especially since—as far as we know—prior work has not explored training diffusion models directly on guided samples. If guided samples sufficiently represent the distribution induced by guidance, this approach might behave similarly to guidance distillation, since both rely on matching the guided-data distribution. We explored this direction experimentally and report the results in Appendix A.12. The findings show that this approach performs slightly worse than guidance distillation
>
> More importantly, this approach is not aligned with our goal of preserving the denoising pipeline by modifying only its input. A bridge-model formulation requires updating the entire diffusion model, affecting all denoising steps and risking catastrophic forgetting. In contrast, our method freezes the pretrained model, modifies only the initial noise, and remains fully compatible with domain-finetuned backbones, few-step samplers, and other variants—without retraining.
>
> The core insight of our method is that **guidance information can be distilled into the initial noise without modifying the diffusion pipeline**. If one allows modifying the denoising network, then guidance distillation already provides a more direct and stable alternative than retraining via a bridge model.
>
> For these reasons, while direct bridge-model training is conceptually appealing, __**NoiseRefine**__ offers an interesting model-preserving alternative that targets the initial noise rather than the full denoising network.
>
> ---
>
> Thank you again for the insightful comments. We have addressed each concern to the best of our ability, and we hope the revised version better conveys both the contribution and the technical soundness of the work. If you have further suggestions, we would be grateful to hear them.

---

### Official Review · Reviewer_86wZ · 2025-10-30

**Soundness:** 4
**Presentation:** 4
**Contribution:** 3
**Rating:** 6
**Confidence:** 4

**Summary:**

The paper aims to address the reliance of Diffusion models on sampling guidance (e.g., CFG), which effectively doubles inference cost. It proposes to learn a (external) noise refining network $g_\phi$ to modify the initial Gaussian noise $x_T$, for the correspondence with the effects of CFG. It also uses methods including training in image space for better performance and using Multistep Score Distillation (MSD) for efficient training, with rounded analysis and sound experiments.

Totally, this work is simple in the method but interesting in the perspective. The core idea (noise refinement for guidance) is actually like a special (but clever) variant of guidance distillation approaches, and some comparisons are unfair or not comprehensive. But it does give an insightful perspective from the initial noise, with convincing (at least for me) motivation, not bad results, and the potential for wider application in various scenarios.

Some questions need to be clarified and solved, but overall, this paper is well-written, easy to follow, rich in content, and deserves a positive rating.

**Strengths:**

1. Totally, it is well-written and easy to follow.
2. The overall motivation is clear. Optimizing the guidance to accelerate the inference of Diffusion models is useful.
3. The perspective from the initial noise is quite interesting. The idea of distilling guidance information into the initial noise, rather than into the denoising network itself, is also insightful and provides a fresh perspective on tackling the guidance overhead problem (with potential and compatibility for wider application in various scenarios).
4. Sound experiments and rounded analysis make the results convincing. The method significantly outperforms the unguided baseline, with an acceptable gap to CFG.

**Weaknesses:**

1. A Special Form of Distillation: While the paper claims their method is "orthogonal" to guidance distillation, it is more accurately a clever variant rather than a new paradigm. The framework fits the knowledge distillation paradigm, the difference from guidance distillation is the locus of learning (the noise refiner $g_\phi$ v.s. the denoising network). This positioning overstates the fundamental novelty, more discussion/comparison/clarification should be made.
2. Loss of Critical Controllability: The method encodes the effect of a *fixed* guidance scale (e.g., CFG scale 7.0) into the network, thereby losing the native, zero-cost flexibility of the CFG scale $w$ at inference time. The workarounds proposed (scaling the residual or conditioning the training) are either heuristics with unverified behavior or add complexity to the training process, undermining the method's simplicity. Especially, the presented results are not intuitive to understand (Fig. 13; Fig. 23): Changing $w$ seems to have no effect on the "richness" of the image at all, but merely affects the degree of distortion, failing to reach the original function of $w$ in the CFG.
3. The Theoretical Soundness of MSD: The entire framework's viability hinges on Multistep Score Distillation (MSD) for gradient approximation. However, its theoretical justification (Proposition 2) relies on a strong assumption ($\partial\epsilon_\theta^{(t)} / \partial x_t \propto I$). This assumption is empirically/visually supported in the appendix but lacks rigorous quantitative evidence.
4. Additional Complexity and Overhead: The training process requires running the expensive guided sampling to generate the teacher signal $x_0^{Guide}$ for every training sample. As for the inference overhead, unlike standard LoRA (which can be merged into the base model's weights to eliminate inference overhead), the noise refining network $g_\phi$ implemented with LoRA cannot be merged, meaning an extra model forward pass is also always required at inference time.

**Questions:**

1. The Study/Evaluation of Controllability: The NoiseRefine is used to replace the role of CFG, but the study of the controllability seems not to be sufficient. Can the details of the training-based approach of controllability be further illustrated? Curious about how the guidance scale $w$ is incorporated as an additional input for training, and how to guarantee the range of $w$ (e.g., show the num of samples of different $w$). Also, have you done contrast experiments of different fixed $w$ of CFG during the training of NoiseRefine, and quantitatively compared the FID/CLIP score?
2. Cost and Performance Comparison with Guidance Distillation: Given that training NoiseRefine also requires running guided sampling, how does its total training budget (e.g., in GPU-hours) compare directly to training a standard guidance distillation model [1] to achieve similar performance?
3. The Architectural Choice for $g_\phi$: Is implementing $g_\phi$ as a LoRA attached to $\epsilon_\theta$ a limitation? Have you experimented with a completely separate, smaller network for $g_\phi$? How would this impact performance and the leveraging of "pretrained knowledge"?
4. The Jacobian Approximation in MSD: The proof of Proposition 2 critically depends on the assumption that $\partial\epsilon_\theta^{(t)} / \partial x_t \propto I$. Does this assumption still hold for Transformer architectures (e.g., DiT, SiT) where attention mechanisms might induce more complex, long-range Jacobian structures? Could you provide a quantitative error bound or an ablation study (e.g., detaching gradients for only a subset of steps) to more rigorously validate this approximation? This theoretical/experimental verification may not be so easy to do, so it is OK as the current results are great, just a comment. But if you can provide an intuitive understanding, or if the validity of this Jacobian Approximation can be verified, it would be really helpful to the research community.


[1] Chenlin Meng, Robin Rombach, Ruiqi Gao, Diederik Kingma, Stefano Ermon, Jonathan Ho, and Tim Salimans. On distillation of guided diffusion models. In Proceedings of the IEEE/CVF Conference on Computer Vision and Pattern Recognition, pp. 14297–14306, 2023.

---

> ### Author Response · Authors · 2025-11-27
>
> **Dear reviewer 86wZ**,
>
> We sincerely appreciate the reviewer’s thorough reading of our manuscript and the thoughtful feedback. We are grateful that you found our perspective—distilling guidance into the initial noise—interesting and potentially useful, and your comments have been very helpful in improving the clarity and positioning of the work. We address each point below.
>
> ---
>
> ### **[Comment W1] Claims about “orthogonal”**
>
> > While the paper claims their method is "orthogonal" to guidance distillation, it is more accurately a clever variant rather than a new paradigm. … This positioning overstates the fundamental novelty, more discussion/comparison/clarification should be made.
> >
>
> ### **[Response]**
>
> Thank you for pointing this out. We agree that the term “orthogonal” may overstate the distinction, and we have revised it to a more precise and neutral phrasing. In the revised manuscript, we now describe our method as *“a distinct approach by refining the initial Gaussian noise—a critical yet under-explored component in diffusion-based generation pipelines.”*
>
> ---
>
> ### **[Comment W2/Q1] Loss of controllability or added complexity**
>
> > **[W2]** The method encodes the effect of a *fixed* guidance scale … thereby losing the native, zero-cost flexibility of the CFG scale at inference time. The workarounds proposed … are either heuristics … or add complexity to the training process, undermining the method's simplicity.
> >
>
> > **[Q1]** The NoiseRefine is used to replace the role of CFG, but the study of the controllability seems not to be sufficient. Can the details of the training-based approach of controllability be further illustrated? …
> >
>
> ### **[Response]**
>
> Since users typically rely on a well-established “best” guidance scale for each model in real-world settings, we focused on a fixed scale for clarity. That said, we fully agree that controllability is an important aspect, and expanding the discussion indeed strengthens our contribution. We provide further clarifications below.
>
> **Training-based controllability preserves the simplicity of our method.**
>
> We agree that the training-free variant of controllability may appear heuristic. In contrast, the training-based approach is more principled and directly aligns with how guidance distillation is commonly implemented. Importantly, it does **not** add meaningful complexity to our method.
>
> Following prior guidance distillation work [1] and its recent adoption in Flux.1-dev [2], we condition on the guidance scale using the **existing timestep embedding**, rather than introducing a new conditioning mechanism. This requires only a **single linear projection layer** to map the guidance scale embedding to the same space as the timestep embedding. Thus, the architecture change is minimal and the overall training pipeline remains nearly identical.
>
> We have added this explanation to the main paper for clarity.
>
> **Details of the training-based approach.**
>
> Concretely, we compute the sinusoidal positional embedding of the guidance scale, project it through a linear layer, and add the result to the timestep embedding. This introduces only one lightweight module. For the sampling range, we uniformly draw ( $w \in [2.0, 9.0]$ ) when training with Stable Diffusion 2.1.
>
> While we have not yet completed the comparison experiments for different fixed guidance scales, the controllability behavior is already reflected in our guidance-scale conditioning results (Fig. 24). We plan to report the fixed-$w$ experiments as well if time allows before the discussion period ends.
>
> **Effect of controlling** $w$**.**
>
> The behavior we observe when controlling the guidance scale in our method is similar to controlling the CFG scale. To make this clear, we added CFG results directly in Fig. 23. As shown, CFG exhibits the same trend where increasing $w$ primarily changes the *degree of distortion* rather than changing the “richness” effects.
>
> We believe this is because Stable Diffusion 2.1 generally requires **higher CFG scales** to exhibit strong oversaturation or simplification effects. If we extend the sampling interval to include much larger guidance scale, we expect to observe the typical oversaturated and overly simplified results associated with very high guidance scales.
>
> However, such extreme guidance scales are rarely used in practice, so we did not include them in our training range. We will explore an expanded guidance-scale range in follow-up experiments as time permits.
> ****
>
> [1] On Distillation of Guided Diffusion Models Chenlin Meng, Robin Rombach, Ruiqi Gao, Diederik P. Kingma, Stefano Ermon, Jonathan Ho, Tim Salimans
>
> [2] Flux.1-dev. https://bfl.ai/blog/24-08-01-bfl

---

> ### Author Response · Authors · 2025-11-27
>
> ### **[Comment W3/Q4] Insufficient evidence for the key assumption in MSD**
>
> > The Theoretical Soundness of MSD: The entire framework's viability hinges on Multistep Score Distillation (MSD) for gradient approximation. However, its theoretical justification (Proposition 2) relies on a strong assumption. This assumption is empirically/visually supported in the appendix but lacks rigorous quantitative evidence.
> >
>
> > The Jacobian Approximation in MSD: The proof of Proposition 2 critically depends on the assumption that ${\partial \epsilon_\theta^{(t)}}/{\partial x_t} \propto I$. Does this assumption still hold for Transformer architectures (e.g., DiT, SiT)… ? Could you provide a quantitative error bound or an ablation study … to more rigorously validate this approximation?
> >
>
> ### **[Response]**
>
> Thank you for raising this important point. We agree that the validity of the Multistep Score Distillation (MSD) approximation is central to the framework, and we have strengthened both the empirical and theoretical discussion as follows.
>
> **Quantitative evaluation of the Jacobian approximation.**
>
> We agree that visualization alone is insufficient to assess how strongly the identity-matrix assumption holds. In the revised appendix, we include a quantitative analysis of the Jacobian matrix. Specifically, we report the distributions of diagonal and off-diagonal entries across timesteps:
>
> - **Off-diagonal values** lie almost entirely within ($[0, 1\times10^{-2}]$).
> - **Diagonal values** primarily lie within ($[1\times10^{-1}, 1]$).
>
> These results (Fig. 30, Stable Diffusion 2.1) show that the Jacobian is highly diagonally dominant in practice, supporting the assumption used in Proposition 2.
>
> **Validity for Transformer-based architectures (e.g., DiT, SiT).**
>
> To examine whether attention mechanisms yield more complex Jacobian structures, we repeat the same analysis for DiT-XL/2 (Fig. 31, Appendix B.3). Interestingly, we observe a *clearer separation* between diagonal and off-diagonal entries than in Stable Diffusion:
>
> - Diagonal elements are close to 1.
> - Off-diagonal elements remain near 0.
>
> This suggests that the identity approximation not only holds for U-Net architectures but also extends robustly to Transformer-based diffusion models.
>
> **Intuitive understanding of the benefit of the Jacobian approximation.**
>
> Our decision to skip the U-Net Jacobian is inspired by Score Distillation Sampling (SDS) [3], in which the Jacobian term is intentionally omitted:
>
> > “In practice, the U-Net Jacobian term is expensive to compute… We found that omitting the U-Net Jacobian term leads to an effective gradient for optimizing DIPs with diffusion models.” — [3]
> >
>
> While our objective is different from SDS, we observe a closely related phenomenon. Backpropagating full gradients through multiple denoising steps effectively creates a long recurrent computation graph, which leads to significant optimization instability—much like long-horizon backpropagation in recurrent networks.
>
> During the rebuttal period, we extended the experiment in Fig. 6 to directly analyze gradient norms (Fig. 33). We found that gradients remain far more stable when the identity approximation is applied (i.e., when gradients through the denoiser are skipped). Further, we computed the gradient norm of the loss with respect to each denoising step and observed that, under full Jacobian backpropagation, the norms become significantly larger or smaller as we move toward earlier steps in the trajectory—that is, clear instances of gradient explosion and vanishing (Fig. 34).
>
> Taken together, these results show that the identity approximation (i.e., gradient skipping) consistently yields a much more stable and reliable optimization signal, helping training proceed smoothly despite the long multi-step computation chain.
>
> Taken together, our findings show that:
>
> 1. **The Jacobian is sufficiently close to an identity matrix in practice**, supported by quantitative evidence across both U-Net and Transformer architectures.
> 2. **It is not exactly identity**, and the small off-diagonal components accumulate during multi-step backpropagation, leading to instability.
> 3. **Skipping Jacobians provides both a practical approximation and a form of regularization** that mitigates long-horizon gradient amplification.
>
> We have included this expanded discussion and experimental evidence in Appendix B.3. We hope this clarifies both the validity and the practical advantages of the MSD approximation.
>
> [3] DreamFusion: Text-to-3D using 2D Diffusion Ben Poole, Ajay Jain, Jonathan T. Barron, Ben Mildenhall

---

> ### Author Response · Authors · 2025-11-27
>
> ### **[Comment W4/Q1]  Additional complexity and overhead**
>
> > **[W4]** Additional Complexity and Overhead: The training process requires running the expensive guided sampling to generate the teacher signal for every training sample. …
> >
>
> > **[Q1]** Cost and Performance Comparison with Guidance Distillation: Given that training NoiseRefine also requires running guided sampling, how does its total training budget (e.g., in GPU-hours) compare directly to training a standard guidance distillation model to achieve similar performance?
> >
>
> ### **[Response]**
>
> **Additional complexity and comparison with guidance distillation.**
>
> Since our work does not claim superior training efficiency to guidance distillation, we believe that a direct cost comparison is not crucial for evaluating the scientific contribution. However, we acknowledge the reviewer’s concern and include a comparison of the two methods’ performance over training steps in Fig. 26 in Appendix A.8. While the noise refining network converges more slowly, the results show that it consistently improves FID throughout training. Although our method requires relatively high training cost, this overhead arises naturally from optimizing the initial noise, as it requires operating on the full diffusion pipeline rather than a single step (but we made it more efficient using MSD).
>
> Because the two models were trained under different hardware environments, we report step-wise FID curves instead of wall-clock comparisons. As shown in the Fig. 26, the noise-refining network converges somewhat more slowly, but it reaches the same FID as guidance distillation at around 2K steps.
>
> Our method applies the loss **after denoising for (N) steps**, whereas standard guidance distillation computes its loss **after a single step**. Thus, in principle, the GPU cost per training step differs by roughly a factor of (N). Consequently, after approximately ($2N$K) effective updates (we used ($N=10$)), the two methods achieve comparable FID.
>
> ---
>
> ### **[Comment Q3] Separate network for $g_\theta$**
>
> > The Architectural Choice for $g_\theta$: Is implementing $g_\theta$ as a LoRA attached to a limitation? Have you experimented with a completely separate, smaller network for $g_\theta$ ? How would this impact performance and the leveraging of "pretrained knowledge"?
> >
>
> ### **[Response]**
>
> Thank you for the question. We evaluated this choice in Appendix C.1. When we train a separate UNet-shaped ($g_\theta$) from scratch (no LoRA, no pretrained weights), it performs better than unguided sampling (FID 42.71 → 37.87) but remain far worse than using a LoRA on the pretrained backbone (FID 13.74). This indicates that leveraging pretrained knowledge is crucial.
>
> We also tried training a smaller standalone refining network early in the experiments. It could refine noise for a *fixed* prompt, but it failed to generalize across prompts, likely because different prompts require very different spatial layouts—something difficult to learn from scratch. In contrast, a LoRA adapter can effectively reuse the pretrained model’s semantic and spatial priors.
>
> In summary, our method does not *strictly* require pretrained knowledge, and we did observe that a standalone refiner can work to some extent in controlled settings (e.g., for a fixed prompt). However, for prompt-dependent noise refinement in large-scale diffusion models, pretrained priors are practically essential. A LoRA-based refiner is simply a much more effective and parameter-efficient way to utilize these priors. We therefore do not view this as a limitation, but rather as a pragmatic design choice: LoRA provides strong generalization at minimal cost, and there is no compelling reason to avoid using it.
>
> ---
>
> We are truly grateful for the reviewer’s careful evaluation and constructive suggestions, which have helped us refine the work. We believe the revisions have substantially improved the clarity and presentation. If you have any further thoughts or suggestions, we would be delighted to continue the discussion.

---

### Author Response · Authors · 2025-12-03
**Final Remark (4/4)**

### **Regarding training costs**

- **Additional complexity of guided sampling and quantitative comparison with guidance distillation [1] (86wZ, ScWm):** We clarified that our goal is not to outperform guidance distillation in training efficiency but to study the effectiveness of refining the noise itself. Following the reviewers’ suggestion, we added a comparison of the two methods over training steps in Fig. 26 (Appendix A.8). While the noise-refining network converges more slowly, the results show that our method reaches the same FID as guidance distillation at around 2K steps and continues to improve thereafter. We also noted that the full-step denoising overhead naturally arises when optimizing the initial noise, since it requires operating on the full diffusion pipeline rather than a single step, but this overhead is significantly reduced through MSD.

## **4. Conclusion**

We believe these updates resolve all major concerns and substantially improve the clarity and reliability of the manuscript. By offering a new perspective on guidance distillation, we hope our work contributes to a deeper understanding of how initial noise shapes generation and encourages further research on learning to modify initial noise.

Thank you for your time and service to the community, especially under the unusual circumstances of this review cycle.

Sincerely,
The Authors

[1] On Distillation of Guided Diffusion Models Chenlin Meng, Robin Rombach, Ruiqi Gao, Diederik P. Kingma, Stefano Ermon, Jonathan Ho, Tim Salimans

---

### Author Response · Authors · 2025-12-03
**Final Remark (3/4)**

### **Regarding evaluation**

- **AutoGuidance should be included as a relevant baseline  (AzoV):** We clarified that AutoGuidance is a sampling-guidance method rather than a separate baseline, and it can be distilled within our framework, similar to the PAG distillation shown in our main experiments.
- **User study should compare CFG vs. NoiseRefine (AzoV):** We clarified that this comparison is included in the Appendix and have moved it into the main manuscript.
- **CFG baseline appears worse than in the original SiT paper (AzoV):**

    We clarified that this gap mainly arises from (i) different samplers and inference settings—our setup uses a 1st-order sampler with 20 NFEs, whereas the original SiT paper used a 2nd-order sampler with 250 NFEs—and (ii) the use of a 100-class ImageNet subset due to GPU constraints. Additionally, the guidance scale differs: we use $w = 4.0$ (preferred for perceptual quality), while the original paper used $w = 1.5$.

    However, following the reviewer’s suggestion, we ran new experiments using the original SiT settings (2nd-order sampler, 250 NFEs, $w = 1.5$), and reported the results in Tab. 13 (Appendix A.13). Due to limited time and GPU budget during the rebuttal period, we trained the models to about 60% of the total steps used in Tab. 2. Still, the results show a clear performance gain over the unguided baseline and a reasonable gap to CFG, as reviewer 86wZ noted. We also note that $w = 1.5$ is rarely used in practice, since $w = 4.0$ is typically preferred due to its superior perceptual quality.

- **Limited evidence for generalizability (AzoV):** To support the claim that the noise refining network is compatible with finetuned diffusion models without additional training, we added quantitative human-preference metrics for two domains in Tab. 4. The results show that, even without domain-specific training, our refined noise achieves performance comparable to guided sampling on fine-tuned models.

### **Regarding MSD**

- **Insufficient evidence supporting the assumption (86wZ):** The reviewer pointed out that the assumption used in the theoretical justification (Proposition 2) is visually supported but lacks rigorous quantitative evidence, and also questioned whether it holds for DiT architectures. We reported the quantitative ranges of the diagonal and off-diagonal entries of the Jacobian, showing that the Jacobian is close to the identity matrix for both Stable Diffusion (Fig. 30) and DiT (Fig. 31).
- **Unexpected cases where the approximation outperforms the full objective (86wZ, AzoV):** The reviewer raised the question of why the approximation could outperform the full objective (AzoV). We provided an explanation using an analogy to the training instability of recurrent neural networks, where gradients may explode or vanish. We also added experiments showing that the gradient norm is significantly more unstable than MSD, demonstrating that gradient exploding/vanishing indeed occurs when backpropagating through multiple denoising steps (Fig. 33–34). In their follow-up response, reviewer AzoV acknowledged that this concern was addressed.
- **Readability of MSD Details and Derivation (AzoV):** To improve clarity and accessibility, we revised Appendix B.2 by adding a **Warm-up** subsection. This section formally defines the full denoising trajectory, introduces consistent notation, and clarifies how the terms relate to one another. By providing this structured buildup, we aim to make the logic and approximations in Proposition 2 easier to follow. We hope this updated presentation will help readers better understand the derivation and motivation behind Multistep Score Distillation (MSD).

### **Regarding sample quality**

- **Insufficient quality compared to CFG (ScWm):** We asked for clarification on what is meant by “significantly worse quality,” since our results show that the refined noise “**significantly outperforms the unguided baseline, with an acceptable gap to CFG**” (reviewer 86wZ). This is supported by Tab. 3, Fig. 7, and Appendix Fig. 33–38, where refined results remain broadly comparable to guided sampling. We also clarified that our goal is not to match CFG in absolute quality. Instead, our contribution is to show that guidance information can be distilled into the initial noise and to analyze *why* this is possible—e.g., through low-frequency structure, cross-attention behavior, and other findings in Fig. 9–12 and Appendix A.1–A.2.
- **Effect of controlling** $w$ **seems different from CFG (86wZ):** We added a row of samples using CFG with the same guidance scale in Fig. 23 and showed that the effect of controlling $w$ exhibits very similar behavior to CFG sampling in Stable Diffusion 2.1.

---

### Author Response · Authors · 2025-12-03
**Final Remark (2/4)**

## **3. Concerns, questions, and our responses**

To provide context on the overall reception, reviewer **a4W6** (rating: 8) was highly satisfied and noted that all concerns were fully addressed in the extensive appendix. Reviewer **86wZ** (rating: 6) also regarded the work as insightful and promising, requesting only minor clarifications.

By contrast, reviewer **ScWm** (rating: 4) raised several concerns in a brief review, and reviewer **AzoV** (rating: 2) initially held a negative view but, after our clarifications, acknowledged that most issues had been resolved, with only a few remaining questions. We summarize these concerns below.

Overall, we believe we have addressed all major concerns and significantly improved the clarity and reliability of the manuscript. In particular, we clarified the conceptual and evaluation-setting misunderstandings raised by the lowest-scoring reviewer (AzoV), who acknowledged that our “clarifications have addressed many of the reviewer’s initial questions.” We also ensured a fair comparison with SiT by adding results under the original paper’s settings.

### **Questions about the method and clarification**

- **Motivation and methodological justification (AzoV):** The reviewer questioned that our initial motivation—learning a mapping between the initial noise and the inversion noise from guided samples—appears contradictory. We clarified that this motivation assumes perfect inversion, which does not hold in practice. This is why we compute the loss in image space rather than noise space, and to avoid confusion, we have made this assumption more explicit in the introduction. The reviewer acknowledged our explanation.
- **Noise refinement in SD-Turbo is conceptually questionable (AzoV):** The reviewer argued that noise refinement seems unnecessary in SD-Turbo because there is no iterative cost to reduce. We clarified that the goal is to show that initial noise still matters in few-step and one-step models, and may even be more important due to limited opportunities to correct errors. Our refined noise provides a stronger starting point for generation, even without training on these models. The reviewer acknowledged this clarification.
- **Is implementing** ($g_\theta$) **as a LoRA a limitation, and how does a separate** ($g_\theta$) **perform? (86wZ):** We clarified that Appendix C.1 includes results from training ($g_\theta$) from scratch, which performs better than unguided generation but worse than LoRA. We also noted that an early standalone refiner worked for a fixed prompt and failed to generalize across prompts, whereas LoRA effectively leverages pretrained semantic (text understanding) and spatial priors. Thus, while standalone training is possible to some extent within our framework, LoRA offers far superior generalization at minimal cost and is therefore a practical design choice rather than a limitation.
- **Question about why noise-to-noise generalization is possible (ScWm):** We clarified that our framework is better understood as a noise-to-layout mapping, analogous to the noise-to-image mapping learned by diffusion models. Just as diffusion models learn a generalizable mapping from noise to images, and our refining network similarly learns to map noise to a coarse layout (low-frequency structure). We refer the reviewer to Appendix A.2 for an extensive analysis of what the refining network learns.
- **Losing simplicity of the method for controllability (86wZ):** We clarified that the proposed architecture for controlling the guidance scale $w$ does not introduce substantial complexity. In practice, we simply reuse the timestep embedding and add a single linear projection layer to match dimensions, a widely adopted design in prior guidance distillation work [1].
- **Clarification on the inference process (AzoV):** We added an explicit description of the inference process at Line 226 in the manuscript
- **Question about whether guided sampling can further improve refined noise (ScWm):** We clarified that Appendix A.5 (“Guided sampling with refined noise”) provides the corresponding results.
- **Question about directly learn the mapping from the initial** $x_T$ **to** $x_0^\text{guide}$ **using a bridge model (ScWm):** We clarified that this approach effectively requires finetuning the entire diffusion model on guided samples, which updates all parameters, alters every denoising step, and risks catastrophic forgetting. In contrast, MSD freezes the pretrained model and modifies only the initial noise, preserving full compatibility with finetuned models, few-step models, and different samplers without retraining. This model-preserving property is essential to our goal of showing that guidance can be distilled into the initial noise itself.

---

### Author Response · Authors · 2025-12-03
**Final Remark (1/4)**

**Dear Area Chairs and Senior Area Chairs,**

We appreciate your time and effort, especially under the recent changes to the review process.

Below, we provide:

1. a brief summary of our contribution,
2. the strengths highlighted by the reviewers, and
3. the main concerns along with how our rebuttal and revision address them.

---

## **1. Summarization of our work**

We tackle the high inference cost of diffusion models caused by sampling guidance methods (e.g., classifier-free guidance), which often **double the computation**, by learning **better initial noise** that can produce comparable results—without requiring guidance at sampling time.

To do this, we train a **noise refining network** that maps pure Gaussian noise to a refined version that mimics the effect of guidance. Our approach includes three key techniques:

1. Using **similarity loss in image space** (instead of noise space) for more effective supervision.
2. A new **Multi-Step Score Distillation (MSD)** method for efficient training.
3. Applying **LoRA** to the refiner to improve performance and training efficiency by leveraging prior knowledge.

Experiments show that the refined noise **outperforms unguided sampling** and achieves **comparable quality to guided sampling**, with much lower cost. We also provide **in-depth analysis** of how the refined noise influences denoising trajectories—an aspect that reviewer a4W6, 86wZ noted as particularly valuable for the community.

---

## **2. Strengths noted by reviewers**

- **Novel and insightful idea — distilling guidance into noise (all reviewers).** Reviewers consistently highlighted the **novelty** of our core idea: distilling sampling guidance into the **initial noise**, rather than modifying the denoising network. This provides a fresh perspective on reducing guidance overhead in diffusion models. This conceptual shift, supported by thorough experiments and analysis, forms a distinct and meaningful contribution.
- **Strong Experimental Validation (86wZ, a4W6):** Experiments clearly validate each component of the method and show consistent improvements. These results demonstrate that **refined noise significantly outperforms unguided sampling**, with an **acceptable gap to CFG**, as noted by reviewer 86wZ.
- **Thorough Analysis and Insights (86wZ, a4W6):** Reviewers appreciated the depth and clarity of our analysis, including empirical justification, ablations, and interpretability-focused visualizations.

---

### Meta-Review · Area_Chair_N5FV · 2026-01-04

**Summary:**

**Summary**: This paper introduces NoiseRefine, a novel method that tackles the high inference cost of diffusion models caused by sampling guidance (e.g., classifier-free guidance) by learning better initial noise. It achieves this by training a noise refining network to minimize the difference between images generated by unguided sampling from the refined noise and those produced by guided sampling from the input Gaussian noise. The approach utilizes a similarity loss in image space and a new Multi-Step Score Distillation (MSD) method for efficient training. Experiments demonstrate that this refined noise outperforms unguided sampling and achieves comparable quality to guided sampling at a much lower computational cost, supported by in-depth analysis of denoising trajectories.

**Strengths**: Most reviewers praised the novel and insightful idea of distilling guidance into initial noise, offering a fresh perspective on reducing guidance overhead. The experimental validation was strong, demonstrating superior performance over unguided sampling and comparable quality to guided sampling at lower cost. The thorough analysis, ablations, and interpretability insights were also highly valued.

**Weaknesses**: Initial concerns spanned several areas: conceptual clarity (e.g., why noise-to-noise mapping generalizes, unclear motivation, theoretical justification of MSD, overclaims as "orthogonal"), experimental evaluation (e.g., perceived quality gap to CFG, how to combine with other CFG methods, training cost analysis), and methodological aspects (e.g., potential loss of controllability).

**Decision**: The paper initially received a mixed set of ratings (8, 6, 4, 2), reflecting both strong enthusiasm for its novel idea and significant skepticism regarding certain aspects. Notably, even the reviewer who scored it a '4' acknowledged its novelty and plug-and-play nature. While initial concerns revolved around theoretical soundness, experimental evaluation, and methodological clarity, the authors provided a comprehensive and highly effective rebuttal during the discussion phase. This included clarifying motivation and methodological justification, conducting additional experiments, offering a detailed analysis of training costs, and significantly improving the manuscript's overall presentation. These efforts successfully addressed most of the raised issues; I thus recommend acceptance.

**Reviewer Concerns:**

Reviewer Concerns
Initial concerns spanned several areas: conceptual clarity (e.g., why noise-to-noise mapping generalizes, unclear motivation, theoretical justification of MSD, overclaims such as "orthogonal"), experimental evaluation (e.g., perceived quality gap to CFG, how to combine with other CFG methods, training cost analysis, seems unfair comparison with baseline), and methodological aspects (e.g., potential loss of controllability).

During the rebuttal phase, most of these concerns were comprehensively addressed:

*   **Improved Motivation and Methodological Justification**: The authors clarified the core motivation and provided more robust justification for their methodology.
*   **Clarity on Controllability**: The discussion around the controllability of the method was significantly improved, explaining how it can be maintained.
*   **Enhanced Justification of MSD**: The theoretical underpinnings and practical effectiveness of the Multi-Step Score Distillation (MSD) were more thoroughly justified, including quantitative evidence.
*   **Clarification on Generalization**: The authors explained why noise-to-noise mapping exhibits generalization properties and supported this with additional experiments and analysis.
*   **Discussion on Bridge Models**: A discussion was added regarding the feasibility and advantages of their approach compared to learning a mapping using a bridge model.
*   **Improved Inference Process Description**: The inference process was clarified for better understanding.
*   **Clarification of SD-Turbo Experiments**: The authors precisely articulated the goal and findings of their experiments on SD-Turbo.
*   **Fair Comparison with Baselines**: The authors provided clarifications and additional experiments to ensure a fair comparison with baseline methods, addressing initial concerns about potential unfairness.

Furthermore, the authors **acknowledged some initial "overclaims"** in the original manuscript during the rebuttal, such as rephrasing the description of their method as "orthogonal." They also **admitted certain trade-offs**, for instance, a **slower training cost** when compared to guidance distillation, though their method demonstrated improved FID scores. Additionally, they conceded that while their approach offers **better cost efficiency** compared to CFG, its performance is generally **comparable**, and in some cases, might be even **slightly inferior to CFG**.

However, **overall, I believe the paper's core strengths, such as its novel and insightful idea of distilling guidance into initial noise and its strong experimental results, are significant highlights.** The remaining weaknesses, now thoroughly discussed and contextualized, are minor in nature. Therefore, I conclude that **the paper merits acceptance.**

**Reviewer Scores:**

Reviewer 86wZ (Rating: 6 -> may be improved during discussion or keep the same), Reviewer ScWm (Rating: 4 -> improved during discussion), Reviewer a4W6 (Rating: 8 -> keep the same), Reviewer AzoV (Rating: 2 -> improved during discussion)

---

### Decision · Program_Chairs · 2026-01-26

Accept (Poster)